# Generalized Jensen-Shannon Divergence Loss for Learning with Noisy Labels

**Erik Englesson**
KTH
Stockholm, Sweden
engless@kth.se

**Hossein Azizpour**
KTH
Stockholm, Sweden
azizpour@kth.se

## Abstract

Prior works have found it beneficial to combine provably noise-robust loss functions *e.g.*, mean absolute error (MAE) with standard categorical loss function *e.g.* cross entropy (CE) to improve their learnability. Here, we propose to use Jensen-Shannon divergence as a noise-robust loss function and show that it interestingly interpolate between CE and MAE with a controllable mixing parameter. Furthermore, we make a crucial observation that CE exhibits lower consistency around noisy data points. Based on this observation, we adopt a generalized version of the Jensen-Shannon divergence for multiple distributions to encourage consistency around data points. Using this loss function, we show state-of-the-art results on both synthetic (CIFAR), and real-world (*e.g.* WebVision) noise with varying noise rates.

## 1 Introduction

Labeled datasets, even the systematically annotated ones, contain noisy labels [1]. Therefore, designing noise-robust learning algorithms are crucial for the real-world tasks. An important avenue to tackle noisy labels is to devise noise-robust loss functions [2, 3, 4, 5]. Similarly, in this work, we propose two new noise-robust loss functions based on two central observations as follows.

Observation I: *Provably-robust loss functions can underfit the training data* [2, 3, 4, 5].
Observation II: *Standard networks show low consistency around noisy data points* [1], see Figure 1.

We first propose to use Jensen-Shannon divergence (JS) as a loss function, which we crucially show interpolates between the noise-robust mean absolute error (MAE) and the cross entropy (CE) that better fits the data through faster convergence. Figure 2 illustrates the CE-MAE interpolation. Regarding Observation II, we adopt the generalized version of Jensen-Shannon divergence (GJS) to encourage predictions on perturbed inputs to be consistent, see Figure 3. Notably, Jensen-Shannon divergence has previously shown promise for test-time robustness to domain shift [6], here we further argue for its *training-time* robustness to *label noise*. The key contributions of this work[2] are:

- We make a novel observation that a network predictions' consistency is reduced for noisy-labeled data when overfitting to noise, which motivates the use of consistency regularization.

- We propose using Jensen-Shannon divergence (JS) and its multi-distribution generalization (GJS) as loss functions for learning with noisy labels. We relate JS to loss functions that are based on the noise-robustness theory of Ghosh *et al.* [2]. In particular, we prove that JS generalizes CE and MAE. Furthermore, we prove that GJS generalizes JS by incorporating consistency regularization in a single principled loss function.

- We provide an extensive set of empirical evidences on several datasets, noise types and rates. They show state-of-the-art results and give in-depth studies of the proposed losses.

---

[1] we call a network *consistent* around a sample ($x$) if it predicts the same class for $x$ and its perturbations ($\tilde{x}$).
[2] implementation available at `https://github.com/ErikEnglesson/GJS`

35th Conference on Neural Information Processing Systems (NeurIPS 2021).

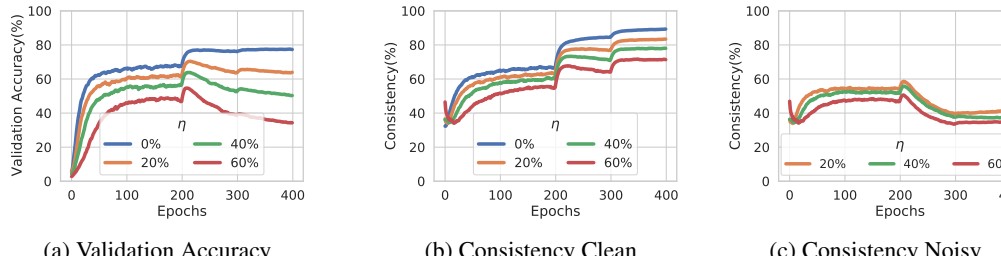

|  | (a) Validation Accuracy | (b) Consistency Clean | (c) Consistency Noisy |

Figure 1: **Evolution of a trained network's consistency as it overfits to noise using CE loss.** Here we plot the evolution of the validation accuracy (a) and network's consistency (as measured by GJS) on clean (b) and noisy (c) examples of the training set of CIFAR-100 for varying symmetric noise rates when learning with the cross-entropy loss. The consistency of the learnt function and the accuracy closely correlate. This suggests that enforcing consistency may help avoid fitting to noise. Furthermore, the consistency is degraded more significantly for the noisy data points.

## 2 Generalized Jensen-Shannon Divergence

We propose two loss functions, the Jensen-Shannon divergence (JS) and its multi-distribution generalization (GJS). In this section, we first provide background and two observations that motivate our proposed loss functions. This is followed by definition of the losses, and then we show that JS generalizes CE and MAE similarly to other robust loss functions. Finally, we show how GJS generalizes JS to incorporate consistency regularization into a single principled loss function. We provide proofs of all theorems, propositions, and remarks in this section in Appendix C.

### 2.1 Background & Motivation

**Supervised Classification.** Assume a general function class[3] $\mathcal{F}$ where each $f \in \mathcal{F}$ maps an input $\boldsymbol{x} \in \mathbb{X}$ to the probability simplex $\Delta^{K-1}$, *i.e.* to a categorical distribution over $K$ classes $y \in \mathbb{Y} = \{1, 2, \ldots, K\}$. We seek $f^* \in \mathcal{F}$ that minimizes a risk $R_{\mathcal{L}}(f) = \mathbb{E}_{\mathcal{D}}[\mathcal{L}(\boldsymbol{e}^{(y)}, f(\boldsymbol{x}))]$, for some loss function $\mathcal{L}$ and joint distribution $\mathcal{D}$ over $\mathbb{X} \times \mathbb{Y}$, where $\boldsymbol{e}^{(y)}$ is a $K$-vector with one at index $y$ and zero elsewhere. In practice, $\mathcal{D}$ is unknown and, instead, we use $\mathcal{S} = \{(\boldsymbol{x}_i, y_i)\}_{i=1}^{N}$ which are independently sampled from $\mathcal{D}$ to minimize an empirical risk $\frac{1}{N} \sum_{i=1}^{N} \mathcal{L}(\boldsymbol{e}^{(y_i)}, f(\boldsymbol{x}_i))$.

**Learning with Noisy Labels.** In this work, the goal is to learn from a noisy training distribution $\mathcal{D}_\eta$ where the labels are changed, with probability $\eta$, from their true distribution $\mathcal{D}$. The noise is called *instance-dependent* if it depends on the input, *asymmetric* if it dependents on the true label, and *symmetric* if it is independent of both $\boldsymbol{x}$ and $y$. Let $f_\eta^*$ be the optimizer of the noisy distribution risk $R_{\mathcal{L}}^\eta(f)$. A loss function $\mathcal{L}$ is then called *robust* if $f_\eta^*$ also minimizes $R_{\mathcal{L}}$. The MAE loss ($\mathcal{L}_{MAE}(\boldsymbol{e}^{(y)}, f(\boldsymbol{x})) := \|\boldsymbol{e}^{(y)} - f(\boldsymbol{x})\|_1$) is robust but not CE [2].

**Issue of Underfitting.** Several works propose such robust loss functions and demonstrate their efficacy in preventing noise fitting [2, 3, 4, 5]. However, all those works have observed slow convergence of such robust loss functions leading to underfitting. This can be contrasted with CE that has fast convergence but overfits to noise. Ghosh *et al.* [2] mentions slow convergence of MAE and GCE [3] extensively analyzes the undefitting thereof. SCE [4] reports similar problems for the reverse cross entropy and proposes a linear combination with CE. Finally, Ma *et al.* [5] observe the same problem and consider a combination of "active" and "passive" loss functions.

**Consistency Regularization.** This encourages a network to have consistent predictions for different perturbations of the same image, which has mainly been used for semi-supervised learning [7].

**Motivation.** In Figure 1, we show the validation accuracy and a measure of consistency during training with the CE loss for varying amounts of noise. First, we note that training with CE loss eventually overfits to noisy labels. Figure 1a, indicates that the higher the noise rate, the more accuracy drop when it starts to overfit to noise. Figure 1(b-c) shows the consistency of predictions for correct and noisy labeled examples of the training set, with the consistency measured as the ratio of examples that have the same class prediction for two perturbations of the same image, see

---

[3] *e.g.* softmax neural network classifiers in this work

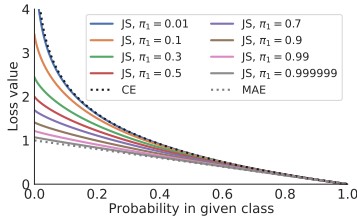

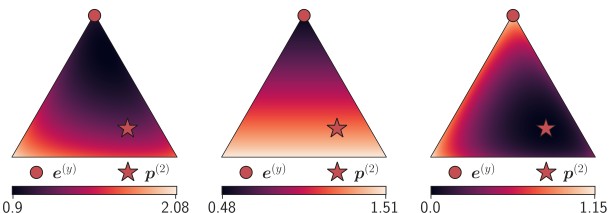

Figure 2: **JS loss generalizes CE and MAE.** The Jensen-Shannon loss ($\mathcal{L}_{\mathrm{JS}}$) for different values of the hyperparameter $\pi_1$. The JS loss interpolates between CE and MAE. For low values of $\pi_1$, $\mathcal{L}_{\mathrm{JS}}$ behaves like CE and for increasing values of $\pi_1$ it behaves more like the noise robust MAE loss.

Figure 3: **GJS Dissection for M=K=3:** The decomposition of $\mathcal{L}_{\mathrm{GJS}}$ (left) into a JS term (middle) and a consistency term (right) from Proposition 2. Each point in the simplex correspond to a $\boldsymbol{p}^{(3)} \in \Delta^2$, where the color represents the value of the loss at that point. It can be seen that there are two ways to minimize $\mathcal{L}_{\mathrm{GJS}}$, either by making the predictions similar to the label (middle) or similar to the other predictions (right) to increase consistency. To better highlight the variations of the losses, each loss has its own range of values.

Appendix B.6 for more details. A clear correlation is observed between the accuracy and consistency of the noisy examples. This suggests that maximizing consistency of predictions may improve the robustness to noise. Next, we define simple loss functions that (i) encourage consistency around data points and (ii) alleviate the "issue of underfitting" by interpolating between CE and MAE.

## 2.2 Definitions

$\boldsymbol{D_{\mathrm{JS}}}$. Let $\boldsymbol{p}^{(1)}, \boldsymbol{p}^{(2)} \in \Delta^{K-1}$ have corresponding weights $\boldsymbol{\pi} = [\pi_1, \pi_2]^T \in \Delta$. Then, the Jensen-Shannon divergence between $\boldsymbol{p}^{(1)}$ and $\boldsymbol{p}^{(2)}$ is

$$D_{\mathrm{JS}_{\boldsymbol{\pi}}}(\boldsymbol{p}^{(1)}, \boldsymbol{p}^{(2)}) \coloneqq H(\boldsymbol{m}) - \pi_1 H(\boldsymbol{p}^{(1)}) - \pi_2 H(\boldsymbol{p}^{(2)}) = \pi_1 D_{\mathrm{KL}}(\boldsymbol{p}^{(1)} \| \boldsymbol{m}) + \pi_2 D_{\mathrm{KL}}(\boldsymbol{p}^{(2)} \| \boldsymbol{m}) \ (1)$$

with $H$ the Shannon entropy, and $\boldsymbol{m} = \pi_1 \boldsymbol{p}^{(1)} + \pi_2 \boldsymbol{p}^{(2)}$. Unlike Kullback–Leibler divergence ($D_{\mathrm{KL}}(\boldsymbol{p}^{(1)} \| \boldsymbol{p}^{(2)})$) or cross entropy (CE), JS is symmetric, bounded, does not require absolute continuity, and has a crucial weighting mechanism ($\boldsymbol{\pi}$), as we will see later.

$\boldsymbol{D_{\mathrm{GJS}}}$. Similar to $D_{\mathrm{KL}}$, $D_{\mathrm{JS}}$ satisfies $D_{\mathrm{JS}_{\boldsymbol{\pi}}}(\boldsymbol{p}^{(1)}, \boldsymbol{p}^{(2)}) \geq 0$, with equality iff $\boldsymbol{p}^{(1)} = \boldsymbol{p}^{(2)}$. For $D_{\mathrm{JS}}$, this is derived from Jensen's inequality for the concave Shannon entropy. This property holds for finite number of distributions and motivates a generalization of $D_{\mathrm{JS}}$ to multiple distributions [8]:

$$D_{\mathrm{GJS}_{\boldsymbol{\pi}}}(\boldsymbol{p}^{(1)}, \ldots, \boldsymbol{p}^{(M)}) \coloneqq H\Big(\sum_{i=1}^{M} \pi_i \boldsymbol{p}^{(i)}\Big) - \sum_{i=1}^{M} \pi_i H(\boldsymbol{p}^{(i)}) = \sum_{i=1}^{M} \pi_i D_{\mathrm{KL}}\Big(\boldsymbol{p}^{(i)} \Big\| \sum_{j=1}^{M} \pi_j \boldsymbol{p}^{(j)}\Big) \ (2)$$

where $M$ is the number of distributions, and $\boldsymbol{\pi} = [\pi_1, \ldots, \pi_M]^T \in \Delta^{M-1}$.

**Loss functions.** We aim to use $D_{\mathrm{JS}}$ and $D_{\mathrm{GJS}}$ divergences, to measure deviation of the predictive distribution(s), $f(\boldsymbol{x})$, from the target distribution, $\boldsymbol{e}^{(y)}$. Without loss of generality, hereafter, we dedicate $\boldsymbol{p}^{(1)}$ to denote the target distribution. JS loss, therefore, can take the form of $D_{\mathrm{JS}_{\boldsymbol{\pi}}}(\boldsymbol{e}^{(y)}, f(\boldsymbol{x}))$. Generalized JS loss is a less straight-forward construction since $D_{\mathrm{GJS}}$ can accommodate more predictive distributions. While various choices can be made for these distributions, in this work, we consider predictions associated with different random perturbations of a sample, denoted by $\mathcal{A}(\boldsymbol{x})$. This choice, as shown later, implies an interesting analogy to *consistency regularization*. The choice, also entails no distinction between the $M-1$ predictive distributions. Therefore, we consider $\pi_2 = \cdots = \pi_M = \frac{1-\pi_1}{M-1}$ in all our experiments. Finally, we scale the loss functions by a constant factor $Z = -(1-\pi_1)\log(1-\pi_1)$. As we will see later, the role of this scaling is merely to strengthen the already existing and desirable behaviors of these losses as $\pi_1$ approaches zero and one. Formally, we have JS and GJS losses:

$$\mathcal{L}_{\mathrm{JS}}(y, f, \boldsymbol{x}) \coloneqq \frac{D_{\mathrm{JS}_{\boldsymbol{\pi}}}(\boldsymbol{e}^{(y)}, f(\tilde{\boldsymbol{x}}))}{Z}, \quad \mathcal{L}_{\mathrm{GJS}}(y, f, \boldsymbol{x}) \coloneqq \frac{D_{\mathrm{GJS}_{\boldsymbol{\pi}}}(\boldsymbol{e}^{(y)}, f(\tilde{\boldsymbol{x}}^{(2)}), \ldots, f(\tilde{\boldsymbol{x}}^{(M)}))}{Z} \ (3)$$

with $\tilde{\boldsymbol{x}}^{(i)} \sim \mathcal{A}(\boldsymbol{x})$. Next, we study the connection between JS and losses which are based on the robustness theory of Ghosh *et al.* [2].

## 2.3 JS's Connection to Robust Losses

Cross Entropy (CE) is the prevalent loss function for deep classifiers with remarkable successes. However, CE is prone to fitting noise [9]. On the other hand, Mean Absolute Error (MAE) is theoretically noise-robust [2]. Evidently, standard optimization algorithms struggle to minimize MAE, especially for more challenging datasets *e.g.* CIFAR-100 [3, 5]. Therefore, there have been several proposals that combine CE and MAE, such as Generalized CE (GCE) [3], Symmetric CE (SCE) [4], and Normalized CE (NCE+MAE) [5]. The rationale is for CE to help with the learning dynamics of MAE. Next, we show JS has CE and MAE as its asymptotes w.r.t. $\boldsymbol{\pi}_1$.

**Proposition 1.** *Let $\boldsymbol{p} \in \Delta^{K-1}$, then*

$$\lim_{\pi_1 \to 0} \mathcal{L}_{\mathrm{JS}}(\boldsymbol{e}^{(y)}, \boldsymbol{p}) = H(\boldsymbol{e}^{(y)}, \boldsymbol{p}), \qquad \lim_{\pi_1 \to 1} \mathcal{L}_{\mathrm{JS}}(\boldsymbol{e}^{(y)}, \boldsymbol{p}) = \frac{1}{2}\|\boldsymbol{e}^{(y)} - \boldsymbol{p}\|_1$$

*where $H(\boldsymbol{e}^{(y)}, \boldsymbol{p})$ is the cross entropy of $\boldsymbol{e}^{(y)}$ relative to $\boldsymbol{p}$.*

Figure 2 depicts how JS interpolates between CE and MAE for $\pi_1 \in (0, 1)$. The proposition reveals an interesting connection to state-of-the-art robust loss functions, however, there are important differences. SCE is not bounded (so it cannot be used in Theorem 1), and GCE is not symmetric, while JS and MAE are both symmetric and bounded. In Appendix B.3, we perform a dissection to better understand how these properties affect learning with noisy labels. GCE is most similar to JS and is compared further in Appendix B.4.

A crucial difference to these other losses is that JS naturally extends to multiple predictive distributions (GJS). Next, we show how GJS generalizes JS by incorporating consistency regularization.

## 2.4 GJS's Connection to Consistency Regularization

In Figure 1, it was shown how the consistency of the noisy labeled examples was reduced when the network overfitted to noise. The following proposition shows how GJS naturally encourages consistency in a single principled loss function.

**Proposition 2.** *Let $\boldsymbol{p}^{(2)}, \ldots, \boldsymbol{p}^{(M)} \in \Delta^{K-1}$ with $M \geq 3$ and $\bar{\boldsymbol{p}}_{>1} = \frac{\sum_{j=2}^{M} \pi_j \boldsymbol{p}^{(j)}}{1-\pi_1}$, then*

$$\mathcal{L}_{\mathrm{GJS}}(\boldsymbol{e}^{(y)}, \boldsymbol{p}^{(2)}, \ldots, \boldsymbol{p}^{(M)}) = \mathcal{L}_{\mathrm{JS}_{\boldsymbol{\pi}'}}(\boldsymbol{e}^{(y)}, \bar{\boldsymbol{p}}_{>1}) + (1-\pi_1)\mathcal{L}_{\mathrm{GJS}_{\boldsymbol{\pi}''}}(\boldsymbol{p}^{(2)}, \ldots, \boldsymbol{p}^{(M)})$$

*where $\boldsymbol{\pi}' = [\pi_1, 1-\pi_1]^T$ and $\boldsymbol{\pi}'' = \frac{[\pi_2, \ldots, \pi_M]^T}{(1-\pi_1)}$.*

Importantly, Proposition 2 shows that GJS can be decomposed into two terms: 1) a JS term between the label and the mean prediction $\bar{\boldsymbol{p}}_{>1}$, and 2) a GJS term, but without the label. Figure 3 illustrates the effect of this decomposition. The first term, similarly to the standard JS loss, encourages the predictions' mean to be closer to the label (Figure 3 middle). However, the second term encourages all predictions to be similar, that is, *consistency regularization* (Figure 3 right).

## 2.5 Noise Robustness

Here, the robustness properties of JS and GJS are analyzed in terms of lower ($B_L$) and upper bounds ($B_U$) for the following theorem, which generalizes the results by Zhang *et al.* [3] to any bounded loss function, even with multiple predictive distributions.

**Theorem 1.** *Under symmetric noise with $\eta < \frac{K-1}{K}$, if $B_L \leq \sum_{i=1}^{K} \mathcal{L}(\boldsymbol{e}^{(i)}, \boldsymbol{x}, f) \leq B_U, \forall \boldsymbol{x}, f$ is satisfied for a loss $\mathcal{L}$, then*

$$0 \leq R_{\mathcal{L}}^{\eta}(f^*) - R_{\mathcal{L}}^{\eta}(f_{\eta}^*) \leq \eta \frac{B_U - B_L}{K-1}, \quad and \quad -\frac{\eta(B_U - B_L)}{K-1-\eta K} \leq R_{\mathcal{L}}(f^*) - R_{\mathcal{L}}(f_{\eta}^*) \leq 0,$$

A tighter bound $B_U - B_L$, implies a smaller worst case risk difference of the optimal classifiers (robust when $B_U = B_L$). Importantly, while $\mathcal{L}(\boldsymbol{e}^{(i)}, \boldsymbol{x}, f) = \mathcal{L}(\boldsymbol{e}^{(i)}, f(\boldsymbol{x}))$ usually, this subtle distinction is useful for losses with multiple predictive distributions, see Equation 3. In Theorem 2 in Appendix C.3, we further prove the robustness of the proposed losses to asymmetric noise.

For losses with multiple predictive distributions, the bounds in Theorem 1 and 2 must hold for any $\boldsymbol{x}$ and $f$, *i.e.*, for any combination of $M-1$ categorical distributions on $K$ classes. Proposition 3 provides such bounds for GJS.

**Proposition 3.** GJS *loss with $M \leq K+1$ satisfies $B_L \leq \sum_{k=1}^{K} \mathcal{L}_{\text{GJS}}(e^{(k)}, p^{(2)}, \dots, p^{(M)}) \leq B_U$ for all $p^{(2)}, \dots, p^{(M)} \in \Delta^{K-1}$, with the following bounds*

$$B_L = \sum_{k=1}^{K} \mathcal{L}_{\text{GJS}}(e^{(k)}, u, \dots, u), \quad B_U = \sum_{k=1}^{K} \mathcal{L}_{\text{GJS}}(e^{(k)}, e^{(1)}, \dots, e^{(M-1)})$$

*where $u \in \Delta^{K-1}$ is the uniform distribution.*

Note the bounds for the JS loss is a special case of Proposition 3 for $M = 2$.

**Remark 1.** $\mathcal{L}_{\text{JS}}$ *and $\mathcal{L}_{\text{GJS}}$ are robust ($B_L = B_U$) in the limit of $\pi_1 \to 1$.*

Remark 1 is intuitive from Section 2.3 which showed that $\mathcal{L}_{\text{JS}}$ is equivalent to the robust MAE in this limit and that the consistency term in Proposition 2 vanishes.

In Proposition 3, the lower bound ($B_L$) is the same for JS and GJS. However, the upper bound ($B_U$) increases for more distributions, which makes JS have a tighter bound than GJS in Theorem 1 and 2. In Proposition 4, we show that JS and GJS have the same bound for the risk difference, given an assumption based on Figure 1 that the optimal classifier on clean data ($f^*$) is at least as consistent as the optimal classifier on noisy data ($f_\eta^*$).

**Proposition 4.** $\mathcal{L}_{\text{JS}}$ *and $\mathcal{L}_{\text{GJS}}$ have the same risk bounds in Theorem 1 and 2 if $\mathbb{E}_{\mathbf{x}}[\mathcal{L}_{\text{GJS}_{\pi''}}^{f^*}(p^{(2)}, \dots, p^{(M)})] \leq \mathbb{E}_{\mathbf{x}}[\mathcal{L}_{\text{GJS}_{\pi''}}^{f_\eta^*}(p^{(2)}, \dots, p^{(M)})]$, where $\mathcal{L}_{\text{GJS}_{\pi''}}^{f}(p^{(2)}, \dots, p^{(M)})$ is the consistency term from Proposition 2.*

## 3 Related Works

Interleaved in the previous sections, we covered most-related works to us, *i.e.* the avenue of identification or construction of *theoretically-motivated robust loss functions* [2, 3, 4, 5]. These works, similar to this paper, follow the theoretical construction of Ghosh *et al.* [2]. Furthermore, Liu&Guo [10] use "peer prediction" to propose a new family of robust loss functions. Different to these works, here, we propose loss functions based on $D_{\text{JS}}$ which holds various desirable properties of those prior works while exhibiting novel ties to consistency regularization; a recent important regularization technique.

Next, we briefly cover other lines of work. A more thorough version can be found in Appendix D.

A direction, that similar to us does not alter training, *reweights a loss function* by confusion matrix [11, 12, 13, 14, 15]. Assuming a class-conditional noise model, loss correction is theoretically motivated and perfectly orthogonal to noise-robust losses.

*Consistency regularization* is a recent technique that imposes smoothness in the learnt function for semi-supervised learning [7] and recently for noisy data [16]. These works use different complex pipelines for such regularization. GJS encourages consistency in a simple way that exhibits other desirable properties for learning with noisy labels. Importantly, Jensen-Shannon-based consistency loss functions have been used to improve test-time robustness to image corruptions [6] and adversarial examples [17], which further verifies the general usefulness of GJS. In this work, we study such loss functions for a different goal: *training-time label-noise robustness*. In this context, our thorough analytical and empirical results are, to the best of our knowledge, novel.

Recently, loss functions with *information-theoretic* motivations have been proposed [18, 19]. JS, with an apparent information-theoretic interpretation, has a strong connection to those. Especially, the latter is a close concurrent work studying JS and other divergences from the family of f-divergences [20]. However, in this work, we consider a generalization to more than two distributions and study the role of $\pi_1$, which they treat as a constant ($\pi_1 = \frac{1}{2}$). These differences lead to improved performance and novel theoretical results, e.g., Proposition 1 and 2. Lastly, another generalization of JS was recently presented by Nielsen [21], where the arithmetic mean is generalized to abstract means.

## 4 Experiments

This section, first, empirically investigates the effectiveness of the proposed losses for learning with noisy labels on synthetic (Section 4.1) and real-world noise (Section 4.2). This is followed by several experiments and ablation studies (Section 4.3) to shed light on the properties of JS and GJS through empirical substantiation of the theories and claims provided in Section 2. All these additional experiments are done on the more challenging CIFAR-100 dataset.

Table 1: **Synthetic Noise Benchmark on CIFAR.** We *reimplement* other noise-robust loss functions into the *same learning setup* and ResNet-34, including label smoothing (LS), Bootstrap (BS), Symmetric CE (SCE), Generalized CE (GCE), and Normalized CE (NCE+RCE). We used *same hyperparameter optimization budget and mechanism* for all the prior works and ours. Mean test accuracy and standard deviation are reported from five runs and the statistically-significant top performers are boldfaced. The thorough analysis is evident from the higher performance of CE in our setup compared to prior works. GJS achieves state-of-the-art results for different noise rates, types, and datasets. Generally, GJS's efficacy is more evident for the more challenging CIFAR-100 dataset.

| Dataset | Method | No Noise | Symmetric Noise Rate | | | | Asymmetric Noise Rate | |
|---|---|---|---|---|---|---|---|---|
| | | 0% | 20% | 40% | 60% | 80% | 20% | 40% |
| CIFAR-10 | CE | **95.77 ± 0.11** | 91.63 ± 0.27 | 87.74 ± 0.46 | 81.99 ± 0.56 | 66.51 ± 1.49 | 92.77 ± 0.24 | 87.12 ± 1.21 |
| | BS | 94.58 ± 0.25 | 91.68 ± 0.32 | 89.23 ± 0.16 | 82.65 ± 0.57 | 16.97 ± 6.36 | 93.06 ± 0.25 | 88.87 ± 1.06 |
| | LS | 95.64 ± 0.12 | 93.51 ± 0.20 | 89.90 ± 0.20 | 83.96 ± 0.58 | 67.35 ± 2.71 | 92.94 ± 0.17 | 88.10 ± 0.50 |
| | SCE | **95.75 ± 0.16** | 94.29 ± 0.14 | 92.72 ± 0.25 | 89.26 ± 0.37 | **80.68 ± 0.42** | 93.48 ± 0.31 | 84.98 ± 0.76 |
| | GCE | **95.75 ± 0.14** | 94.24 ± 0.18 | 92.82 ± 0.11 | 89.37 ± 0.27 | **79.19 ± 2.04** | 92.83 ± 0.36 | 87.00 ± 0.99 |
| | NCE+RCE | 95.36 ± 0.09 | 94.27 ± 0.18 | 92.03 ± 0.31 | 87.30 ± 0.35 | 77.89 ± 0.61 | **93.87 ± 0.03** | 86.83 ± 0.84 |
| | JS | **95.89 ± 0.10** | 94.52 ± 0.21 | 93.01 ± 0.22 | 89.64 ± 0.15 | 76.06 ± 0.85 | 92.18 ± 0.31 | 87.99 ± 0.55 |
| | GJS | **95.91 ± 0.09** | **95.33 ± 0.18** | **93.57 ± 0.16** | **91.64 ± 0.22** | 79.11 ± 0.31 | **93.94 ± 0.25** | **89.65 ± 0.37** |
| CIFAR-100 | CE | 77.60 ± 0.17 | 65.74 ± 0.22 | 55.77 ± 0.83 | 44.42 ± 0.84 | 10.74 ± 4.08 | 66.85 ± 0.32 | 49.45 ± 0.37 |
| | BS | 77.65 ± 0.29 | 72.92 ± 0.50 | 68.52 ± 0.54 | 53.80 ± 1.76 | 13.83 ± 4.41 | 73.79 ± 0.43 | **64.67 ± 0.69** |
| | LS | 78.60 ± 0.04 | 74.88 ± 0.15 | 68.41 ± 0.20 | 54.58 ± 0.47 | 26.98 ± 1.07 | 73.17 ± 0.46 | 57.20 ± 0.85 |
| | SCE | 78.29 ± 0.24 | 74.21 ± 0.37 | 68.23 ± 0.29 | 59.28 ± 0.58 | 26.80 ± 1.11 | 70.86 ± 0.44 | 51.12 ± 0.37 |
| | GCE | 77.65 ± 0.17 | 75.02 ± 0.24 | 71.54 ± 0.39 | 65.21 ± 0.16 | **49.68 ± 0.84** | 72.13 ± 0.39 | 51.50 ± 0.71 |
| | NCE+RCE | 74.66 ± 0.21 | 72.39 ± 0.24 | 68.79 ± 0.29 | 62.18 ± 0.35 | 31.63 ± 3.59 | 71.35 ± 0.16 | 57.80 ± 0.52 |
| | JS | 77.95 ± 0.39 | 75.41 ± 0.28 | 71.12 ± 0.30 | 64.36 ± 0.34 | 45.05 ± 0.93 | 71.70 ± 0.36 | 49.36 ± 0.25 |
| | GJS | **79.27 ± 0.29** | **78.05 ± 0.25** | **75.71 ± 0.25** | **70.15 ± 0.30** | 44.49 ± 0.53 | **74.60 ± 0.47** | 63.70 ± 0.22 |

**Experimental Setup.** We use ResNet 34 and 50 for experiments on CIFAR and WebVision datasets respectively and optimize them using SGD with momentum. The complete details of the training setup can be found in Appendix A. Most importantly, we take three main measures to ensure a fair and reliable comparison throughout the experiments: 1) we reimplement all the loss functions we compare with in a single shared learning setup, 2) we use the same hyperparameter optimization budget and mechanism for all the prior works and ours, and 3) we train and evaluate five networks for individual results, where in each run the synthetic noise, network initialization, and data-order are differently randomized. The thorough analysis is evident from the higher performance of CE in our setup compared to prior works. Where possible, we report mean and standard deviation and denote the statistically-significant top performers with student t-test.

## 4.1 Synthetic Noise Benchmarks: CIFAR

Here, we evaluate the proposed loss functions on the CIFAR datasets with two types of synthetic noise: symmetric and asymmetric. For symmetric noise, the labels are, with probability $\eta$, re-sampled from a uniform distribution over all labels. For asymmetric noise, we follow the standard setup of Patrini *et al.* [22]. For CIFAR-10, the labels are modified, with probability $\eta$, as follows: *truck → automobile*, *bird → airplane*, *cat ↔ dog*, and *deer → horse*. For CIFAR-100, labels are, with probability $\eta$, cycled to the next sub-class of the same "super-class", *e.g.* the labels of super-class "vehicles 1" are modified as follows: *bicycle → bus → motorcycle → pickup truck → train → bicycle*.

We compare with other noise-robust loss functions such as label smoothing (LS) [23], Bootstrap (BS) [24], Symmetric Cross-Entropy (SCE) [4], Generalized Cross-Entropy (GCE) [3], and the NCE+RCE loss of Ma *et al.* [5]. Here, we do not compare to methods that propose a full pipeline since, first, a conclusive comparison would require re-implementation and individual evaluation of several components and second, robust loss functions can be considered orthogonal to them.

**Results.** Table 1 shows the results for symmetric and asymmetric noise on CIFAR-10 and CIFAR-100. GJS performs similarly or better than other methods for different noise rates, noise types, and data sets. Generally, GJS's efficacy is more evident for the more challenging CIFAR-100 dataset. For example, on 60% uniform noise on CIFAR-100, the difference between GJS and the second best (GCE) is 4.94 percentage points, while our results on 80% noise is lower than GCE. We attribute this to the high sensitivity of the results to the hyperparameter settings in such a high-noise rate which are also generally unrealistic (WebVision has ∼20%). The performance of JS is consistently

Table 2: **Real-world Noise Benchmark on WebVision.** Mean test accuracy and standard deviation from five runs are reported for the validation sets of (mini) WebVision and ILSVRC12. GJS with two networks correspond to the mean prediction of two independently trained GJS networks with different seeds for data augmentation and weight initialization. Here, GJS uses $Z = 1$. Results marked with † are from Zheltonozhskii *et al.* [26].

| Method | Architecture | Augmentation | Networks | WebVision | | ILSVRC12 | |
|---|---|---|---|---|---|---|---|
| | | | | Top 1 | Top 5 | Top 1 | Top 5 |
| ELR+ [27]† | Inception-ResNet-V2 | Mixup | 2 | 77.78 | **91.68** | 70.29 | 89.76 |
| DivideMix [16]† | Inception-ResNet-V2 | Mixup | 2 | 77.32 | **91.64** | **75.20** | 90.84 |
| DivideMix [16]† | ResNet-50 | Mixup | 2 | $76.32 \pm 0.36$ | $90.65 \pm 0.16$ | $74.42 \pm 0.29$ | $\mathbf{91.21 \pm 0.12}$ |
| CE | ResNet-50 | ColorJitter | 1 | $70.69 \pm 0.66$ | $88.64 \pm 0.17$ | $67.32 \pm 0.57$ | $88.00 \pm 0.49$ |
| JS | ResNet-50 | ColorJitter | 1 | $74.56 \pm 0.32$ | $91.09 \pm 0.08$ | $70.36 \pm 0.12$ | $90.60 \pm 0.09$ |
| GJS | ResNet-50 | ColorJitter | 1 | $77.99 \pm 0.35$ | $90.62 \pm 0.28$ | $74.33 \pm 0.46$ | $90.33 \pm 0.20$ |
| GJS | ResNet-50 | ColorJitter | 2 | $\mathbf{79.28 \pm 0.24}$ | $91.22 \pm 0.30$ | $\mathbf{75.50 \pm 0.17}$ | $\mathbf{91.27 \pm 0.26}$ |

similar to the top performance of the prior works across different noise rates, types and datasets. In Section 4.3, we substantiate the importance of the consistency term, identified in Proposition 2, when going from JS to GJS that helps with the learning dynamics and reduce the susceptibility to noise. In Appendix B.1, we provide results for GJS on instance-dependent synthetic noise [25]. Next, we test the proposed losses on a naturally-noisy dataset to see their efficacy in a real-world scenario.

## 4.2   Real-World Noise Benchmark: WebVision

WebVision v1 is a large-scale image dataset collected by crawling Flickr and Google, which resulted in an estimated 20% of noisy labels [28]. There are 2.4 million images of the same thousand classes as ILSVRC12. Here, we use a smaller version called mini WebVision [29] consisting of the first 50 classes of the Google subset. We compare CE, JS, and GJS on WebVision following the same rigorous procedure as for the synthetic noise. However, upon request by the reviewers, we also compare with the reported results of some state-of-the-art elaborate techniques. This comparison deviates from our otherwise systematic analysis.

**Results.**   Table 2, as the common practice, reports the performances on the validation sets of WebVision and ILSVRC12 (first 50 classes). Both JS and GJS exhibit large margins with standard CE, especially for top-1 accuracy. Top-5 accuracy, due to its admissibility of wrong top predictions, can obscure the susceptibility to noise-fitting and thus indicates smaller but still significant improvements.

The two state-of-the-art methods on this dataset were DivideMix [16] and ELR+ [27]. Compared to our setup, both these methods use a stronger network (Inception-ResNet-V2 vs ResNet-50), stronger augmentations (Mixup vs color jittering) and co-train two networks. Furthermore, ELR+ uses an exponential moving average of weights and DivideMix treats clean and noisy labeled examples differently after separating them using Gaussian mixture models. Despite these differences, GJS performs as good or better in terms of top-1 accuracy on WebVision and significantly outperforms ELR+ on ILSVRC12 (70.29 vs 74.33). The importance of these differences becomes apparent as 1) the top-1 accuracy for DivideMix degrades when using ResNet-50, and 2) the performance of GJS improves by adding one of their components, *i.e.* the use of two networks. We train an ensemble of two independent networks with the GJS loss and average their predictions (last row of Table 2). This simple extension, which requires no change in the training code, gives significant improvements. To the best of our knowledge, this is the highest reported top-1 accuracy on WebVision and ILSVRC12 when no pre-training is used.

In Appendix B.2, we show state-of-the-art results when using GJS on two other real-world noisy datasets: ANIMAL-10N [30] and Food-101N [31].

So far, the experiments demonstrated the robustness of the proposed loss function (regarding Proposition 3) via the significant improvement of the final accuracy on noisy datasets. While this was central and informative, it is also important to investigate whether this improvement comes from the theoretical properties that were argued for JS and GJS. In what follows, we devise several such experiments, in an effort to substantiate the theoretical claims and conjectures.

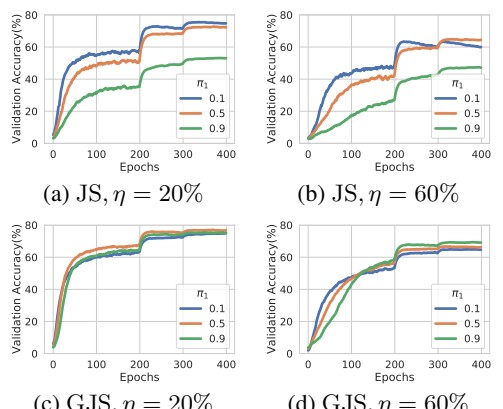

(a) JS, $\eta = 20\%$      (b) JS, $\eta = 60\%$

(c) GJS, $\eta = 20\%$     (d) GJS, $\eta = 60\%$

Figure 4: **Effect of $\pi_1$.** Validation accuracy of JS and GJS during training with symmetric noise on CIFAR100. From Proposition 1, JS behaves like CE and MAE for low and high values of $\pi_1$, respectively. The signs of noise-fitting for $\pi_1 = 0.1$ on 60% noise (b), and slow learning of $\pi_1 = 0.9$ (a-b), show this in practice. The GJS loss does not exhibit overfitting for low values of $\pi_1$ and learns quickly for large values of $\pi_1$ (c-d).

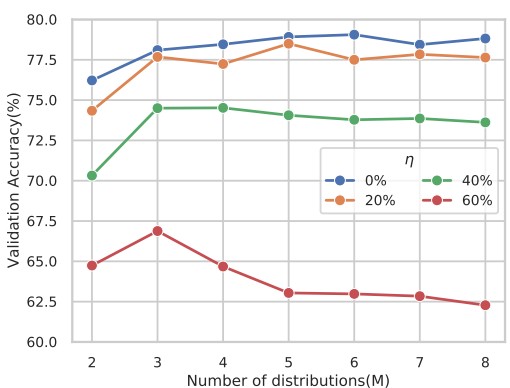

Figure 5: **Effect of M.** Validation accuracy for increasing number of distributions ($M$) and different symmetric noise rates on CIFAR-100 with $\pi_1 = \frac{1}{2}$. For all noise rates, using three instead of two distributions results in a higher accuracy. Going beyond three distributions is only helpful for lower noise rates. For simplicity we use $M = 3$ (corresponding to two augmentations) for all of our experiments.

### 4.3 Towards a Better Understanding of the Jensen-Shannon-based Loss Functions

Here, we study the behavior of the losses for different distribution weights $\pi_1$, number of distributions $M$, and epochs. We also provide insights on why GJS performs better than JS.

**How does $\pi_1$ control the trade-off of robustness and learnability?** In Figure 4, we plot the validation accuracy during training for both JS and GJS at different values of $\pi_1$ and noise rates $\eta$. From Proposition 1, we expect JS to behave as CE for low values of $\pi_1$ and as MAE for larger values of $\pi_1$. Figure 4 (a-b) confirms this. Specifically, $\pi_1 = 0.1$ learns quickly and performs well for low noise but overfits for $\eta = 0.6$ (characteristic of non-robust CE), on the other hand, $\pi_1 = 0.9$ learns slowly but is robust to high noise rates (characteristic of noise-robust MAE).

In Figure 4 (c-d), we observe three qualitative improvements of GJS over JS: 1) no signs of overfitting to noise for large noise rates with low values of $\pi_1$, 2) better learning dynamics for large values of $\pi_1$ that otherwise learns slowly, and 3) converges to a higher validation accuracy.

**How many distributions to use?** Figure 5 depicts validation accuracy for varying number of distributions $M$. For all noise rates, we observe a performance increase going from $M = 2$ to $M = 3$. However, the performance of $M > 3$ depends on the noise rate. For lower noise rates, having more than three distributions can improve the performance. For higher noise rates *e.g.* 60%, having $M > 3$ degrades the performance. We hypothesise this is due to: 1) at high noise rates, there are only a few correctly labeled examples that can help guide the learning, and 2) going from $M = 2$ to $M = 3$ adds a consistency term, while $M > 3$ increases the importance of the consistency term in Proposition 2. Therefore, for a large enough M, the loss will find it easier to keep the consistency term low (keep predictions close to uniform as at the initialization), instead of generalizing based on the few clean examples. For simplicity, we have used $M = 3$ for all experiments with GJS.

**Is the improvements of GJS over JS due to mean prediction or consistency?** Proposition 2 decomposed GJS into a JS term with a mean prediction ($\bar{\boldsymbol{p}}_{>1}$) and a consistency term operating on all distributions but the target. In Table 3, we compare the performance of JS and GJS to GJS without the consistency term, i.e., $\mathcal{L}_{\mathrm{JS}_{\boldsymbol{\pi}'}}(e^{(y)}, \bar{\boldsymbol{p}}_{>1})$. The results suggest that the improvement of GJS over JS can be attributed to the consistency term.

Figure 4 (a-b) showed that JS improves the learning dynamics of MAE by blending it with CE, controlled by $\pi_1$. Similarly, we see here that the consistency term also improves the learning dynamics (underfitting and convergence speed) of MAE. Interestingly, Figure 4 (c-d), shows the higher values of $\pi_1$ (closer to MAE) work best for GJS, hinting that, the consistency term improves the learning dynamics of MAE so much so that the role of CE becomes less important.

Table 3: **Effect of Consistency.** Validation accuracy for JS, GJS w/o the consistency term in Proposition 2, and GJS for $40\%$ noise on the CIFAR-100 dataset. Using the mean of two predictions in the JS loss does not improve performance. On the other hand, adding the consistency term significantly helps.

| Method | Accuracy |
|---|---|
| $\mathcal{L}_{\mathrm{JS}}(\boldsymbol{e}^{(y)}, \boldsymbol{p}^{(2)})$ | 71.0 |
| $\mathcal{L}_{\mathrm{JS}_{\boldsymbol{\pi}'}}(\boldsymbol{e}^{(y)}, \bar{\boldsymbol{p}}_{>1})$ | 68.7 |
| $\mathcal{L}_{\mathrm{GJS}}(\boldsymbol{e}^{(y)}, \boldsymbol{p}^{(2)}, \boldsymbol{p}^{(3)})$ | 74.3 |

Table 4: **Effect of GJS.** Validation accuracy when using different loss functions for clean and noisy examples of the CIFAR-100 training set with 40% symmetric noise. Noisy examples benefit significantly more from GJS than clean examples (74.1 vs 72.9).

| Method | | $\pi_1$ | | |
|---|---|---|---|---|
| Clean | Noisy | 0.1 | 0.5 | 0.9 |
| JS | JS | 70.0 | **71.5** | 55.3 |
| GJS | JS | 72.6 | **72.9** | 70.2 |
| JS | GJS | 71.0 | **74.1** | 68.0 |
| GJS | GJS | 71.3 | **74.7** | 73.8 |

**Is GJS mostly helping the clean or noisy examples?** To better understand the improvements of GJS over JS, we perform an ablation with different losses for clean and noisy examples, see Table 4. We observe that using GJS instead of JS improves performance in all cases. Importantly, using GJS only for the noisy examples performs significantly better than only using it for the clean examples (74.1 vs 72.9). The best result is achieved when using GJS for both clean and noisy examples but still close to the noisy-only case (74.7 vs 74.1).

**How is different choices of perturbations affecting GJS?** In this work, we use stochastic augmentations for $\mathcal{A}$, see Appendix A.1 for details. Table 5 reports validation results on 40% symmetric and asymmetric noise on CIFAR-100 for varying types of augmentation. We observe that all methods improve their performance with stronger augmentation and that GJS achieves the best results in all cases. Also, note that we use weak augmentation for all naturally-noisy datasets (WebVision, ANIMAL-10N, and Food-101N) and still get state-of-the-art results.

**How fast is the convergence?** We found that some baselines (especially the robust NCE+RCE) had slow convergence. Therefore, we used 400 epochs for all methods to make sure all had time to converge properly. Table 6 shows results on 40% symmetric and asymmetric noise on CIFAR-100 when the number of epochs has been reduced by half.

**Is training with the proposed losses leading to more consistent networks?** Our motivation for investigating losses based on Jensen-Shannon divergence was partly due to the observation in Figure 1 that consistency and accuracy correlate when learning with CE loss. In Figure 6, we compare CE, JS, and GJS losses in terms of validation accuracy and consistency during training on CIFAR-100 with 40% symmetric noise. We find that the networks trained with JS and GJS losses are more consistent and has higher accuracy. In Appendix B.7, we report the consistency of the networks in Table 1.

**Summary of experiments in the appendix.** Due to space limitations, we report several important experiments in the appendix. We evaluate the effectiveness of GJS on 1) instance-dependent synthetic noise (Section B.1), and 2) real-world noisy datasets ANIMAL-10N and Food-101N (Section B.2). We also investigate the importance of 1) losses being symmetric and bounded for learning with noisy labels (Section B.3), and 2) a clean vs noisy validation set for hyperparameter selection and the effect of a single set of parameters for all noise rates (Section B.5).

Table 5: **Effect of Augmentation Strategy.** Validation accuracy for training w/o CutOut(-CO) or w/o RandAug(-RA) or w/o both(weak) on 40% symmetric and asymmetric noise on CIFAR-100. All methods improves by stronger augmentations. GJS performs best for all types of augmentations.

| Method | Symmetric | | | | Asymmetric | | | |
|---|---|---|---|---|---|---|---|---|
| | Full | -CO | -RA | Weak | Full | -CO | -RA | Weak |
| GCE | 70.8 | 64.2 | 64.1 | 58.0 | 51.7 | 44.9 | 46.6 | 42.9 |
| NCE+RCE | 68.5 | 66.6 | 68.3 | 61.7 | 57.5 | 52.1 | 49.5 | 44.4 |
| GJS | **74.8** | **71.3** | **70.6** | **66.5** | **62.6** | **56.8** | **52.2** | **44.9** |

Table 6: **Effect of Number of Epochs.** Validation accuracy for training with 200 and 400 epochs for 40% symmetric and asymmetric noise on CIFAR-100. GJS still outperforms the baselines and NCE+RCE's performance is reduced heavily by the decrease in epochs.

| Method | Symmetric | | Asymmetric | |
|---|---|---|---|---|
| | 200 | 400 | 200 | 400 |
| GCE | 70.3 | 70.8 | 39.1 | 51.7 |
| NCE+RCE | 60.0 | 68.5 | 35.0 | 57.5 |
| GJS | **72.9** | **74.8** | **43.2** | **62.6** |

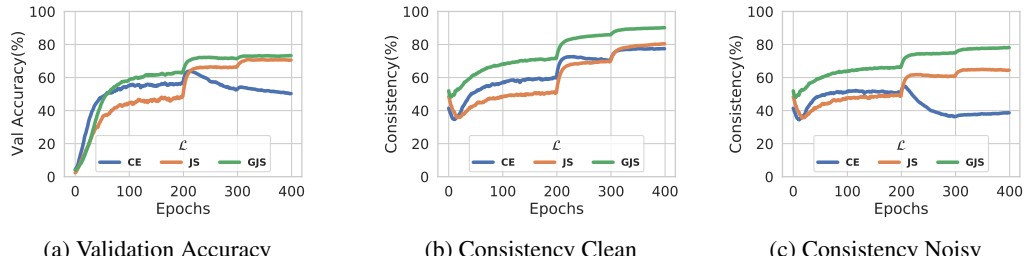

| (a) Validation Accuracy | (b) Consistency Clean | (c) Consistency Noisy |

Figure 6: **Evolution of a trained network's consistency for the CE, JS, and GJS losses.** We plot the evolution of the validation accuracy (a) and network's consistency on clean (b) and noisy (c) examples of the training set of CIFAR-100 when learning with 40% symmetric noise. All losses use the same learning rate and weight decay and both JS and GJS use $\pi_1 = 0.5$. The consistency of the learnt function and the accuracy closely correlate. The accuracy and consistency of JS and GJS improve during training, while both degrade when learning with CE loss.

## 5 Limitations & Future Directions

We empirically showed that the consistency of the network around noisy data degrades as it fits noise and accordingly proposed a loss based on generalized Jensen-Shannon divergence (GJS). While we empirically verified the significant role of consistency regularization in robustness to noise, we only theoretically showed the robustness ($B_L = B_U$) of GJS at its limit ($\pi_1 \to 1$) where the consistency term gradually vanishes. Therefore, the main limitation is the lack of a theoretical proof of the robustness of the consistency term in Proposition 2. This is, in general, an important but understudied area, also for the literature of self- or semi-supervised learning and thus is of utmost importance for future works.

Secondly, we had an important observation that GJS with $M > 3$ might not perform well under high noise rates. While we have some initial conjectures, this phenomenon deserves a systematic analysis both empirically and theoretically.

Finally, a minor practical limitation is the added computations for GJS forward passes, however this applies to *training time* only and in all our experiments, we only use one extra prediction ($M = 3$).

## 6 Final Remarks

We first made two central observations that (i) robust loss functions have an underfitting issue and (ii) consistency of noise-fitting networks is significantly lower around noisy data points. Correspondingly, we proposed two loss functions, JS and GJS, based on Jensen-Shannon divergence that (i) interpolates between noise-robust MAE and fast-converging CE, and (ii) encourages consistency around training data points. This simple proposal led to state-of-the-art performance on both synthetic and real-world noise datasets even when compared to the more elaborate pipelines such as DivideMix or ELR+. Furthermore, we discussed their robustness within the theoretical construction of Ghosh *et al.* [2]. By drawing further connections to other seminal loss functions such as CE, MAE, GCE, and consistency regularization, we uncovered other desirable or informative properties. We further empirically studied different aspects of the losses that corroborate various theoretical properties.

Overall, we believe the paper provides informative theoretical and empirical evidence for the usefulness of two simple and novel JS divergence-based loss functions for learning under noisy data that achieve state-of-the-art results. At the same time, it opens interesting future directions.

**Ethical Considerations.** Considerable resources are needed to create labeled data sets due to the burden of manual labeling process. Thus, the creators of large annotated datasets are mostly limited to well-funded companies and academic institutions. In that sense, developing robust methods against label noise enables less affluent organizations or individuals to benefit from labeled datasets since imperfect or automatic labeling can be used instead. On the other hand, proliferation of such harvested datasets can increase privacy concerns arising from redistribution and malicious use.

**Acknowledgement.** This work was partially supported by the Wallenberg AI, Autonomous Systems and Software Program (WASP) funded by the Knut and Alice Wallenberg Foundation.

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
