# A Training Details

All proposed losses and baselines use the same training settings, which are described in detail here.

## A.1 CIFAR

**General training details.** For all the results on the CIFAR datasets, we use a PreActResNet-34 with a standard SGD optimizer with Nesterov momentum, and a batch size of 128. For the network, we use three stacks of five residual blocks with 32, 64, and 128 filters for the layers in these stacks, respectively. The learning rate is reduced by a factor of 10 at 50% and 75% of the total 400 epochs. For data augmentation, we use RandAugment [32] with $N = 1$ and $M = 3$ using random cropping (size 32 with 4 pixels as padding), random horizontal flipping, normalization and lastly Cutout [33] with length 16. We set random seeds for all methods to have the same network weight initialization, order of data for the data loader, train-validation split, and noisy labels in the training set. We use a clean validation set corresponding to 10% of the training data. A clean validation set is commonly provided with real-world noisy datasets [28, 34]. Any potential gain from using a clean instead of a noisy validation set is the same for all methods since all share the same setup.

**JS and GJS implementation.** We implement the Jensen-Shannon-based losses using the definitions based on KL divergence, see Equation 2. To make sure the gradients are propagated through the target argument, we do not use the KL divergence in PyTorch. Instead, we write our own based on the official implementation.

**Search for learning rate and weight decay.** We do a separate hyperparameter search for learning rate and weight decay on 40% noise using both asymmetric and symmetric noises on CIFAR datasets. For CIFAR-10, we search for learning rates in $[0.001, 0.005, 0.01, 0.05, 0.1]$ and weight decays in $[1e-4, 5e-4, 1e-3]$. The method-specific hyperparameters used for this search were 0.9, 0.7, (0.1,1.0), 0.7, (1.0,1.0), 0.5, 0.5 for BS($\beta$), LS($\epsilon$), SCE($\alpha, \beta$), GCE($q$), NCE+RCE($\alpha, \beta$), JS($\pi_1$) and GJS($\pi_1$), respectively. For CIFAR-100, we search for learning rates in $[0.01, 0.05, 0.1, 0.2, 0.4]$ and weight decays in $[1e-5, 5e-5, 1e-4]$. The method-specific hyperparameters used for this search were 0.9, 0.7, (6.0,0.1), 0.7, (10.0,0.1), 0.5, 0.5 for BS($\beta$), LS($\epsilon$), SCE($\alpha, \beta$), GCE($q$), NCE+RCE($\alpha, \beta$), JS($\pi_1$) and GJS($\pi_1$), respectively. Note that, these fixed method-specific hyperparameters for both CIFAR-10 and CIFAR-100 are taken from their corresponding papers for this initial search of learning rate and weight decay but they will be further optimized systematically in the next steps.

**Search for method-specific parameters.** We fix the obtained best learning rate and weight decay for all other noise rates, but then for each noise rate/type, we search for method-specific parameters. For the methods with a single hyperparameter, BS ($\beta$), LS ($\epsilon$), GCE ($q$), JS ($\pi_1$), GJS ($\pi_1$), we try values in $[0.1, 0.3, 0.5, 0.7, 0.9]$. On the other hand, NCE+RCE and SCE have three hyperparameters, *i.e.* $\alpha$ and $\beta$ that scale the two loss terms, and $A := \log(0)$ for the RCE term. We set $A = \log(1e-4)$ and do a grid search for three values of $\alpha$ and two of beta $\beta$ (six in total) around the best reported parameters from each paper.[4]

**Test evaluation.** The best parameters are then used to train on the full training set with five different seeds. The final parameters that were used to get the results in Table 1 are shown in Table 7.

For completeness, in Appendix B.5, we provide results for a less thorough hyperparameter search(more similar to related work) which also use a noisy validation set.

## A.2 WebVision

All methods train a randomly initialized ResNet-50 model from PyTorch using the SGD optimizer with Nesterov momentum, and a batch size of 32 for GJS and 64 for CE and JS. For data augmentation, we do a random resize crop of size 224, random horizontal flips, and color jitter (torchvision ColorJitter transform with brightness=0.4, contrast=0.4, saturation=0.4, hue=0.2). We use a fixed weight decay of $1e-4$ and do a grid search for the best learning rate in $[0.1, 0.2, 0.4]$ and $\pi_1 \in [0.1, 0.3, 0.5, 0.7, 0.9]$. The learning rate is reduced by a multiplicative factor of 0.97 every epoch, and we train for a total of 300 epochs. The best starting learning rates were 0.4, 0.2, 0.1 for CE, JS and GJS, respectively.

---

[4]We also tried using $\beta = 1 - \alpha$, and mapping the best parameters from the papers to this range, combined with a similar search as for the single parameter methods, but this resulted in worse performance.

Table 7: **Hyperparameters for CIFAR.** A hyperparameter search over learning rates and weight decays, was done for 40% noise on both symmetric and asymmetric noise for the CIFAR datasets. The best parameters for each method are shown in this table, where the format is [learning rate, weight decay]. The hyperparameters for zero percent noise uses the same settings as for the symmetric noise. For the best learning rate and weight decay, another search is done for method-specific hyperparameters, and the best values are shown here. For methods with a single hyperparameter, the value correspond to their respective hyperparameter, *i.e.* BS ($\beta$), LS ($\epsilon$), GCE ($q$), JS ($\pi_1$), GJS ($\pi_1$). For NCE+RCE and SCE the value correspond to [$\alpha$, $\beta$].

| Dataset | Method | Learning Rate & Weight Decay | | Method-specific Hyperparameters | | | | | | |
| | | Sym Noise | Asym Noise | No Noise | Sym Noise | | | | Asym Noise | |
| | | 20-80% | 20-40% | 0% | 20% | 40% | 60% | 80% | 20% | 40% |
|---|---|---|---|---|---|---|---|---|---|---|
| CIFAR-10 | CE | [0.05, 1e-3] | [0.1, 1e-3] | - | - | - | - | - | - | - |
| | BS | [0.1, 1e-3] | [0.1, 1e-3] | 0.5 | 0.5 | 0.7 | 0.7 | 0.9 | 0.7 | 0.5 |
| | LS | [0.1, 5e-4] | [0.1, 1e-3] | 0.1 | 0.5 | 0.9 | 0.7 | 0.1 | 0.1 | 0.1 |
| | SCE | [0.01, 5e-4] | [0.05, 1e-3] | [0.2, 0.1] | [0.05, 0.1] | [0.1, 0.1] | [0.2, 1.0] | [0.1,1.0] | [0.1, 0.1] | [0.2, 1.0] |
| | GCE | [0.01, 5e-4] | [0.1, 1e-3] | 0.5 | 0.7 | 0.7 | 0.7 | 0.9 | 0.1 | 0.1 |
| | NCE+RCE | [0.005, 1e-3] | [0.05, 1e-4] | [10, 0.1] | [10, 0.1] | [10, 0.1] | [1.0, 0.1] | [10,1.0] | [10, 0.1] | [1.0, 0.1] |
| | JS | [0.01, 5e-4] | [0.1, 1e-3] | 0.1 | 0.7 | 0.7 | 0.9 | 0.9 | 0.3 | 0.3 |
| | GJS | [0.1, 5e-4] | [0.1, 1e-3] | 0.5 | 0.3 | 0.9 | 0.1 | 0.1 | 0.3 | 0.3 |
| CIFAR-100 | CE | [0.4, 1e-4] | [0.2, 1e-4] | - | - | - | - | - | - | - |
| | BS | [0.4, 1e-4] | [0.4, 1e-4] | 0.7 | 0.5 | 0.5 | 0.5 | 0.9 | 0.3 | 0.3 |
| | LS | [0.2, 5e-5] | [0.4, 1e-4] | 0.1 | 0.7 | 0.7 | 0.7 | 0.9 | 0.5 | 0.7 |
| | SCE | [0.2, 1e-4] | [0.4, 5e-5] | [0.1, 0.1] | [0.1, 0.1] | [0.1, 0.1] | [0.1, 1.0] | [0.1,0.1] | [0.1, 1.0] | [0.1, 1.0] |
| | GCE | [0.4, 1e-5] | [0.2, 1e-4] | 0.5 | 0.5 | 0.5 | 0.7 | 0.7 | 0.7 | 0.7 |
| | NCE+RCE | [0.2, 5e-5] | [0.2, 5e-5] | [20, 0.1] | [20, 0.1] | [20, 0.1] | [20, 0.1] | [20,0.1] | [20, 0.1] | [10, 0.1] |
| | JS | [0.2, 1e-4] | [0.1, 1e-4] | 0.1 | 0.1 | 0.3 | 0.5 | 0.3 | 0.5 | 0.5 |
| | GJS | [0.2, 5e-5] | [0.4, 1e-4] | 0.3 | 0.3 | 0.5 | 0.9 | 0.1 | 0.5 | 0.1 |

Both JS and GJS used $\pi_1 = 0.1$. With the best learning rate and $\pi_1$, we ran four more runs with new seeds for the network initialization and data loader.

# B   Additional Experiments and Insights

## B.1   Instance-Dependent Synthetic Noise

In Section 4.1, we showed results on two types of synthetic noise: symmetric ($\eta$) and asymmetric ($\eta(y)$). Although these noise types are simple to empirically and theoretically analyze, they might be different from noise observed in real-world datasets. Recently, a new type of synthetic noise has been proposed by Zhang *et al.* [25], where the risks of mislabeling an example of class $i$ to class $j$ vary per example ($\eta_{ij}(\boldsymbol{x})$). This type of noise is called instance-dependent and is more similar the noise in real-world datasets.

In Table 8, we compare CE, Generalized CE (GCE) and GJS on three different types of 35% instance-dependent noise on the CIFAR datasets. The training setup is the same as for the results in Table 1, described in detail in Section A.1. For all methods, we search for the best hyperparameters on the Type-I noise and use the same settings for the other two types. For CIFAR-10, the optimal hyperparameters (learning rate, weight decay, method-specific) were: (0.1, 1e-3, -), (0.005, 1e-3, 0.9), (0.001, 5e-4, 0.5) for CE, GCE, and GJS, respectively. For CIFAR-100, they were: (0.1, 5e-4, -), (0.4, 5e-5, 0.7), (0.1, 5e-4, 0.3) for CE, GCE, and GJS, respectively.

On the simpler CIFAR-10, GCE and GJS perform similarly, but on the more challenging CIFAR-100, GJS significantly outperform GCE.

## B.2   Real-World Noise: ANIMAL-10N & Food-101N

Here, we evaluate GJS on two naturally-noisy datasets: ANIMAL-10N [30] and Food-101N [31].

Table 8: **Instance-Dependent Synthetic Noise Benchmark on CIFAR.** We *reimplement* the Generalized CE (GCE) loss function into the *same learning setup* and a ResNet-34 network. We used *same hyperparameter optimization budget and mechanism* for all methods. We evaluate on 35% noise for the three types of instance-dependent synthetic noise proposed by Zhang *et al.* [25]. Mean test accuracy and standard deviation are reported from five runs and the statistically-significant top performers are boldfaced. As for the symmetric and asymmetric synthetic noise, the efficacy of GJS is more evident on the more challenging CIFAR-100 dataset, where GJS significantly outperforms the baselines.

| Dataset | Method | No Noise | Instance-Dependent Noise | | |
|---|---|---|---|---|---|
| | | 0% | Type-I | Type-II | Type-III |
| CIFAR-10 | CE | $94.35 \pm 0.10$ | $83.16 \pm 0.36$ | $81.18 \pm 0.38$ | $81.80 \pm 0.13$ |
| | GCE | $94.00 \pm 0.08$ | $\mathbf{86.50 \pm 0.16}$ | $\mathbf{83.80 \pm 0.26}$ | $\mathbf{84.85 \pm 0.12}$ |
| | GJS | $\mathbf{94.78 \pm 0.06}$ | $85.98 \pm 0.12$ | $83.81 \pm 0.12$ | $84.83 \pm 0.26$ |
| CIFAR-100 | CE | $77.60 \pm 0.17$ | $62.46 \pm 0.31$ | $63.51 \pm 0.41$ | $62.44 \pm 0.47$ |
| | GCE | $77.65 \pm 0.17$ | $65.62 \pm 0.32$ | $65.84 \pm 0.35$ | $65.85 \pm 0.32$ |
| | GJS | $\mathbf{79.27 \pm 0.29}$ | $\mathbf{68.49 \pm 0.14}$ | $\mathbf{69.21 \pm 0.16}$ | $\mathbf{69.04 \pm 0.16}$ |

**Food-101N.** The dataset contains 301k images classified as 101 different food recipes. The images were collected using Google, Bing, Yelp, and TripAdvisor. The noise rate is estimated to be 20%.

We follow the same training setup as the recent label correction method called Progressive Label Correction (PLC) [25], *i.e.* we use the same network architecture, augmentation strategy, optimizer, batch size, number of epochs, and learning rate scheduling. We use an initial learning rate and weight decay of 0.001, and $\pi_1 = 0.3$.

**ANIMAL-10N.** The dataset contains 55k images of 10 classes. The 10 classes can be grouped into 5 pairs of similar classes that are more likely to be confused: (cat, lynx), (jaguar, cheetah), (wolf, coyote), (chimpanzee, orangutan), (hamster, guinea pig). The images were collected using Google and Bing. The noise rate is estimated to be 8%.

We use the same training setup(network, optimizer, number of epochs, learning rate scheduling, etc) as PLC, but use cropping instead of random horizontal flipping as augmentation to reduce the risk of both augmentations being equal for GJS. We use an initial learning rate of 0.05, a weight decay of 5e-4, and $\pi_1 = 0.5$.

**Results.** The mean test accuracy and standard deviation from three runs for ANIMAL-10N and Food-101N are in Table 9 and 10, respectively. The results for all baselines are from Zhang *et al.* [25]. Our GJS loss outperforms all other methods on both datasets.

Table 9: **Real-world Noise: ANIMAL-10N.**

| Method | Accuracy |
|---|---|
| CE | $79.4 \pm 0.14$ |
| SELFIE | $81.8 \pm 0.09$ |
| PLC | $83.4 \pm 0.43$ |
| **GJS** | $\mathbf{84.2 \pm 0.07}$ |

Table 10: **Real-world Noise: Food-101N.**

| Method | Accuracy |
|---|---|
| CE | 81.67 |
| CleanNet | 83.95 |
| PLC | $85.28 \pm 0.04$ |
| **GJS** | $\mathbf{86.56 \pm 0.13}$ |

## B.3 Towards a better understanding of JS

In Proposition 2, we showed that JS is an important part of GJS, and therefore deserves attention. Here, we make a systematic *ablation study* to empirically examine the contribution of the difference(s) between JS loss and CE. We decompose the JS loss following the gradual construction of the Jensen-Shannon divergence in the work of Lin [8]. This construction, interestingly, lends significant empirical evidence to bounded losses' robustness to noise, in connection to Theorem 1 and 2 and Proposition 3.

Let $KL(\boldsymbol{p}, \boldsymbol{q})$ denote the KL-divergence of a predictive distribution $\boldsymbol{q} \in \Delta^{K-1}$ from a target distribution $\boldsymbol{p} \in \Delta^{K-1}$. $KL$ divergence is neither symmetric nor bounded. $K$ divergence, proposed by Lin *et al.* [8], is a bounded version defined as $K(\boldsymbol{p}, \boldsymbol{q}) := KL(\boldsymbol{p}, (\boldsymbol{p} + \boldsymbol{q})/2) = KL(\boldsymbol{p}, \boldsymbol{m})$. However, this divergence is not symmetric. A simple way to achieve symmetry is to take the average

Table 11: **Ablation Study of JS.** A comparison of JS and other KL-based divergences and their relationship to symmetry and boundedness. The distribution $\boldsymbol{m}$ is the mean of $\boldsymbol{p}$ and $\boldsymbol{q}$.

| Method | Formula | Symmetric | Bounded |
|--------|---------|-----------|---------|
| KL | $KL(\boldsymbol{p}, \boldsymbol{q})$ | | |
| KL' | $KL(\boldsymbol{q}, \boldsymbol{p})$ | | |
| Jeffrey's | $(KL(\boldsymbol{p}, \boldsymbol{q}) + KL(\boldsymbol{q}, \boldsymbol{p}))/2$ | ✓ | |
| K | $KL(\boldsymbol{p}, \boldsymbol{m})$ | | ✓ |
| K' | $KL(\boldsymbol{q}, \boldsymbol{m})$ | | ✓ |
| JS | $(KL(\boldsymbol{p}, \boldsymbol{m}) + KL(\boldsymbol{q}, \boldsymbol{m}))/2$ | ✓ | ✓ |

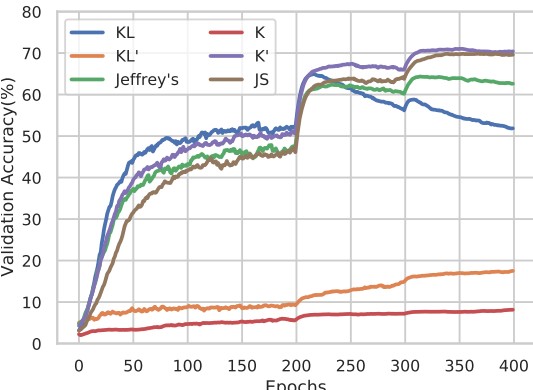

Figure 7: **Ablation Study of JS.** Validation accuracy of the divergences in Table 11 are plotted during training with $40\%$ symmetric noise on the CIFAR-100 dataset. Notably, the only two losses that show signs of overfitting ($KL$ and Jeffrey's) are unbounded. Interestingly, $K$ (bounded $KL$) makes the learning slower, while $K'$ (bounded $KL'$) considerably improves the learning dynamics. Finally, it can be seen that, JS, in contrast to its unbounded version (Jeffrey's), does not overfit to noise.

of forward and reverse versions of a divergence. For $KL$ and $K$, this gives rise to Jeffrey's divergence and JS with $\boldsymbol{\pi} = [\frac{1}{2}, \frac{1}{2}]^T$, respectively. Table 11 provides an overview of these divergences and Figure 7 shows their validation accuracy during training on CIFAR-100 with 40% symmetric noise.

**Bounded.** Notably, the only two losses that show signs of overfitting ($KL$ and Jeffrey's) are unbounded. Interestingly, $K$ (bounded $KL$) makes the learning much slower, while $K'$ (bounded $KL'$) considerably improves the learning dynamics. Finally, it can be seen that, JS, in contrast to its unbounded version (Jeffrey's), does not overfit to noise.

**Symmetry.** The Jeffrey's divergence performs better than either of its two constituent $KL$ terms. This is not as clear for JS, where $K'$ is performing surprisingly well on its own. In the proof of Proposition 1, we show that $K' \to$ MAE as $\pi_1 \to 1$, while $K$ goes to zero, which could explain why $K'$ seems to be robust to noise. Furthermore, $K'$, which is a component of JS, is reminiscent of label smoothing.

Beside the bound and symmetry, other notable properties of JS and GJS are the connections to MAE and consistency losses. Next section investigates the effect of hyperparameters that substantiates the connection to MAE (Proposition 1).

### B.4 Comparison between JS and GCE

We were pleasantly surprised by the finding in Proposition 1 that JS generalizes CE and MAE, similarly to GCE. Here, we highlight differences between JS and GCE.

**Theoretical properties.** Our inspiration to study JS came from the *symmetric* loss function of SCE, and the *bounded* loss of GCE. JS has *both* properties and a rich history in the field of information theory. This is also one of the reasons we studied these properties in Section B.3. Finally, JS generalizes naturally to more than two distributions.

**Gradients.** The gradients of CE/KL, GCE, JS and MAE with respect to logit $z_i$ of prediction $\boldsymbol{p} = [p_1, p_2, \ldots, p_K]$, given a label $\boldsymbol{e}^{(y)}$, are of the form $-\frac{\partial p_y}{\partial z_i} g(p_y)$ with $g(p_y)$ being $\frac{1}{p_y}$, $\frac{1}{p_y^{1-q}}$, $(1 - \pi_1) \log \left( \frac{\pi_1}{(1-\pi_1)p_y} + 1 \right)/Z$, and 1, for each of these losses respectively. Note that, $q$ is the hyperparameter of GCE and $p_y$ denotes the $y$th component of $\boldsymbol{p}$.

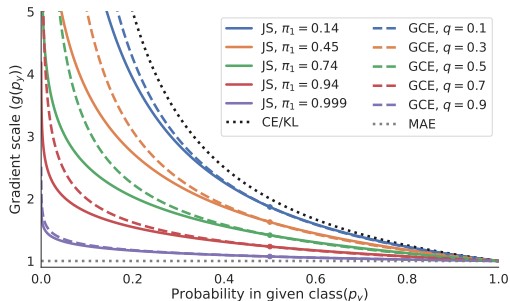

Figure 8: **Comparison between JS and GCE.** A comparison of gradients scales between JS and GCE. For each $q$ of GCE, a corresponding $\pi_1$ of JS is chosen such that the gradient scales are equal at $p_y = \frac{1}{2}$.

In Figure 8, these gradients are compared by varying the hyperparameter of GCE, $q \in [0.1, 0.3, 0.5, 0.7, 0.9]$, and finding the corresponding $\pi$ for $JS_\pi$ such that the two gradients are equal at $p_y = \frac{1}{2}$.

Looking at the behaviour of the different losses at low-$p_y$ regime, intuitively, a high gradient scale for low $p_y$ means a large parameter update for deviating from the given class. This can make noise free learning faster by pushing the probability to the correct class, which is what CE does. However, if the given class is incorrect (noisy) this can cause overfitting. The gradient scale of MAE induces same update magnitude for $p_y$, which can give the network more freedom to deviate from noisy classes, at the cost of slower learning for the correctly labeled examples.

Comparing GCE and $JS_\pi$ in Figure 8, it can be seen that $JS_\pi$ generally penalize lower probability in the given class less than what GCE does. In this sense, $JS_\pi$ behaves more like MAE.

For a derivation of the gradients of $D_{JS}$, see Section C.6.

**Label distributions.** GCE requires the label distribution to be onehot which makes it harder to incorporate GCE in many of the elaborate state-of-the-art methods that use "soft labels" *e.g.*, Mixup, co-training, or knowledge distillation.

## B.5   Noisy Validation Set & Single Set of Parameters

Our systematic procedure to search for hyperparameters (A.1) is done to have a more conclusive comparison to other methods. The most common procedure in related works is for each dataset, all methods use the same learning rate and weight decay(chosen seemingly arbitrary), and each method uses a single set of method-specific parameters for all noise rates and types. Baselines typically use the same method-specific parameters as reported in their respective papers. First, using the same learning rate and weight decay is problematic when comparing loss functions that have different gradient magnitudes. Second, directly using the parameters reported for the baselines is also problematic since the optimal hyperparameters depend on the training setup, which could be different, e.g., network architecture, augmentation, learning rate schedule, etc. Third, using a fixed method-specific parameter for all noise rates makes the results highly dependent on this choice. Lastly, it is not possible to know if other methods would have performed better if a proper hyperparameter search was done.

Here, for completeness, we use the same setup as in Section A.1, except we use the same learning rate and weight decay for all methods and search for hyperparameters based on a noisy validation set (more similar to related work).

The learning rate and weight decay for all methods are chosen based on noisy validation accuracy for CE on 40% symmetric noise for each dataset. The optimal learning rates and weight decays([lr,wd]) were [0.05, 1e-3] and [0.4, 1e-4] for CIFAR-10 and CIFAR-100, respectively. The method-specific parameters are found by a similar search as in Section A.1, except it is only done for 40% symmetric noise and the optimal parameters are used for all other noise rates and types. For CIFAR-10, the optimal method-specific hyperparameters were 0.5, 0.5, (0.1,0.1), 0.5, (10, 0.1), 0.5, 0.3 for BS($\beta$), LS($\epsilon$), SCE($\alpha, \beta$), GCE($q$), NCE+RCE($\alpha, \beta$), JS($\pi_1$) and GJS($\pi_1$), respectively. For CIFAR-100, the

Table 12: **Synthetic Noise Benchmark on CIFAR.** We *reimplement* other noise-robust loss functions into the *same learning setup* and ResNet-34, including label smoothing (LS), Bootstrap (BS), Symmetric CE (SCE), Generalized CE (GCE), and Normalized CE (NCE+RCE). We used *same hyperparameter optimization budget and mechanism* for all the prior works and ours. All methods use the same learning rate and weight decay and use the optimal method-specific parameters from a search on 40% symmetric noise based on noisy validation accuracy. Mean test accuracy and standard deviation are reported from five runs and the statistically-significant top performers are boldfaced.

| Dataset | Method | No Noise | Symmetric Noise Rate | | | | Asymmetric Noise Rate | |
|---|---|---|---|---|---|---|---|---|
| | | 0% | 20% | 40% | 60% | 80% | 20% | 40% |
| CIFAR-10 | CE | **95.66 ± 0.18** | 91.47 ± 0.28 | 87.31 ± 0.29 | 81.96 ± 0.38 | 65.28 ± 0.90 | 92.80 ± 0.64 | 85.82 ± 0.42 |
| | BS | 95.47 ± 0.11 | 93.65 ± 0.23 | 90.77 ± 0.30 | 49.80 ± 20.64 | 32.91 ± 5.43 | 93.86 ± 0.14 | 85.37 ± 1.07 |
| | LS | 95.45 ± 0.15 | 93.52 ± 0.09 | 89.94 ± 0.17 | 84.13 ± 0.80 | 62.76 ± 2.00 | 92.71 ± 0.41 | 83.61 ± 1.21 |
| | SCE | 94.92 ± 0.18 | 93.41 ± 0.20 | 90.99 ± 0.20 | 86.04 ± 0.31 | 41.04 ± 4.56 | 93.26 ± 0.13 | 84.46 ± 1.22 |
| | GCE | 94.94 ± 0.09 | 93.79 ± 0.19 | 91.45 ± 0.17 | 86.00 ± 0.20 | 62.01 ± 2.54 | 93.23 ± 0.12 | 85.92 ± 0.61 |
| | NCE+RCE | 94.31 ± 0.16 | 92.79 ± 0.16 | 90.31 ± 0.23 | 84.80 ± 0.47 | 34.47 ± 14.66 | 92.99 ± 0.15 | 87.00 ± 1.05 |
| | JS | 94.74 ± 0.21 | 93.53 ± 0.23 | 91.57 ± 0.22 | 86.21 ± 0.48 | 65.87 ± 2.92 | 92.97 ± 0.26 | 86.42 ± 0.36 |
| | GJS | **95.86 ± 0.10** | **95.20 ± 0.11** | **94.13 ± 0.19** | **89.65 ± 0.26** | **76.74 ± 0.75** | **94.81 ± 0.10** | **90.29 ± 0.26** |
| CIFAR-100 | CE | 77.84 ± 0.17 | 65.74 ± 0.06 | 55.57 ± 0.55 | 44.60 ± 0.79 | 10.74 ± 5.11 | 66.61 ± 0.45 | 50.42 ± 0.44 |
| | BS | 77.63 ± 0.25 | 73.01 ± 0.28 | 68.35 ± 0.43 | 54.07 ± 1.16 | 2.43 ± 0.49 | 69.75 ± 0.35 | 50.61 ± 0.32 |
| | LS | 77.60 ± 0.28 | 74.22 ± 0.30 | 66.84 ± 0.28 | 54.09 ± 0.71 | 21.00 ± 2.14 | 73.30 ± 0.42 | **57.02 ± 0.57** |
| | SCE | 77.46 ± 0.39 | 73.26 ± 0.29 | 66.96 ± 0.27 | 54.09 ± 0.49 | 13.26 ± 2.31 | 71.22 ± 0.33 | 49.91 ± 0.28 |
| | GCE | 76.70 ± 0.39 | 74.14 ± 0.32 | 70.41 ± 0.40 | 62.14 ± 0.27 | 12.38 ± 3.74 | 69.40 ± 0.30 | 48.54 ± 0.30 |
| | NCE+RCE | 73.23 ± 0.34 | 70.19 ± 0.27 | 65.61 ± 0.87 | 50.33 ± 1.58 | 5.55 ± 1.67 | 69.47 ± 0.25 | 56.32 ± 0.33 |
| | JS | 77.20 ± 0.53 | 74.47 ± 0.25 | 70.12 ± 0.39 | 61.69 ± 0.63 | **27.77 ± 4.11** | 67.21 ± 0.37 | 49.39 ± 0.13 |
| | GJS | **78.76 ± 0.32** | **77.14 ± 0.45** | **74.69 ± 0.12** | **64.06 ± 0.52** | 12.95 ± 2.40 | **74.44 ± 0.49** | 52.34 ± 0.81 |

optimal method-specific hyperparameters were 0.5, 0.7, (0.1, 0.1), 0.5, (20, 0.1), 0.1, 0.5 for BS($\beta$), LS($\epsilon$), SCE($\alpha, \beta$), GCE($q$), NCE+RCE($\alpha, \beta$), JS($\pi_1$) and GJS($\pi_1$), respectively. The results with this setup can be seen in Table 12.

### B.6 Consistency Measure

In this section, we provide more details about the consistency measure used in Figure 1. To be independent of any particular loss function, we considered a measure similar to standard Top-1 accuracy. We measure the ratio of samples that predict the same class on both the original image and an augmented version of it

$$\frac{1}{N} \sum_{i=1}^{N} \mathbb{1}\left( \arg\max_y f(\boldsymbol{x}_i) = \arg\max_y f(\tilde{\boldsymbol{x}}_i) \right) \tag{4}$$

where the sum is over all the training examples, and $\mathbb{1}$ is the indicator function, the argmax is over the predicted probability of $K$ classes, and $\tilde{\boldsymbol{x}}_i \sim \mathcal{A}(\boldsymbol{x}_i)$ is an augmented version of $\boldsymbol{x}_i$. Notably, this measure does not depend on the labels.

In the experiment in Figure 1, the original images are only normalized, while the augmented images use the same augmentation strategy as the benchmark experiments, see Section A.1.

### B.7 Consistency of Trained Networks on CIFAR

In Table 13, we report the training consistency of the networks used for the main CIFAR results in Table 1. We use the same consistency measure (Section B.6) as was used in Figure 1 and Figure 6. When learning with noisy labels, the networks trained with GJS is significantly more consistent than all the other methods. This is directly in line with Proposition 2, that shows how $\mathcal{L}_{\text{GJS}}$ encourages consistency.

In Table 1, we noticed better performance for CE compared to reported results in related work, which we mainly attribute to our thorough hyperparameter search. In Table 13, we observe better consistency for CE than in Figure 1, which we believe is for the same reason. Compared to Figure 1, the networks trained with the CE loss in Table 13 use a higher learning rate and weight decay, both of which have a regularizing effect, which could help against overfitting to noise.

Table 13: **Consistency of Trained Networks on CIFAR.** The training consistency of the networks from Table 1. Mean train consistency and standard deviation are reported from five runs and the networks with significantly higher consistency are boldfaced. As observed in Figure 1, the consistency is reduced for all methods for increasing noise rates. When learning with noisy labels, the networks trained with GJS are the most consistent for all noise rates and datasets.

| Dataset | Method | No Noise | Symmetric Noise Rate | | | | Asymmetric Noise Rate | |
|---|---|---|---|---|---|---|---|---|
| | | 0% | 20% | 40% | 60% | 80% | 20% | 40% |
| CIFAR-10 | CE | $94.35 \pm 0.10$ | $88.17 \pm 0.19$ | $82.66 \pm 0.37$ | $75.75 \pm 0.29$ | $64.28 \pm 1.15$ | $89.28 \pm 0.20$ | $85.26 \pm 0.67$ |
| | BS | $91.18 \pm 0.22$ | $86.50 \pm 0.24$ | $82.90 \pm 0.31$ | $75.59 \pm 0.51$ | $\mathbf{70.68 \pm 24.17}$ | $89.27 \pm 0.12$ | $85.77 \pm 0.72$ |
| | LS | $94.22 \pm 0.12$ | $90.20 \pm 0.18$ | $84.42 \pm 0.06$ | $77.29 \pm 0.17$ | $62.16 \pm 2.07$ | $89.31 \pm 0.22$ | $85.76 \pm 0.49$ |
| | SCE | $94.65 \pm 0.18$ | $91.11 \pm 0.12$ | $88.98 \pm 0.14$ | $84.70 \pm 0.20$ | $75.73 \pm 0.20$ | $90.16 \pm 0.19$ | $83.69 \pm 0.36$ |
| | GCE | $94.00 \pm 0.08$ | $91.12 \pm 0.07$ | $89.00 \pm 0.15$ | $84.58 \pm 0.17$ | $75.86 \pm 0.41$ | $89.07 \pm 0.27$ | $84.88 \pm 0.51$ |
| | NCE+RCE | $92.99 \pm 0.16$ | $91.15 \pm 0.17$ | $88.00 \pm 0.15$ | $82.01 \pm 0.33$ | $73.24 \pm 0.69$ | $91.09 \pm 0.10$ | $85.27 \pm 0.37$ |
| | JS | $\mathbf{94.95 \pm 0.06}$ | $91.46 \pm 0.10$ | $89.31 \pm 0.09$ | $84.77 \pm 0.11$ | $70.57 \pm 0.68$ | $87.47 \pm 0.07$ | $84.26 \pm 0.21$ |
| | GJS | $94.78 \pm 0.06$ | $\mathbf{94.24 \pm 0.12}$ | $\mathbf{91.21 \pm 0.05}$ | $\mathbf{90.36 \pm 0.08}$ | $\mathbf{78.42 \pm 0.29}$ | $\mathbf{91.88 \pm 0.17}$ | $\mathbf{89.08 \pm 0.36}$ |
| CIFAR-100 | CE | $86.24 \pm 0.49$ | $71.33 \pm 0.27$ | $59.45 \pm 0.51$ | $46.67 \pm 0.71$ | $33.07 \pm 1.96$ | $78.26 \pm 0.10$ | $71.94 \pm 0.30$ |
| | BS | $86.04 \pm 0.32$ | $77.59 \pm 0.54$ | $70.70 \pm 0.50$ | $65.44 \pm 1.60$ | $33.78 \pm 1.77$ | $76.45 \pm 0.57$ | $72.54 \pm 0.74$ |
| | LS | $88.40 \pm 0.07$ | $80.83 \pm 0.11$ | $73.18 \pm 0.09$ | $59.11 \pm 0.10$ | $36.69 \pm 0.39$ | $78.78 \pm 0.49$ | $67.76 \pm 0.37$ |
| | SCE | $85.72 \pm 0.11$ | $79.60 \pm 0.20$ | $71.50 \pm 0.24$ | $61.63 \pm 0.80$ | $39.98 \pm 1.08$ | $75.40 \pm 0.70$ | $63.66 \pm 0.33$ |
| | GCE | $85.63 \pm 0.19$ | $82.22 \pm 0.14$ | $77.69 \pm 0.13$ | $68.00 \pm 0.25$ | $53.28 \pm 0.83$ | $76.32 \pm 0.21$ | $64.77 \pm 0.43$ |
| | NCE+RCE | $78.14 \pm 0.16$ | $75.04 \pm 0.19$ | $70.59 \pm 0.29$ | $63.60 \pm 0.41$ | $43.63 \pm 2.00$ | $74.07 \pm 0.31$ | $64.47 \pm 0.30$ |
| | JS | $85.99 \pm 0.24$ | $82.58 \pm 0.28$ | $75.92 \pm 0.38$ | $66.80 \pm 0.58$ | $48.09 \pm 1.14$ | $78.25 \pm 0.14$ | $66.94 \pm 0.46$ |
| | GJS | $\mathbf{89.54 \pm 0.10}$ | $\mathbf{87.73 \pm 0.13}$ | $\mathbf{85.67 \pm 0.15}$ | $\mathbf{79.09 \pm 0.19}$ | $\mathbf{59.74 \pm 0.70}$ | $\mathbf{84.52 \pm 0.13}$ | $\mathbf{74.98 \pm 0.25}$ |

# C  Proofs

## C.1  JS's Connection to CE and MAE

**Proposition 1.** *Let $p \in \Delta^{K-1}$, then*

$$\lim_{\pi_1 \to 0} \mathcal{L}_{\mathrm{JS}}(e^{(y)}, p) = H(e^{(y)}, p), \qquad \lim_{\pi_1 \to 1} \mathcal{L}_{\mathrm{JS}}(e^{(y)}, p) = \frac{1}{2}\|e^{(y)} - p\|_1$$

*where $H(e^{(y)}, p)$ is the cross entropy of $e^{(y)}$ relative to $p$.*

*Proof of Proposition 1.* We want to show

$$\lim_{\pi_1 \to 0} \mathcal{L}_{\mathrm{JS}}(e^{(y)}, p) = \lim_{\pi_1 \to 0} \frac{\mathrm{JS}_{\pi}(e^{(y)}, p)}{H(1 - \pi_1)} = H(e^{(y)}, p) \tag{5}$$

$$\lim_{\pi_1 \to 1} \mathcal{L}_{\mathrm{JS}}(e^{(y)}, p) = \lim_{\pi_1 \to 1} \frac{\mathrm{JS}_{\pi}(e^{(y)}, p)}{H(1 - \pi_1)} = \frac{1}{2}\|e^{(y)} - p\|_1 \tag{6}$$

More specifically, we have $\mathrm{JS}_{\pi}(e^{(y)}, p) = \pi_1 D_{\mathrm{KL}}(e^{(y)}\|m) + \pi_2 D_{\mathrm{KL}}(p\|m)$, where $m = \pi_1 e^{(y)} + \pi_2 p$, and

$$\lim_{\pi_1 \to 0} \frac{\pi_1 D_{\mathrm{KL}}(e^{(y)}\|m)}{H(1 - \pi_1)} = H(e^{(y)}, p) \tag{7}$$

$$\lim_{\pi_1 \to 1} \frac{\pi_2 D_{\mathrm{KL}}(p\|m)}{H(1 - \pi_1)} = \frac{1}{2}\|e^{(y)} - p\|_1 \tag{8}$$

First, the we prove Equations 7 and 8, then show that the other two limits are zero.

Proof of Equation 7.

$$\lim_{\pi_1 \to 0} \frac{\pi_1 D_{\mathrm{KL}}(e^{(y)}\|m)}{H(1 - \pi_1)} = \lim_{\pi_1 \to 0} \frac{-\pi_1 \log(m_y)}{-(1 - \pi_1)\log(1 - \pi_1)} \tag{9}$$

$$= \lim_{\pi_1 \to 0} \log(m_y) \frac{1}{1 - \pi_1} \frac{\pi_1}{\log(1 - \pi_1)} \tag{10}$$

$$= \lim_{\pi_1 \to 0} \log(m_y) \frac{1}{1 - \pi_1} \cdot -(1 - \pi_1) \tag{11}$$

$$= \log p_y \cdot 1 \cdot -1 = H(e^{(y)}, p^{(2)}) \tag{12}$$

where we used L'Hôpital's rule for $\lim_{\pi_1 \to 0} \frac{\pi_1}{\log{(1-\pi_1)}}$ which is indeterminate of the form $\frac{0}{0}$.

Proof of Equation 8. Before taking the limit, we first rewrite the equation

$$\frac{\pi_2 D_{\mathrm{KL}}(\boldsymbol{p}\|\boldsymbol{m})}{H(1-\pi_1)} = -\frac{1}{\log{(1-\pi_1)}} \sum_{k=1}^{K} p_k \log \frac{p_k}{m_k} \tag{13}$$

$$= -\frac{1}{\log{(1-\pi_1)}} \left[ p_y \log \frac{p_y}{m_y} + \sum_{k \neq y}^{K} p_k \log \frac{p_k}{(1-\pi_1)p_k} \right] \tag{14}$$

$$= -\frac{1}{\log{(1-\pi_1)}} \left[ p_y \log \frac{p_y}{m_y} - \log{(1-\pi_1)} \sum_{k \neq y}^{K} p_k \right] \tag{15}$$

$$= -\frac{1}{\log{(1-\pi_1)}} \left[ p_y \log \frac{p_y}{m_y} - \log{(1-\pi_1)}(1-p_y) \right] \tag{16}$$

$$= -p_y \log \frac{p_y}{m_y} \frac{1}{\log{(1-\pi_1)}} + 1 - p_y \tag{17}$$

Now, we take the limit

$$\lim_{\pi_1 \to 1} \frac{\pi_2 D_{\mathrm{KL}}(\boldsymbol{p}\|\boldsymbol{m})}{H(1-\pi_1)} = \lim_{\pi_1 \to 1} -p_y \log \frac{p_y}{m_y} \frac{1}{\log{(1-\pi_1)}} + 1 - p_y \tag{18}$$

$$= 0 \cdot 0 + 1 - p_y \tag{19}$$

$$= \frac{1}{2}(1 - p_y + 1 - p_y) \tag{20}$$

$$= \frac{1}{2}\left(1 - p_y + \sum_{k \neq y}^{K} p_k\right) \tag{21}$$

$$= \frac{1}{2} \sum_{k=1}^{K} \left| e_k^{(y)} - p_k \right| = \frac{1}{2}\|\boldsymbol{e}^{(y)} - \boldsymbol{p}\|_1 \tag{22}$$

What is left to show is that the last two terms goes to zero in their respective limits.

$$\lim_{\pi_1 \to 1} \frac{\pi_1 D_{\mathrm{KL}}(\boldsymbol{e}^{(y)}\|\boldsymbol{m})}{H(1-\pi_1)} = \lim_{\pi_1 \to 1} \frac{-\pi_1 \log{(m_y)}}{-(1-\pi_1)\log{(1-\pi_1)}} \tag{23}$$

$$= \lim_{\pi_1 \to 1} \frac{-\pi_1 \log{(\pi_1 + (1-\pi_1)p_y)}}{-(1-\pi_1)\log{(1-\pi_1)}} \tag{24}$$

$$= \lim_{\pi_1 \to 1} \frac{\pi_1}{\log{(1-\pi_1)}} \frac{\log{(\pi_1 + (1-\pi_1)p_y)}}{1-\pi_1} \tag{25}$$

$$= 0 \cdot (p_y - 1) = 0 \tag{26}$$

Finally, the last term. Starting from Equation 17, we get

$$\lim_{\pi_1 \to 0} \frac{\pi_2 D_{\mathrm{KL}}(\boldsymbol{p}^{(2)}\|\boldsymbol{m})}{H(1-\pi_1)} = \lim_{\pi_1 \to 0} -p_y \frac{\log \frac{p_y}{m_y}}{\log{(1-\pi_1)}} + 1 - p_y \tag{27}$$

$$= \lim_{\pi_1 \to 0} -p_y \left( -\frac{1-p_y}{\pi_1 + (1-\pi_1)p_y} \cdot -(1-\pi_1) \right) + 1 - p_y \tag{28}$$

$$= \lim_{\pi_1 \to 0} -p_y \left( \frac{(1-p_y)(1-\pi_1)}{\pi_1 + (1-\pi_1)p_y} \right) + 1 - p_y \tag{29}$$

$$= \lim_{\pi_1 \to 0} p_y \left( \frac{-1 + \pi_1 + (1-\pi_1)p_y}{\pi_1 + (1-\pi_1)p_y} \right) + 1 - p_y \tag{30}$$

$$= \lim_{\pi_1 \to 0} p_y \left( \frac{-1}{\pi_1 + (1-\pi_1)p_y} + 1 \right) + 1 - p_y \tag{31}$$

$$= -1 + p_y + 1 - p_y = 0 \tag{32}$$

where L'Hôpital's rule was used for $\lim_{\pi_1 \to 0} -p_y \frac{\log \frac{p_y}{m_y}}{\log(1-\pi_1)}$ which is indeterminate of the form $\frac{0}{0}$. $\qquad\square$

## C.2 GJS's Connection to Consistency Regularization

**Proposition 2.** *Let* $\boldsymbol{p}^{(2)}, \ldots, \boldsymbol{p}^{(M)} \in \Delta^{K-1}$ *with* $M \geq 3$ *and* $\bar{\boldsymbol{p}}_{>1} = \frac{\sum_{j=2}^{M} \pi_j \boldsymbol{p}^{(j)}}{1-\pi_1}$, *then*

$$\mathcal{L}_{\mathrm{GJS}}(\boldsymbol{e}^{(y)}, \boldsymbol{p}^{(2)}, \ldots, \boldsymbol{p}^{(M)}) = \mathcal{L}_{\mathrm{JS}_{\boldsymbol{\pi}'}}(\boldsymbol{e}^{(y)}, \bar{\boldsymbol{p}}_{>1}) + (1-\pi_1)\mathcal{L}_{\mathrm{GJS}_{\boldsymbol{\pi}''}}(\boldsymbol{p}^{(2)}, \ldots, \boldsymbol{p}^{(M)})$$

*where* $\boldsymbol{\pi}' = [\pi_1, 1-\pi_1]^T$ *and* $\boldsymbol{\pi}'' = \frac{[\pi_2, \ldots, \pi_M]^T}{(1-\pi_1)}$.

*Proof of Proposition 2.* The Generalized Jensen-Shannon divergence can be simplified as below

$$\mathrm{GJS}_{\boldsymbol{\pi}}(\boldsymbol{e}^{(y)}, \boldsymbol{p}^{(2)}, \ldots, \boldsymbol{p}^{(M)}) = H(\pi_1 \boldsymbol{e}^{(y)} + (1-\pi_1)\bar{\boldsymbol{p}}_{>1}) - \sum_{j=2}^{M} \pi_j H(\boldsymbol{p}^{(j)}) \tag{33}$$

$$= H(\pi_1 + (1-\pi_1)m_y) + \sum_{i \neq y}^{K} H((1-\pi_1)m_i) - \sum_{j=2}^{M} \pi_j H(\boldsymbol{p}^{(j)}) \tag{34}$$

$$= /H(\pi_2 p_i) = p_i H(\pi_2) + \pi_2 H(p_i)/ \tag{35}$$

$$= H(\pi_1 + (1-\pi_1)m_y) + \sum_{i \neq y}^{K} [m_i H(1-\pi_1) + (1-\pi_1)H(m_i)] - \sum_{j=2}^{M} \pi_j H(\boldsymbol{p}^{(j)}) \tag{36}$$

$$= H(\pi_1 + (1-\pi_1)m_y) + \sum_{i \neq y}^{K} [m_i H(1-\pi_1)] - (1-\pi_1)H(m_y) \tag{37}$$

$$+ (1-\pi_1)\left(H(\bar{\boldsymbol{p}}_{>1}) - \frac{1}{1-\pi_1}\sum_{j=2}^{M} \pi_j H(\boldsymbol{p}^{(j)})\right) \tag{38}$$

$$= H(\pi_1 + (1-\pi_1)m_y) + \sum_{i \neq y}^{K} [m_i H(1-\pi_1) + (1-\pi_1)(H(m_i) - H(m_i))] \tag{39}$$

$$- (1-\pi_1)H(m_y) + (1-\pi_1)D_{\mathrm{GJS}_{\boldsymbol{\pi}''}}(\boldsymbol{p}^{(2)}, \ldots, \boldsymbol{p}^{(M)}) \tag{40}$$

$$= / \text{ Equation 35}/ \tag{41}$$

$$= H(\pi_1 + (1-\pi_1)m_y) + \sum_{i \neq y}^{K} H((1-\pi_1)m_i) - (1-\pi_1)H(\bar{\boldsymbol{p}}_{>1}) \tag{42}$$

$$+ (1-\pi_1)D_{\mathrm{GJS}_{\boldsymbol{\pi}''}}(\boldsymbol{p}^{(2)}, \ldots, \boldsymbol{p}^{(M)}) \tag{43}$$

$$= H(\pi_1 \boldsymbol{e}^{(y)} + (1-\pi_1)\bar{\boldsymbol{p}}_{>1}) - (1-\pi_1)H(\bar{\boldsymbol{p}}_{>1}) + (1-\pi_1)D_{\mathrm{GJS}_{\boldsymbol{\pi}''}}(\boldsymbol{p}^{(2)}, \ldots, \boldsymbol{p}^{(M)}) \tag{44}$$

$$= D_{\mathrm{JS}_{\boldsymbol{\pi}'}}(\boldsymbol{e}^{(y)}, \bar{\boldsymbol{p}}_{>1}) + (1-\pi_1)D_{\mathrm{GJS}_{\boldsymbol{\pi}''}}(\boldsymbol{p}^{(2)}, \ldots, \boldsymbol{p}^{(M)}) \tag{45}$$

where $\boldsymbol{\pi}' = [\pi_1, 1-\pi_1]$ and $\boldsymbol{\pi}'' = [\pi_2, \ldots, \pi_M]/(1-\pi_1)$. $\qquad\square$

That is, when using onehot labels, the generalized Jensen-Shannon divergence is a combination of two terms, one term encourages the mean prediction to be similar to the label and another term that encourages consistency between the predictions. For $M = 2$, the consistency term is zero.

## C.3 Noise Robustness

The proofs of the theorems in this sections are generalizations of the proofs by Zhang *et al.* [3]. The original theorems are specific to their particular GCE loss and cannot directly be used for other loss functions. We generalize the theorems to be useful for any loss function satisfying certain conditions(bounded and conditions in Lemma 1). To be able to use the theorems for GJS, we also generalize them to work for more than a single predictive distribution. Here, we use $(\boldsymbol{x}, y)$ to denote

a sample from $\mathcal{D}$ and $(\boldsymbol{x}, \tilde{y})$ to denote a sample from $\mathcal{D}_\eta$. Let $\eta_{ij}$ denote the probability that a sample of class $i$ was changed to class $j$ due to noise.

### C.3.1 Symmetric Noise

**Theorem 1.** *Under symmetric noise with $\eta < \frac{K-1}{K}$, if $B_L \leq \sum_{i=1}^{K} \mathcal{L}(\boldsymbol{e}^{(i)}, \boldsymbol{x}, f) \leq B_U$, $\forall \boldsymbol{x}, f$ is satisfied for a loss $\mathcal{L}$, then*

$$0 \leq R_\mathcal{L}^\eta(f^*) - R_\mathcal{L}^\eta(f_\eta^*) \leq \eta \frac{B_U - B_L}{K - 1}, \quad and \quad -\frac{\eta(B_U - B_L)}{K - 1 - \eta K} \leq R_\mathcal{L}(f^*) - R_\mathcal{L}(f_\eta^*) \leq 0,$$

*Proof of Theorem 1.* For any function, $f$, mapping an input $\boldsymbol{x} \in \mathbb{X}$ to $\Delta^{K-1}$, we have

$$R_\mathcal{L}(f) = \mathbb{E}_\mathcal{D}[\mathcal{L}(\boldsymbol{e}^{(y)}, \boldsymbol{x}, f)] = \mathbb{E}_{\boldsymbol{x}, y}[\mathcal{L}(\boldsymbol{e}^{(y)}, \boldsymbol{x}, f)]$$

and for uniform noise with noise rate $\eta$, the probability of a class not changing label due to noise is $\eta_{ii} = 1 - \eta$, while the probability of changing from one class to any other is $\eta_{ij} = \frac{\eta}{K-1}$. Therefore,

$$\begin{aligned}
R_\mathcal{L}^\eta(f) &= \mathbb{E}_{\mathcal{D}_\eta}[\mathcal{L}(\boldsymbol{e}^{(\tilde{y})}, \boldsymbol{x}, f)] = \mathbb{E}_{\boldsymbol{x}, \tilde{y}}[\mathcal{L}(\boldsymbol{e}^{(\tilde{y})}, \boldsymbol{x}, f)] \\
&= \mathbb{E}_{\boldsymbol{x}} \mathbb{E}_{y|\boldsymbol{x}} \mathbb{E}_{\tilde{y}|y, \boldsymbol{x}}[\mathcal{L}(\boldsymbol{e}^{(\tilde{y})}, \boldsymbol{x}, f)] \\
&= \mathbb{E}_{\boldsymbol{x}} \mathbb{E}_{y|\boldsymbol{x}}[(1 - \eta)\mathcal{L}(\boldsymbol{e}^{(y)}, \boldsymbol{x}, f) + \frac{\eta}{K-1} \sum_{i \neq y}^{K} \mathcal{L}(\boldsymbol{e}^{(i)}, \boldsymbol{x}, f)] \\
&= \mathbb{E}_{\boldsymbol{x}} \mathbb{E}_{y|\boldsymbol{x}} \left[ (1 - \eta)\mathcal{L}(\boldsymbol{e}^{(y)}, \boldsymbol{x}, f) + \frac{\eta}{K-1} \Big( \sum_{i=1}^{K} \mathcal{L}(\boldsymbol{e}^{(i)}, \boldsymbol{x}, f) - \mathcal{L}(\boldsymbol{e}^{(y)}, \boldsymbol{x}, f) \Big) \right] \\
&= \Big( 1 - \eta - \frac{\eta}{K-1} \Big) R_\mathcal{L}(f) + \frac{\eta}{K-1} \mathbb{E}_{\boldsymbol{x}} \mathbb{E}_{y|\boldsymbol{x}} \left[ \sum_{i=1}^{K} \mathcal{L}(\boldsymbol{e}^{(i)}, \boldsymbol{x}, f) \right] \\
&= \Big( 1 - \frac{\eta K}{K-1} \Big) R_\mathcal{L}(f) + \frac{\eta}{K-1} \mathbb{E}_{\boldsymbol{x}} \mathbb{E}_{y|\boldsymbol{x}} \left[ \sum_{i=1}^{K} \mathcal{L}(\boldsymbol{e}^{(i)}, \boldsymbol{x}, f) \right]
\end{aligned}$$

Using the bounds $B_L \leq \sum_{k=1}^{K} \mathcal{L}(\boldsymbol{e}^{(k)}, \boldsymbol{x}, f) \leq B_U$, we get:

$$\Big( 1 - \frac{\eta K}{K-1} \Big) R_\mathcal{L}(f) + \frac{\eta B_L}{K-1} \leq R_\mathcal{L}^\eta(f) \leq \Big( 1 - \frac{\eta K}{K-1} \Big) R_\mathcal{L}(f) + \frac{\eta B_U}{K-1}$$

With these bounds, the difference between $R_\mathcal{L}^\eta(f^*)$ and $R_\mathcal{L}^\eta(f_\eta^*)$ can be bounded as follows

$$R_\mathcal{L}^\eta(f^*) - R_\mathcal{L}^\eta(f_\eta^*) \leq \Big( 1 - \frac{\eta K}{K-1} \Big) R_\mathcal{L}(f^*) + \frac{\eta B_U}{K-1} - \Big( \Big( 1 - \frac{\eta K}{K-1} \Big) R_\mathcal{L}(f_\eta^*) + \frac{\eta B_L}{K-1} \Big) =$$

$$= \Big( 1 - \frac{\eta K}{K-1} \Big) (R_\mathcal{L}(f^*) - R_\mathcal{L}(f_\eta^*)) + \frac{\eta(B_U - B_L)}{K-1} \leq \frac{\eta(B_U - B_L)}{K-1}$$

where the last inequality follows from the assumption on the noise rate, $(1 - \frac{\eta K}{K-1}) > 0$, and that $f^*$ is the minimizer of $R_\mathcal{L}(f)$ so $R_\mathcal{L}(f^*) - R_\mathcal{L}(f_\eta^*) \leq 0$. Similarly, since $f_\eta^*$ is the minimizer of $R_\mathcal{L}^\eta(f)$, we have $R_\mathcal{L}^\eta(f^*) - R_\mathcal{L}^\eta(f_\eta^*) \geq 0$, which is the lower bound. $\qquad \square$

### C.3.2 Asymmetric Noise

**Lemma 1.** *Consider the following conditions for a loss with label $\boldsymbol{e}^{(i)}$, for any $i \in \{1, 2, \ldots, K\}$ and M-1 distributions $\boldsymbol{p}^{(2)}, \ldots, \boldsymbol{p}^{(M)} \in \Delta^{K-1}$:*

$$i)\ \mathcal{L}(\boldsymbol{e}^{(i)}, \boldsymbol{p}^{(2)}, \ldots, \boldsymbol{p}^{(M)}) = 0 \iff \boldsymbol{p}^{(2)}, \ldots, \boldsymbol{p}^{(M)} = \boldsymbol{e}^{(i)},$$

$$ii)\ 0 \leq \mathcal{L}(\boldsymbol{e}^{(i)}, \boldsymbol{p}^{(2)}, \ldots, \boldsymbol{p}^{(M)}) \leq C_1,$$

$$iii)\ \mathcal{L}(\boldsymbol{e}^{(i)}, \boldsymbol{e}^{(j)}, \ldots, \boldsymbol{e}^{(j)}) = C_2 \leq C_1, \ with\ i \neq j.$$

*where $C_1, C_2$ are constants.*

**Theorem 2.** *Let $\mathcal{L}$ be any loss function satisfying the conditions in Lemma 1. Under class dependent noise, when the probability of the noise not changing label is larger than changing it to any other class($\eta_{yi} < \eta_{yy}$, for all $i \neq y$, with $y$ being the true label), and if $R^{\eta}_{\mathcal{L}}(f^*) = 0$, then*

$$0 \leq R^{\eta}_{\mathcal{L}}(f^*) - R^{\eta}_{\mathcal{L}}(f^*_{\eta}) \leq (B_U - B_L)\mathbb{E}_{\mathcal{D}}[\eta_{yy}] + (C_1 - C_2)\mathbb{E}_{\mathcal{D}}[\sum_{i \neq y}^{K}(\eta_{yy} - \eta_{yi})], \quad (46)$$

*where $B_L \leq \sum_{i=1}^{K} \mathcal{L}(e^{(i)}, x, f) \leq B_U$ for all $x$ and $f$, $f^*$ is the global minimizer of $R_{\mathcal{L}}(f)$, and $f^*_{\eta}$ is the global minimizer of $R^{\eta}_{\mathcal{L}}(f)$.*

*Proof of Theorem 2.* For class dependent noisy(asymmetric) and any function, $f$, mapping an input $x \in \mathbb{X}$ to $\Delta^{K-1}$, we have

$$R^{\eta}_{\mathcal{L}}(f) = \mathbb{E}_{\mathcal{D}}[\eta_{yy}\mathcal{L}(e^{(y)}, x, f)] + \mathbb{E}_{\mathcal{D}}[\sum_{i \neq y}^{K} \eta_{yi}\mathcal{L}(e^{(i)}, x, f)]$$

$$= \mathbb{E}_{\mathcal{D}}[\eta_{yy}\Big(\sum_{i=1}^{K} \mathcal{L}(e^{(i)}, x, f) - \sum_{i \neq y}^{K} \mathcal{L}(e^{(i)}, x, f)\Big)] + \mathbb{E}_{\mathcal{D}}[\sum_{i \neq y}^{K} \eta_{yi}\mathcal{L}(e^{(i)}, x, f)]$$

$$= \mathbb{E}_{\mathcal{D}}[\eta_{yy}\sum_{i=1}^{K} \mathcal{L}(e^{(i)}, x, f)] - \mathbb{E}_{\mathcal{D}}[\sum_{i \neq y}^{K}(\eta_{yy} - \eta_{yi})\mathcal{L}(e^{(i)}, x, f)]$$

By using the bounds $B_L, B_U$ we get

$$R^{\eta}_{\mathcal{L}}(f) \leq B_U \mathbb{E}_{\mathcal{D}}[\eta_{yy}] - \mathbb{E}_{\mathcal{D}}[\sum_{i \neq y}^{K}(\eta_{yy} - \eta_{yi})\mathcal{L}(e^{(i)}, x, f)]$$

$$R^{\eta}_{\mathcal{L}}(f) \geq B_L \mathbb{E}_{\mathcal{D}}[\eta_{yy}] - \mathbb{E}_{\mathcal{D}}[\sum_{i \neq y}^{K}(\eta_{yy} - \eta_{yi})\mathcal{L}(e^{(i)}, x, f)]$$

Hence,

$$R^{\eta}_{\mathcal{L}}(f^*) - R^{\eta}_{\mathcal{L}}(f^*_{\eta}) \leq (B_U - B_L)\mathbb{E}_{\mathcal{D}}[\eta_{yy}] + \quad (47)$$

$$+ \mathbb{E}_{\mathcal{D}}[\sum_{i \neq y}^{K}(\eta_{yy} - \eta_{yi})\big(\mathcal{L}(e^{(i)}, x, f^*_{\eta}) - \mathcal{L}(e^{(i)}, x, f^*)\big)]$$

From the assumption that $R_{\mathcal{L}}(f^*) = 0$, we have $\mathcal{L}(e^{(y)}, x, f^*) = 0$. Using the conditions on the loss function from Lemma 1, for all $i \neq y$, we get

$$\mathcal{L}(e^{(i)}, x, f^*_{\eta}) - \mathcal{L}(e^{(i)}, x, f^*) = /\ \mathcal{L}(e^{(y)}, x, f^*) = 0 \text{ and } i) /$$
$$= \mathcal{L}(e^{(i)}, x, f^*_{\eta}) - \mathcal{L}(e^{(i)}, e^{(y)})$$
$$= /\ iii) /$$
$$= \mathcal{L}(e^{(i)}, x, f^*_{\eta}) - C_2$$
$$= /\ ii) /$$
$$\leq C_1 - C_2$$

By our assumption on the noise rates, we have $\eta_{yy} - \eta_{yi} > 0$. We have

$$R^{\eta}_{\mathcal{L}}(f^*) - R^{\eta}_{\mathcal{L}}(f^*_{\eta}) \leq (B_U - B_L)\mathbb{E}_{\mathcal{D}}[\eta_{yy}] + (C_1 - C_2)\mathbb{E}_{\mathcal{D}}[\sum_{i \neq y}^{K}(\eta_{yy} - \eta_{yi})]$$

Since $f^*_{\eta}$ is the global minimizer of $R^{\eta}_{\mathcal{L}}(f)$ we have $R^{\eta}_{\mathcal{L}}(f^*) - R^{\eta}_{\mathcal{L}}(f^*_{\eta}) \geq 0$, which is the lower bound. $\qquad\square$

**Remark 2.** *The generalized Jensen-Shannon Divergence satisfies the conditions in Lemma 1, with*

$$C_1 = H(\boldsymbol{\pi}), \quad C_2 = H(\pi_1) + H(1 - \pi_1).$$

*Proof of Remark 2.* i). Follows directly from Jensen's inequality for the Shannon entropy. ii). The lower bound follows directly from Jensen's inequality for the non-negative Shannon entropy. The upper bound is shown below

$$
\begin{aligned}
D_{\mathrm{GJS}_{\boldsymbol{\pi}}}(\boldsymbol{p}^{(1)}, \boldsymbol{p}^{(2)}, \dots, \boldsymbol{p}^{(M)}) &= \sum_{j=1}^{M} \pi_j D_{\mathrm{KL}}(\boldsymbol{p}^{(j)} \| \boldsymbol{m}) \\
&= \sum_{j=1}^{M} \left[ \pi_j \sum_{l=1}^{K} p_l^{(j)} \log \left( \frac{p_l^{(j)}}{m_l} \right) \right] \\
&= \sum_{j=1}^{M} \left[ \pi_j \sum_{l=1}^{K} p_l^{(j)} \left( \log \left( \frac{\pi_j p_l^{(j)}}{m_l} \right) + \log \frac{1}{\pi_j} \right) \right] \\
&= \sum_{j=1}^{M} \left[ \pi_j \sum_{l=1}^{K} \left[ p_l^{(j)} \log \left( \frac{\pi_j p_l^{(j)}}{m_l} \right) - p_l^{(j)} \log \pi_j \right] \right] \\
&= \sum_{j=1}^{M} \left[ -\pi_j \log \pi_j + \pi_j \sum_{l=1}^{K} p_l^{(j)} \log \left( \frac{\pi_j p_l^{(j)}}{m_l} \right) \right] \\
&= \sum_{j=1}^{M} \left[ H(\pi_j) + \pi_j \sum_{l=1}^{K} p_l^{(j)} \log \left( \frac{\pi_j p_l^{(j)}}{m_l} \right) \right] \\
&= \sum_{j=1}^{M} \left[ H(\pi_j) + \pi_j \sum_{l=1}^{K} p_l^{(j)} \log \left( \frac{p_l^{(j)}}{p_l^{(j)} + \frac{1}{\pi_j} \sum_{i \neq j}^{M} \pi_i p_l^{(i)}} \right) \right] \\
&\leq \sum_{j=1}^{M} H(\pi_j) = H(\boldsymbol{\pi})
\end{aligned}
$$

where the inequality holds with equality iff $\frac{1}{\pi_j} \sum_{i \neq j}^{M} \pi_i p_l^{(i)} = 0$ when $p_l^{(j)} > 0$ for all $j \in \{1, 2, \dots, M\}$ and $l \in \{1, 2, \dots, K\}$. Hence, GJS is bounded above by $H(\boldsymbol{\pi})$.

iii). Let the label be $\boldsymbol{e}^{(i)}$ and the other M-1 distributions be $\boldsymbol{e}^{(j)}$ with $i \neq j$ then

$$
D_{\mathrm{GJS}_{\boldsymbol{\pi}}} = H \left( \pi_1 \boldsymbol{e}^{(i)} + \sum_{l=2}^{M} \pi_l \boldsymbol{e}^{(j)} \right) - \pi_1 H(\boldsymbol{e}^{(i)}) - \sum_{l=2}^{M} \pi_l H(\boldsymbol{e}^{(j)}) = H(\pi_1 \boldsymbol{e}^{(i)} + (1 - \pi_1) \boldsymbol{e}^{(j)}) \tag{48}
$$

Notably, $C_1 = C_2$ for $M = 2$. $\qquad \square$

### C.3.3 Improving GJS Risk Difference Bounds

**Proposition 4.** $\mathcal{L}_{\mathrm{JS}}$ *and* $\mathcal{L}_{\mathrm{GJS}}$ *have the same risk bounds in Theorem 1 and 2 if* $\mathbb{E}_{\mathbf{x}}[\mathcal{L}_{\mathrm{GJS}_{\boldsymbol{\pi}''}}^{f^*}(\boldsymbol{p}^{(2)}, \dots, \boldsymbol{p}^{(M)})] \leq \mathbb{E}_{\mathbf{x}}[\mathcal{L}_{\mathrm{GJS}_{\boldsymbol{\pi}''}}^{f_{\tilde{\eta}}^*}(\boldsymbol{p}^{(2)}, \dots, \boldsymbol{p}^{(M)})]$, *where* $\mathcal{L}_{\mathrm{GJS}_{\boldsymbol{\pi}''}}^{f}(\boldsymbol{p}^{(2)}, \dots, \boldsymbol{p}^{(M)})$ *is the consistency term from Proposition 2.*

*Proof of Proposition 4.*
**Symmetric Noise** From the proof of Theorem 1, we have for any function, $f$, mapping an input $\boldsymbol{x} \in \mathbb{X}$ to $\Delta^{K-1}$

$$
R_{\mathcal{L}}^{\eta}(f) = \left( 1 - \frac{\eta K}{K - 1} \right) R_{\mathcal{L}}(f) + \frac{\eta}{K - 1} \mathbb{E}_{\boldsymbol{x}} \mathbb{E}_{y|\boldsymbol{x}} \left[ \sum_{i=1}^{K} \mathcal{L}(\boldsymbol{e}^{(i)}, \boldsymbol{x}, f) \right]
$$

Using Proposition 2 for GJS, we get

$$R^\eta_{\mathcal{L}_{\mathrm{GJS}}}(f) = \left(1 - \frac{\eta K}{K-1}\right) R_{\mathcal{L}_{\mathrm{GJS}}}(f) + \frac{\eta}{K-1} \mathbb{E}_{\boldsymbol{x}} \mathbb{E}_{y|\boldsymbol{x}} \left[ \sum_{i=1}^{K} \mathcal{L}^f_{\mathrm{JS}_{\boldsymbol{\pi}'}}(\boldsymbol{e}^{(i)}, \bar{\boldsymbol{p}}_{>1}) \right]$$
$$+ (1 - \pi_1) \frac{\eta K}{K-1} \mathbb{E}_{\boldsymbol{x}} \mathbb{E}_{y|\boldsymbol{x}} \left[ \mathcal{L}^f_{\mathrm{GJS}_{\boldsymbol{\pi}''}}(\boldsymbol{p}^{(2)}, \ldots, \boldsymbol{p}^{(M)}) \right]$$

Let $B^{\mathrm{JS}}_L$, $B^{\mathrm{JS}}_U$ be the lower and upper bound for JS (M=2) in Proposition 5. These bounds 5 holds for any $\boldsymbol{p}^{(2)} \in \Delta^{K-1}$ and therefore also holds for $\bar{\boldsymbol{p}}_{>1}$. Hence, we have

$$R^\eta_{\mathcal{L}_{\mathrm{GJS}}}(f) \geq \left(1 - \frac{\eta K}{K-1}\right) R_{\mathcal{L}_{\mathrm{GJS}}}(f) + \frac{\eta B^{\mathrm{JS}}_L}{K-1} + (1 - \pi_1) \frac{\eta K}{K-1} \mathbb{E}_{\boldsymbol{x}} \mathbb{E}_{y|\boldsymbol{x}} \left[ \mathcal{L}^f_{\mathrm{GJS}_{\boldsymbol{\pi}''}}(\boldsymbol{p}^{(2)}, \ldots, \boldsymbol{p}^{(M)}) \right]$$

$$R^\eta_{\mathcal{L}_{\mathrm{GJS}}}(f) \leq \left(1 - \frac{\eta K}{K-1}\right) R_{\mathcal{L}_{\mathrm{GJS}}}(f) + \frac{\eta B^{\mathrm{JS}}_U}{K-1} + (1 - \pi_1) \frac{\eta K}{K-1} \mathbb{E}_{\boldsymbol{x}} \mathbb{E}_{y|\boldsymbol{x}} \left[ \mathcal{L}^f_{\mathrm{GJS}_{\boldsymbol{\pi}''}}(\boldsymbol{p}^{(2)}, \ldots, \boldsymbol{p}^{(M)}) \right]$$

With these bounds, the difference between $R^\eta_{\mathcal{L}}(f^*)$ and $R^\eta_{\mathcal{L}}(f^*_\eta)$ can be bounded as follows

$$R^\eta_{\mathcal{L}_{\mathrm{GJS}}}(f^*) - R^\eta_{\mathcal{L}_{\mathrm{GJS}}}(f^*_\eta) \leq \left(1 - \frac{\eta K}{K-1}\right)(R_{\mathcal{L}_{\mathrm{GJS}}}(f^*) - R_{\mathcal{L}_{\mathrm{GJS}}}(f^*_\eta)) + \frac{\eta(B^{\mathrm{JS}}_U - B^{\mathrm{JS}}_L)}{K-1}$$
$$+ \frac{(1 - \pi_1)\eta K}{K-1} \mathbb{E}_{\boldsymbol{x}} \mathbb{E}_{y|\boldsymbol{x}} \left[ \mathcal{L}^{f^*}_{\mathrm{GJS}_{\boldsymbol{\pi}''}}(\boldsymbol{p}^{(2)}, \ldots, \boldsymbol{p}^{(M)}) - \mathcal{L}^{f^*_\eta}_{\mathrm{GJS}_{\boldsymbol{\pi}''}}(\boldsymbol{p}^{(2)}, \ldots, \boldsymbol{p}^{(M)}) \right]$$
$$\leq \frac{\eta(B^{\mathrm{JS}}_U - B^{\mathrm{JS}}_L)}{K-1}$$

where the last inequality follows from the assumption on the noise rate, $(1 - \frac{\eta K}{K-1}) > 0$, that $f^*$ is the minimizer of $R_{\mathcal{L}}(f)$ so $R_{\mathcal{L}}(f^*) - R_{\mathcal{L}}(f^*_\eta) \leq 0$, and the assumption on the consistency of $f^*$ and $f^*_\eta$. Similarly, since $f^*_\eta$ is the minimizer of $R^\eta_{\mathcal{L}}(f)$, we have $R^\eta_{\mathcal{L}}(f^*) - R^\eta_{\mathcal{L}}(f^*_\eta) \geq 0$, which is the lower bound. Hence, we have shown that $\mathcal{L}_{\mathrm{JS}}$ and $\mathcal{L}_{\mathrm{GJS}}$ have the same bounds for the risk difference for symmetric noise.

**Asymmetric Noise** For class dependent noisy(asymmetric) and any function, $f$, mapping an input $\boldsymbol{x} \in \mathbb{X}$ to $\Delta^{K-1}$, we have

$$R^\eta_{\mathcal{L}_{\mathrm{GJS}}}(f) = \mathbb{E}_{\mathcal{D}}[\sum_{i=1}^{K} \eta_{yi} \mathcal{L}_{\mathrm{GJS}}(\boldsymbol{e}^{(i)}, \boldsymbol{x}, f)]$$
$$= \mathbb{E}_{\mathcal{D}}[\eta_{yy} \mathcal{L}^f_{\mathrm{JS}_{\boldsymbol{\pi}'}}(\boldsymbol{e}^{(y)}, \bar{\boldsymbol{p}}_{>1}) + \sum_{i \neq y}^{K} \eta_{yi} \mathcal{L}^f_{\mathrm{JS}_{\boldsymbol{\pi}'}}(\boldsymbol{e}^{(i)}, \bar{\boldsymbol{p}}_{>1})$$
$$+ (1 - \pi_1) \mathcal{L}^f_{\mathrm{GJS}_{\boldsymbol{\pi}''}}(\boldsymbol{p}^{(2)}, \ldots, \boldsymbol{p}^{(M)})]$$
$$= \mathbb{E}_{\mathcal{D}}[\eta_{yy} \left( \sum_{i=1}^{K} \mathcal{L}^f_{\mathrm{JS}_{\boldsymbol{\pi}'}}(\boldsymbol{e}^{(i)}, \bar{\boldsymbol{p}}_{>1}) - \sum_{i \neq y}^{K} \mathcal{L}^f_{\mathrm{JS}_{\boldsymbol{\pi}'}}(\boldsymbol{e}^{(i)}, \bar{\boldsymbol{p}}_{>1}) \right) + \sum_{i \neq y}^{K} \eta_{yi} \mathcal{L}^f_{\mathrm{JS}_{\boldsymbol{\pi}'}}(\boldsymbol{e}^{(i)}, \bar{\boldsymbol{p}}_{>1})$$
$$+ (1 - \pi_1) \mathcal{L}^f_{\mathrm{GJS}_{\boldsymbol{\pi}''}}(\boldsymbol{p}^{(2)}, \ldots, \boldsymbol{p}^{(M)})]$$
$$= \mathbb{E}_{\mathcal{D}}[\eta_{yy} \sum_{i=1}^{K} \mathcal{L}^f_{\mathrm{JS}_{\boldsymbol{\pi}'}}(\boldsymbol{e}^{(i)}, \bar{\boldsymbol{p}}_{>1}) - \sum_{i \neq y}^{K} (\eta_{yy} - \eta_{yi}) \mathcal{L}^f_{\mathrm{JS}_{\boldsymbol{\pi}'}}(\boldsymbol{e}^{(i)}, \bar{\boldsymbol{p}}_{>1})$$
$$+ (1 - \pi_1) \mathcal{L}^f_{\mathrm{GJS}_{\boldsymbol{\pi}''}}(\boldsymbol{p}^{(2)}, \ldots, \boldsymbol{p}^{(M)})]$$

where Proposition 2 was used to separate GJS into a JS and a consistency term. By using the bounds $B_L^{\text{JS}}, B_U^{\text{JS}}$ we get

$$R_{\mathcal{L}_{\text{GJS}}}^{\eta}(f) \leq \mathbb{E}_{\mathcal{D}}[\eta_{yy}B_U^{\text{JS}} - \sum_{i \neq y}^{K}(\eta_{yy} - \eta_{yi})\mathcal{L}_{\text{JS}_{\pi'}}^{f}(\boldsymbol{e}^{(i)}, \bar{\boldsymbol{p}}_{>1}) + (1 - \pi_1)\mathcal{L}_{\text{GJS}_{\pi''}}^{f}(\boldsymbol{p}^{(2)}, \ldots, \boldsymbol{p}^{(M)})]$$

$$R_{\mathcal{L}_{\text{GJS}}}^{\eta}(f) \geq \mathbb{E}_{\mathcal{D}}[\eta_{yy}B_L^{\text{JS}} - \sum_{i \neq y}^{K}(\eta_{yy} - \eta_{yi})\mathcal{L}_{\text{JS}_{\pi'}}^{f}(\boldsymbol{e}^{(i)}, \bar{\boldsymbol{p}}_{>1}) + (1 - \pi_1)\mathcal{L}_{\text{GJS}_{\pi''}}^{f}(\boldsymbol{p}^{(2)}, \ldots, \boldsymbol{p}^{(M)})]$$

Hence,

$$R_{\mathcal{L}}^{\eta}(f^*) - R_{\mathcal{L}}^{\eta}(f_\eta^*) \leq (B_U^{\text{JS}} - B_L^{\text{JS}})\mathbb{E}_{\mathcal{D}}[\eta_{yy}]$$

$$+ \mathbb{E}_{\mathcal{D}}[\sum_{i \neq y}^{K}(\eta_{yy} - \eta_{yi})\big(\mathcal{L}_{\text{JS}_{\pi'}}^{f_\eta^*}(\boldsymbol{e}^{(i)}, \bar{\boldsymbol{p}}_{>1}) - \mathcal{L}_{\text{JS}_{\pi'}}^{f^*}(\boldsymbol{e}^{(i)}, \bar{\boldsymbol{p}}_{>1})\big)] \tag{49}$$

$$+ (1 - \pi_1)\Big(\mathbb{E}_{\mathcal{D}}[\mathcal{L}_{\text{GJS}_{\pi''}}^{f^*}(\boldsymbol{p}^{(2)}, \ldots, \boldsymbol{p}^{(M)})] - \mathbb{E}_{\mathcal{D}}[\mathcal{L}_{\text{GJS}_{\pi''}}^{f_\eta^*}(\boldsymbol{p}^{(2)}, \ldots, \boldsymbol{p}^{(M)})]\Big) \tag{50}$$

From the assumption that $R_{\mathcal{L}_{\text{GJS}}}(f^*) = 0$, we have $\mathcal{L}_{\text{GJS}}(\boldsymbol{e}^{(y)}, \boldsymbol{x}, f^*) = 0$. Using the conditions on the loss function from Lemma 1, for all $i \neq y$, we get

$$\mathcal{L}_{\text{JS}_{\pi'}}^{f_\eta^*}(\boldsymbol{e}^{(i)}, \bar{\boldsymbol{p}}_{>1}) - \mathcal{L}_{\text{JS}_{\pi'}}^{f^*}(\boldsymbol{e}^{(i)}, \bar{\boldsymbol{p}}_{>1}) = /\ \mathcal{L}_{\text{GJS}}(\boldsymbol{e}^{(y)}, \boldsymbol{x}, f^*) = 0 \text{ and } i)\ /$$

$$= \mathcal{L}_{\text{JS}_{\pi'}}^{f_\eta^*}(\boldsymbol{e}^{(i)}, \bar{\boldsymbol{p}}_{>1}) - \mathcal{L}_{\text{JS}_{\pi'}}^{f^*}(\boldsymbol{e}^{(i)}, \boldsymbol{e}^{(y)})$$

$$= /\ iii) \text{ and Remark 2 } /$$

$$= \mathcal{L}_{\text{JS}_{\pi'}}^{f_\eta^*}(\boldsymbol{e}^{(i)}, \bar{\boldsymbol{p}}_{>1}) - C_1$$

$$\leq 0$$

From above and our assumption on the noise rates ($\eta_{yy} - \eta_{yi} > 0$), we have that the term in Equation 49 is less or equal to zero. Due to the assumption on the consistency of $f^*$ and $f_\eta^*$ in Proposition 4, this is also the case for the term in Equation 50. We have

$$R_{\mathcal{L}_{\text{GJS}}}^{\eta}(f^*) - R_{\mathcal{L}_{\text{GJS}}}^{\eta}(f_\eta^*) \leq (B_U^{\text{JS}} - B_L^{\text{JS}})\mathbb{E}_{\mathcal{D}}[\eta_{yy}]$$

Since $f_\eta^*$ is the global minimizer of $R_{\mathcal{L}_{\text{GJS}}}^{\eta}(f)$ we have $R_{\mathcal{L}_{\text{GJS}}}^{\eta}(f^*) - R_{\mathcal{L}_{\text{GJS}}}^{\eta}(f_\eta^*) \geq 0$, which is the lower bound. Hence, we have shown that $\mathcal{L}_{\text{JS}}$ and $\mathcal{L}_{\text{GJS}}$ have the same bounds for the risk difference for asymmetric noise. $\qquad\square$

## C.4 Bounds

In this section, we first introduce some useful definitions and relate them to JS. Then, the bounds for JS and GJS are proven.

### C.4.1 Another Definition of Jensen-Shannon divergence

$$f_{\pi_1}(t) := \Big[H(\pi_1 t + 1 - \pi_1) - \pi_1 H(t)\Big], t > 0 \tag{51}$$

$$f_{\pi_1}(0) := \lim_{t \to 0} f_{\pi_1}(t) \tag{52}$$

$$0 f_{\pi_1}\Big(\frac{0}{0}\Big) := 0, \tag{53}$$

$$0 f_{\pi_1}(0) := 0 \tag{54}$$

**Remark 3.** *The Jensen-Shannon divergence can be rewritten using Equation 51 as follows*

$$D_{\text{JS}_{\boldsymbol{\pi}}}(\boldsymbol{p}^{(1)}, \boldsymbol{p}^{(2)}) = \sum_{k=1}^{K} p_k^{(2)} f_{\pi_1}\Big(\frac{p_k^{(1)}}{p_k^{(2)}}\Big) \tag{55}$$

*Proof of Remark 3.*

$$\sum_{k=1}^{K} p_k^{(2)} f_{\pi_1}\left(\frac{p_k^{(1)}}{p_k^{(2)}}\right) = \sum_{k=1}^{K} p_k^{(2)} \left[\pi \frac{p_k^{(1)}}{p_k^{(2)}} \log(\frac{p_k^{(1)}}{p_k^{(2)}}) - (\pi \frac{p_k^{(1)}}{p_k^{(2)}} + 1 - \pi)\log(\pi \frac{p_k^{(1)}}{p_k^{(2)}} + 1 - \pi)\right] \tag{56}$$

$$= \sum_{k=1}^{K} \pi p_k^{(1)} \log(\frac{p_k^{(1)}}{p_k^{(2)}}) - (\pi p_k^{(1)} + (1-\pi)p_k^{(2)})\log(\frac{\pi p_k^{(1)} + (1-\pi)p_k^{(2)}}{p_k^{(2)}}) \tag{57}$$

$$= \sum_{k=1}^{K} \pi p_k^{(1)} \log(\frac{p_k^{(1)}}{p_k^{(2)}}) - \pi p_k^{(1)} \log(\frac{\pi p_k^{(1)} + (1-\pi)p_k^{(2)}}{p_k^{(2)}}) - (1-\pi)p_k^{(2)} \log(\frac{\pi p_k^{(1)} + (1-\pi)p_k^{(2)}}{p_k^{(2)}}) \tag{58}$$

$$= \sum_{k=1}^{K} \pi p_k^{(1)} \log(\frac{p_k^{(1)}}{\pi p_k^{(1)} + (1-\pi)p_k^{(2)}}) + (1-\pi)p_k^{(2)} \log(\frac{p_k^{(2)}}{\pi p_k^{(1)} + (1-\pi)p_k^{(2)}}) \tag{59}$$

$$= \sum_{k=1}^{K} \pi D_{\mathrm{KL}}\left(p_k^{(1)}, \pi p_k^{(1)} + (1-\pi)p_k^{(2)}\right) + (1-\pi)D_{\mathrm{KL}}\left(p_k^{(2)}, \pi p_k^{(1)} + (1-\pi)p_k^{(2)}\right) \tag{60}$$

$$= D_{\mathrm{JS}_\pi}(\boldsymbol{p}^{(1)}, \boldsymbol{p}^{(2)}) \tag{61}$$

$\square$

### C.4.2 Bounds for JS

**Proposition 5.** $\mathcal{L}_{\mathrm{JS}}$ *has* $B_L \leq \sum_{k=1}^{K} \mathcal{L}_{\mathrm{JS}}(\boldsymbol{e}^{(k)}, f(\boldsymbol{x})) \leq B_U$ *with*

$$B_L = \sum_{k=1}^{K} \mathcal{L}_{\mathrm{JS}}(\boldsymbol{e}^{(k)}, \boldsymbol{u}), \quad B_U = \sum_{k=1}^{K} \mathcal{L}_{\mathrm{JS}}(\boldsymbol{e}^{(k)}, \boldsymbol{e}^{(1)})$$

*where $\boldsymbol{u}$ is the uniform distribution.*

*Proof of Proposition 5.*
First we start with two observations: 1) $\sum_{k=1}^{K} \mathcal{L}_{\mathrm{JS}}(\boldsymbol{e}^{(k)}, \boldsymbol{p})$ is strictly convex. 2) $\sum_{k=1}^{K} \mathcal{L}_{\mathrm{JS}}(\boldsymbol{e}^{(k)}, \boldsymbol{p})$ is invariant to permutations of the components of $\boldsymbol{p}$.

First, we show Observation 1). This is done by using Remark 3 and showing that the second derivatives are larger than zero

$$f_{\pi_1}(t) := \left[H(\pi_1 t + 1 - \pi_1) - \pi_1 H(t)\right], t > 0 \tag{62}$$

$$f'_{\pi_1}(t) = \left[\pi_1(-\log(\pi_1 t + 1 - \pi_1) + \log(t))\right] \tag{63}$$

$$f''_{\pi_1}(t) = \frac{\pi_1(1-\pi_1)}{\pi_1 t^2 + t(1-\pi_1)} \tag{64}$$

Hence, $f_{\pi_1}(t)$ is strictly convex, since $\pi_1 > 0$ and $t > 0$, then $f''_{\pi_1}(t) > 0$. With Remark 3, and that the sum of strictly convex functions is also strictly convex, it follows that $\sum_{k=1}^{K} \mathcal{L}_{\mathrm{JS}}(\boldsymbol{e}^{(k)}, \boldsymbol{p})$ is strictly convex.

Next, we show Observation 2), *i.e.* that $\sum_{k=1}^{K} \mathcal{L}_{\mathrm{JS}}(e^{(k)}, p)$ is invariant to permutations of $p$

$$\sum_{k=1}^{K} D_{\mathrm{JS}}(e^{(k)}, p) = \sum_{k=1}^{K} \left[ H(\pi_1 e^{(k)} + \pi_2 p) - \pi_2 H(p) \right] \tag{65}$$

$$= \sum_{k=1}^{K} \left[ H(\pi_1 + \pi_2 p_k) + \sum_{i \neq k} H(\pi_2 p_i) - \pi_2 H(p) \right] \tag{66}$$

$$= \sum_{k=1}^{K} H(\pi_1 + \pi_2 p_k) + \sum_{k=1}^{K} \sum_{i \neq k} H(\pi_2 p_i) - \pi_2 K H(p) \tag{67}$$

$$= \sum_{k=1}^{K} H(\pi_1 + \pi_2 p_k) + \sum_{k=1}^{K} \left[ H(\pi_2 p) - H(\pi_2 p_k) \right] - \pi_2 K H(p) \tag{68}$$

$$= \sum_{k=1}^{K} H(\pi_1 + \pi_2 p_k) + (K-1) H(\pi_2 p) - \pi_2 K H(p) \tag{69}$$

$$= \sum_{k=1}^{K} H(\pi_1 + \pi_2 p_k) + (K-1)(H(\pi_2) + \pi_2 H(p)) - \pi_2 K H(p) \tag{70}$$

$$= \sum_{k=1}^{K} H(\pi_1 + \pi_2 p_k) + (K-1) H(\pi_2) - \pi_2 H(p) \tag{71}$$

Clearly, a permutation of the components of $p$ does not change the first sum or $H(p)$, since it would simply reorder the summands. Hence, $\sum_{k=1}^{K} \mathcal{L}_{\mathrm{JS}}(e^{(k)}, p)$ is invariant to permutations of $p$.

**Lower bound:**
The minimizer of a strictly convex function $\left( \sum_{k=1}^{K} \mathcal{L}_{\mathrm{JS}}(e^{(k)}, p) \right)$ over a compact convex set $\left( \Delta^{K-1} \right)$ is unique. Since $u$ is the only element of $\Delta^{K-1}$ that is the same under permutation, it is the unique minimum of $\sum_{k=1}^{K} \mathcal{L}_{\mathrm{JS}}(e^{(k)}, p)$ for $p \in \Delta^{K-1}$.

**Upper bound:**
The maximizer of a strictly convex function $\left( \sum_{k=1}^{K} \mathcal{L}_{\mathrm{JS}}(e^{(k)}, p) \right)$ over a compact convex set $\left( \Delta^{K-1} \right)$ is at its extreme points $\left( e^{(i)} \text{ for } i \in \{1, 2, \ldots, K\} \right)$. All extreme points have the same value according to Observation 2).

$\square$

### C.4.3 Bounds for GJS

**Proposition 3.** GJS *loss with* $M \leq K+1$ *satisfies* $B_L \leq \sum_{k=1}^{K} \mathcal{L}_{\mathrm{GJS}}(e^{(k)}, p^{(2)}, \ldots, p^{(M)}) \leq B_U$ *for all* $p^{(2)}, \ldots, p^{(M)} \in \Delta^{K-1}$, *with the following bounds*

$$B_L = \sum_{k=1}^{K} \mathcal{L}_{\mathrm{GJS}}(e^{(k)}, u, \ldots, u), \quad B_U = \sum_{k=1}^{K} \mathcal{L}_{\mathrm{GJS}}(e^{(k)}, e^{(1)}, \ldots, e^{(M-1)})$$

*where* $u \in \Delta^{K-1}$ *is the uniform distribution.*

*Proof of Proposition 3.*
**Lower bound:** Using Proposition 2 to rewrite GJS into a JS and a consistency term, we get

$$\sum_{k=1}^{K} D_{\text{GJS}_{\pi}}(e^{(k)}, p^{(2)}, \dots, p^{(M)}) = \sum_{k=1}^{K} \left[ D_{\text{JS}_{\pi'}}(e^{(k)}, \bar{p}_{>1}) + (1 - \pi_1) D_{\text{GJS}_{\pi''}}(p^{(2)}, \dots, p^{(M)}) \right]$$
(72)

$$= \sum_{k=1}^{K} D_{\text{JS}_{\pi'}}(e^{(k)}, \bar{p}_{>1}) + (1 - \pi_1) K D_{\text{GJS}_{\pi''}}(p^{(2)}, \dots, p^{(M)}) \quad (73)$$

$$\geq \sum_{k=1}^{K} D_{\text{JS}_{\pi'}}(e^{(k)}, u) + (1 - \pi_1) K D_{\text{GJS}_{\pi''}}(p^{(2)}, \dots, p^{(M)}) \quad (74)$$

$$\geq \sum_{k=1}^{K} D_{\text{JS}_{\pi'}}(e^{(k)}, u) \quad (75)$$

where the first inequality comes from the lower bound of Proposition 5, and the second inequality comes from
$(1 - \pi_1) K D_{\text{GJS}_{\pi''}}(p^{(2)}, \dots, p^{(M)})$ being non-negative. The inequalities holds with equality if and only if
$p^{(2)} = \dots = p^{(M)} = u$. Notably, the lower bound of JS is the same as that of GJS.

**Upper bound:**
Let's denote $A(p^{(2)}, \dots, p^{(M)}) = \sum_{k=1}^{K} \mathcal{L}_{\text{GJS}}(e^{(k)}, p^{(2)}, \dots, p^{(M)})$. First we start by making 5 observations:
Observation 1: $\Delta_{M-1}^{K-1} = \Delta^{K-1} \times \Delta^{K-1} \times \dots \times \Delta^{K-1}$ is a compact convex set.

Observation 2: $A$ is strictly convex over $\Delta_{M-1}^{K-1}$.
Observation 3: From Observations 1 and 2 we have that the maximizer of $A$ should be at extreme points of $\Delta_{M-1}^{K-1}$, *i.e.*, a unit vector in every $M - 1$ individual $\Delta^{K-1}$ subspaces of $\Delta_{M-1}^{K-1}$.
Observation 4: $A$ is symmetric w.r.t. permutations of the components of predictive distributions $p^{(i)}$.

Unlike for JS, the extreme points of $\Delta_{M-1}^{K-1}$ do not necessarily map to the same value of $A$. Hence, what is left to show is that the set of extreme points with all predictive distributions being *distinct* unit vectors maps to the maximum value of $A$.

Given Observation 3, all the M distributions are unit vectors, therefore the maximum is of the form
$A(p^{(2)}, \dots, p^{(M)}) = \sum_{k=1}^{K} H(\pi_1 e^{(k)} + (1 - \pi_1) \bar{p}_{>1})$, where $\bar{p}_{>1} := \sum_{j=2}^{M} \pi_j p^{(j)} / (1 - \pi_1)$.
Furthermore, at most $M - 1$ components of $\bar{p}_{>1}$ are non-zero (if all predictions are distinct). From Observation 4, we can WLOG permute $\bar{p}_{>1}$ such that the first $M - 1$ components are the largest ones. Let $\bar{p}_{>1}^{\complement} \in \Delta^{M-2}$ denote the subset of these first $M - 1$ components of $\bar{p}_{>1} \in \Delta^{K-1}$. Then, for all predictive distributions being unit vectors, we have

$$A(p^{(2)}, \dots, p^{(M)}) = \sum_{k=1}^{K} H(\pi_1 e^{(k)} + (1 - \pi_1) \bar{p}_{>1}) \quad (76)$$

$$= \sum_{k=1}^{M-1} \left[ H(\pi_1 + (1 - \pi_1) m_{>1,k}) + (K - 1) H((1 - \pi_1) m_{>1,k}) \right] + \sum_{k=M}^{K} H(\pi_1) \quad (77)$$

$$= \sum_{k=1}^{M-1} H(\pi_1 + (1 - \pi_1) m_{>1,k}) + (K - 1) H((1 - \pi_1) \bar{p}_{>1}^{\complement}) + \sum_{k=M}^{K} H(\pi_1) \quad (78)$$

$$\leq (M - 1) H\left( \frac{1}{M-1} \sum_{k=1}^{M-1} \left[ \pi_1 + (1 - \pi_1) m_{>1,k} \right] \right) + (K - 1) H((1 - \pi_1)) \bar{p}_{>1}^{\complement}) + \sum_{k=M}^{K} H(\pi_1)$$
(79)

$$= (M - 1) H\left( \pi_1 + \frac{1 - \pi_1}{M - 1} \right) + (K - 1) H((1 - \pi_1) \bar{p}_{>1}^{\complement}) + \sum_{k=M}^{K} H(\pi_1) \quad (80)$$

$$\leq (M - 1) H\left( \pi_1 + \frac{1 - \pi_1}{M - 1} \right) + (K - 1) H((1 - \pi_1) u) + \sum_{k=M}^{K} H(\pi_1) \quad (81)$$

The first inequality follows from Jensen's inequality and the second from the uniform distribution maximizes entropy. Both inequalities hold with equality iff $m_{>1,1} = \cdots = m_{>1,M-1}$. Hence, the maximum is achieved if $\bar{\boldsymbol{p}}^{\subseteq}_{>1} = \boldsymbol{u} \in \Delta^{M-2}$, which is only possible if all $M-1$ predictive distributions are distinct unit vectors.

$\square$

### C.5 Robustness of Jensen-Shannon losses

In this section, we prove that the lower ($B_L$) and upper ($B_U$) bounds become the same for JS and GJS as $\pi_1 \to 1$ as stated in Remark 1.

**Remark 1.** *$\mathcal{L}_{\text{JS}}$ and $\mathcal{L}_{\text{GJS}}$ are robust ($B_L = B_U$) in the limit of $\pi_1 \to 1$.*

*Proof of Remark 1 for* JS.
**Lower bound:**

$$\sum_{k=1}^{K} D_{\text{JS}_\pi}(\boldsymbol{e}^{(y)}, \mathbf{u}) = \sum_{k=1}^{K} H(\pi_1 \boldsymbol{e}^{(k)} + \pi_2 \mathbf{u}) - \pi_2 H(\mathbf{u}) \tag{82}$$

$$= K[H(\pi_1 \boldsymbol{e}^{(1)} + \pi_2 \mathbf{u}) - \pi_2 H(\mathbf{u})] \tag{83}$$

$$= K[H(\pi_1 + \pi_2/K) + (K-1)H(\pi_2/K) - K\pi_2 H(\frac{1}{K})] \tag{84}$$

$$= /H(\pi_2/K) = -\pi_2/K(\log \pi_2 + \log 1/K) = \frac{1}{K}H(\pi_2) + \pi_2 H(1/K)/ \tag{85}$$

$$= K[H(\pi_1 + \pi_2/K) + (K-1)(\frac{1}{K}H(\pi_2) + \pi_2 H(\frac{1}{K})) - K\pi_2 H(\frac{1}{K})] \tag{86}$$

$$= K[H(\pi_1 + \pi_2/K) + (K-1)\frac{1}{K}H(\pi_2) - \pi_2 H(\frac{1}{K})] \tag{87}$$

If one now normalize($Z = H(\pi_2) = H(1 - \pi_1)$) and take the limit as $\pi_1 \to 1$ we get:

$$\lim_{\pi_1 \to 1} \sum_{k=1}^{K} \mathcal{L}_{\text{JS}}(\boldsymbol{e}^{(y)}, \mathbf{u}) = \lim_{\pi_1 \to 1} (K-1) + K\frac{H(\pi_1 + \pi_2/K) - \pi_2 H(\frac{1}{K})}{H(\pi_2)} \tag{88}$$

$$= \lim_{\pi_1 \to 1} (K-1) + K\frac{-(K-1)(1 + \log(\pi_1 + \pi_2/K))/K - \log(1/K)/K}{\log(1 - \pi_1) + 1} \tag{89}$$

$$= \lim_{\pi_1 \to 1} (K-1) - \frac{(K-1)(1 + \log(\pi_1 + \pi_2/K)) - \log(1/K)}{\log(1 - \pi_1) + 1} \tag{90}$$

$$= \lim_{\pi_1 \to 1} (K-1) - ((K-1)(1 + \log(\pi_1 + \pi_2/K)) - \log(1/K))\frac{1}{\log(1 - \pi_1) + 1} \tag{91}$$

$$= (K-1) - (K-1 - \log(1/K)) \cdot 0 \tag{92}$$

$$= K - 1 \tag{93}$$

where L'Hôpital's rule was used for the fraction in Equation 88 which is indeterminate of the form $\frac{0}{0}$.
**Upper bound:**

$$\sum_{k=1}^{K} \mathcal{L}_{\text{JS}}(\boldsymbol{e}^{(k)}, \boldsymbol{e}^{(1)}) = \frac{1}{H(\pi_2)} \sum_{k=1}^{K} H(\pi_1 \boldsymbol{e}^{(k)} + \pi_2 \boldsymbol{e}^{(1)}) \tag{94}$$

$$= \frac{1}{H(\pi_2)}[(K-1)H(\pi_2) + (K-1)H(\pi_1) + H(\pi_1 + \pi_2)] \tag{95}$$

$$= (K-1)[1 + \frac{H(\pi_1)}{H(\pi_2)}] \tag{96}$$

$$= (K-1)\left[1 + \frac{\pi_1 \log \pi_1}{(1 - \pi_1) \log(1 - \pi_1)}\right] \tag{97}$$

Taking the limit as $\pi_1 \to 1$ gives

$$\lim_{\pi_1 \to 1} \sum_{k=1}^{K} \mathcal{L}_{\text{JS}}(\boldsymbol{e}^{(k)}, \boldsymbol{e}^{(1)}) = \lim_{\pi_1 \to 1} (K-1) \left[ 1 + \pi_1 \frac{1}{\log(1-\pi_1)} \frac{\log \pi_1}{(1-\pi_1)} \right] \tag{98}$$

$$= \lim_{\pi_1 \to 1} (K-1) \left[ 1 + \pi_1 \frac{1}{\log(1-\pi_1)} \frac{1}{\pi_1} \frac{1}{-1} \right] \tag{99}$$

$$= (K-1)[1 + 1 \cdot 0 \cdot 1 \cdot -1] \tag{100}$$

$$= K - 1 \tag{101}$$

where L'Hôpital's rule was used for $\lim_{\pi_1 \to 1} \frac{\log \pi_1}{(1-\pi_1)}$ which is indeterminate of the form $\frac{0}{0}$.
Hence, $B_L = B_U = K - 1$. $\qquad\square$

Next, we look at the robustness of the generalized Jensen-Shannon loss.

*Proof of Remark 1 for* GJS.
Proposition 2, shows that GJS can be rewritten as a JS term and a consistency term. From the proof of Remark 1 for JS above, it follows that the JS term satisfies $B_L = B_U$ as $\pi_1$ approaches 1. Hence, it is enough to show that the consistency term of GJS also becomes a constant in this limit. The consistency term is the generalized Jensen-Shannon divergence

$$\lim_{\pi_1 \to 1} (1-\pi_1) \mathcal{L}_{\text{GJS}_{\boldsymbol{\pi}''}}(\boldsymbol{p}^{(2)}, \ldots, \boldsymbol{p}^{(M)}) = \lim_{\pi_1 \to 1} \frac{(1-\pi_1)}{H(1-\pi_1)} D_{\text{GJS}_{\boldsymbol{\pi}''}}(\boldsymbol{p}^{(2)}, \ldots, \boldsymbol{p}^{(M)}) \tag{102}$$

$$= \lim_{\pi_1 \to 1} -\frac{1}{\log(1-\pi_1)} D_{\text{GJS}_{\boldsymbol{\pi}''}}(\boldsymbol{p}^{(2)}, \ldots, \boldsymbol{p}^{(M)}) \tag{103}$$

$$= 0 \tag{104}$$

where $\boldsymbol{\pi}'' = [\pi_2, \ldots, \pi_M]/(1-\pi_1)$. $D_{\text{GJS}_{\boldsymbol{\pi}''}}(\boldsymbol{p}^{(2)}, \ldots, \boldsymbol{p}^{(M)})$ is bounded and $-\frac{1}{\log(1-\pi_1)}$ goes to zero as $\pi_1 \to 1$, hence the limit of the product goes to zero. $\qquad\square$

## C.6 Gradients of Jensen-Shannon Divergence

The partial derivative of the Jensen-Shannon divergence is

$$\frac{\partial \{H(\boldsymbol{m}) - \pi_1 H(\boldsymbol{e}^{(y)}) - (1-\pi_1) H(\boldsymbol{p})\}}{\partial z_i}$$

where $\boldsymbol{m} = \pi_1 \boldsymbol{e}^{(y)} + \pi_2 \boldsymbol{p} = \pi_1 \boldsymbol{e}^{(y)} + (1-\pi_1)\boldsymbol{p}$, and $p_j = e^{z_j} / \sum_{k=1}^{K} e^{z_k}$. Note the difference between $e^z$ which is the exponential function while $\boldsymbol{e}^{(y)}$ is a onehot label. We take the partial derivative of each term separately, but first the partial derivative of the $j$th component of a softmax output with respect to the $i$th component of the corresponding logit

$$\frac{\partial p_j}{\partial z_i} = \frac{\partial}{\partial z_i} \frac{e^{z_j}}{\sum_{k=1}^{K} e^{z_k}} \tag{105}$$

$$= \frac{\frac{\partial e^{z_j}}{\partial z_i} \sum_{k=1}^{K} e^{z_k} - e^{z_j} \frac{\partial \sum_{k=1}^{K} e^{z_k}}{\partial z_i}}{\left( \sum_{k=1}^{K} e^{z_k} \right)^2} \tag{106}$$

$$= \frac{\mathbb{1}(i=j) e^{z_j} \sum_{k=1}^{K} e^{z_k} - e^{z_j} e^{z_i}}{\left( \sum_{k=1}^{K} e^{z_k} \right)^2} \tag{107}$$

$$= \frac{\mathbb{1}(i=j) e^{z_j} - p_j e^{z_i}}{\sum_{k=1}^{K} e^{z_k}} \tag{108}$$

$$= \mathbb{1}(i=j) p_j - p_j p_i \tag{109}$$

$$= p_j (\mathbb{1}(i=j) - p_i) \tag{110}$$

$$= p_i (\mathbb{1}(i=j) - p_j) = \frac{\partial p_i}{\partial z_j} \tag{111}$$

where $\mathbb{1}(i = j)$ is the indicator function, *i.e.* 1 when $i = j$ and zero otherwise. Using the above, we get

$$\sum_{j=1}^{K} \frac{\partial p_j}{\partial z_i} = p_i \sum_{j=1}^{K} (\mathbb{1}(i = j) - p_j) = p_i(1 - 1) = 0 \tag{112}$$

First, the partial derivative of $H(\boldsymbol{p})$ wrt $z_i$

$$\frac{\partial H(\boldsymbol{p})}{\partial z_i} = -\sum_{j=1}^{K} \frac{\partial p_j \log p_j}{\partial z_i} \tag{113}$$

$$= -\sum_{j=1}^{K} \frac{\partial p_j}{\partial z_i} \log p_j + p_j \frac{\partial \log p_j}{\partial z_i} \tag{114}$$

$$= -\sum_{j=1}^{K} \frac{\partial p_j}{\partial z_i} \log p_j + p_j \frac{1}{p_j} \frac{\partial p_j}{\partial z_i} \tag{115}$$

$$= -\sum_{j=1}^{K} \frac{\partial p_j}{\partial z_i} (\log p_j + 1) \tag{116}$$

$$= / \text{ Equation 112 } / \tag{117}$$

$$= -\sum_{j=1}^{K} \frac{\partial p_j}{\partial z_i} \log p_j \tag{118}$$

Next, the partial derivative of $H(\boldsymbol{m})$ wrt $z_i$

$$\frac{\partial \{H(\boldsymbol{m})\}}{\partial z_i} = \frac{\partial \{\pi_1 H(\boldsymbol{e}^{(y)}, \boldsymbol{m}) + (1 - \pi_1) H(\boldsymbol{p}, \boldsymbol{m})\}}{\partial z_i} \tag{119}$$

$$= -\sum_{j=1}^{K} \left[ \pi_1 \frac{e_j^{(y)} \partial \log(m_j)}{\partial z_i} + (1 - \pi_1) \frac{\partial \{p_j \log(m_j)\}}{\partial z_i} \right] \tag{120}$$

$$= -\sum_{j=1}^{K} \left[ \pi_1 e_j^{(y)} \frac{\partial \log(m_j)}{\partial z_i} + (1 - \pi_1) \left( \frac{\partial p_j}{\partial z_i} \log(m_j) + p_j \frac{\partial \log(m_j)}{\partial z_i} \right) \right] \tag{121}$$

$$= -\sum_{j=1}^{K} \left[ m_j \frac{\partial \log(m_j)}{\partial z_i} + (1 - \pi_1) \frac{\partial p_j}{\partial z_i} \log(m_j) \right] \tag{122}$$

$$= -\sum_{j=1}^{K} \left[ (1 - \pi_1) \frac{\partial p_j}{\partial z_i} + (1 - \pi_1) \frac{\partial p_j}{\partial z_i} \log(m_j) \right] \tag{123}$$

$$= -\sum_{j=1}^{K} (1 - \pi_1) \frac{\partial p_j}{\partial z_i} \left[ 1 + \log(m_j) \right] = / \text{ Equation 112 } / \tag{124}$$

$$= -(1 - \pi_1) \sum_{j=1}^{K} \frac{\partial p_j}{\partial z_i} \log(m_j) \tag{125}$$

The partial derivative of the Jensen-Shannon divergence with respect to logit $z_i$ is

$$\frac{\partial \{H(\boldsymbol{m}) - \pi_1 H(\boldsymbol{e}^{(y)}) - (1 - \pi_1) H(\boldsymbol{p})\}}{\partial z_i} = \frac{\partial \{H(\boldsymbol{m}) - (1 - \pi_1) H(\boldsymbol{p})\}}{\partial z_i} \tag{126}$$

$$= -(1 - \pi_1) \sum_{j=1}^{K} \frac{\partial p_j}{\partial z_i} \left( \log(m_j) - \log p_j \right) \tag{127}$$

$$= -(1 - \pi_1) \left[ \sum_{j=1}^{K} \frac{\partial p_j}{\partial z_i} \log \frac{m_j}{p_j} \right] \tag{128}$$

If we now make use of the fact that the label is $\boldsymbol{e}^{(y)}$, we can write the partial derivative wrt to $z_i$ as

$$\frac{\partial\{H(\boldsymbol{m}) - \pi_1 H(\boldsymbol{e}^{(y)}) - (1-\pi_1)H(\boldsymbol{p})\}}{\partial z_i} = \tag{129}$$

$$= -(1-\pi_1)\Big[\sum_{j=1}^{K}\frac{\partial p_j}{\partial z_i}\log\left(\frac{\pi_1 e_j^{(y)}}{p_j} + (1-\pi_1)\right)\Big] \tag{130}$$

$$= -(1-\pi_1)\Big[\frac{\partial p_y}{\partial z_i}\log\left(\frac{\pi_1}{p_y} + (1-\pi_1)\right) + \sum_{j\neq y}^{K}\frac{\partial p_j}{\partial z_i}\log\left(1-\pi_1\right)\Big] \tag{131}$$

$$= -(1-\pi_1)\Big[\frac{\partial p_y}{\partial z_i}\log\left(\frac{\pi_1}{p_y} + (1-\pi_1)\right) + \log\left(1-\pi_1\right)\sum_{j\neq y}^{K}\frac{\partial p_j}{\partial z_i}\Big] \tag{132}$$

$$= \Big/\text{Eq 112} \Leftrightarrow \sum_{j\neq y}^{K}\frac{\partial p_j}{\partial z_i} = -\frac{\partial p_y}{\partial z_i}\Big/ \tag{133}$$

$$= -(1-\pi_1)\frac{\partial p_y}{\partial z_i}\Big[\log\left(\frac{\pi_1}{p_y} + (1-\pi_1)\right) - \log\left(1-\pi_1\right)\Big] \tag{134}$$

$$= -(1-\pi_1)\frac{\partial p_y}{\partial z_i}\log\left(\frac{\pi_1}{(1-\pi_1)p_y} + 1\right) \tag{135}$$

## D  Extended Related Works

Most related to us is the avenue of handling noisy labels in deep learning via the identification and construction of *noise-robust loss functions* [2, 3, 4, 5]. Ghosh *et al.* [2] derived sufficient conditions for a loss function, in empirical risk minimization (ERM) settings, to be robust to various kinds of sample-independent noise, including symmetric, symmetric non-uniform, and class-conditional. They further argued that, while CE is not a robust loss function, mean absolute error (MAE) is a loss that satisfies the robustness conditions and empirically demonstrated its effectiveness. On the other hand, Zhang *et al.* [3] pointed out the challenges of training with MAE and proposed GCE which generalizes both MAE and CE losses. Tuning for this trade-off, GCE alleviates MAE's training difficulties while retaining some desirable noise-robustness properties. In a similar fashion, symmetric cross entropy (SCE) [4] spans the spectrum of reverse CE as a noise-robust loss function and the standard CE. Recently, Ma *et al.* [5] proposed a normalization mechanism to make arbitrary loss functions robust to noise. They, too, further combine two complementary loss functions to improve the data fitting while keeping robust to noise. The current work extends on this line of works.

Several other directions are pursued to improve training of deep networks under noisy labeled datasets. This includes methods to *identify and remove* noisy labels [35, 36] or *identify and correct* noisy labels in a joint label-parameter optimization [37, 38] and those works that design an *elaborate training pipeline* for dealing with noise [16, 39, 40]. In contrast to these directions, this work proposes a robust loss function based on Jensen-Shannon divergence (JS) without altering other aspects of training. In the following, we review the directions that are most related to this paper.

A close line of works to ours *reweight a loss function* by a known or estimated class-conditional noise model [11]. This direction has been commonly studied for deep networks with a standard cross entropy (CE) loss [12, 13, 14, 15]. Assuming a class-conditional noise model, loss correction is theoretically well motivated.

A common regularization technique called *label smoothing* [41] has been recently proposed that operates similarly to the loss correction methods. While its initial purpose was for deep networks to avoid overfitting, label smoothing has been shown to have a noticeable effect when training with noisy sets by alleviating the fit to the noise [23, 24].

*Consistency regularization* is a recently-developed technique that encourages smoothness in the learnt decision boundary by requiring minimal shifts in the learnt function when small perturbations are applied to an input sample. This technique has become increasingly common in the state-of-the-art semi-supervised learning [42, 43, 44] and recently for dealing with noisy data [16]. These methods

use various complicated pipelines to integrate consistency regularization in training. This work shows that a multi-distribution generalization of JS can neatly incorporate such regularization.

Hendrycks *et al.* [6] recently proposed AugMix, a novel data augmentation strategy in combination with a GJS consistency loss to improve uncertainty estimation and robustness to image corruptions at test-time. Our work is orthogonal since we consider the task of learning under noisy labels at training time and conduct the corresponding experiments. We also investigate and derive the theoretical properties of the proposed loss functions. Finally, our losses are solely implemented based on JS/GJS instead of a combination of CE and GJS in case of AugMix. However, we find it promising that GJS improves robustness to both training-time label noise and test-time image corruption, which further strengthens the significance of the JS-based loss functions.

Finally, recently, Xu *et al.* [18]; Wei & Liu [19] propose loss functions with *information theory* motivations. Jensen-Shannon divergence, with inherent information theoretic interpretations, naturally posits a strong connection of our work to those. Especially, the latter is a close *concurrent* work that studies the general family of $f$-divergences but takes a different and complementary angle. In this work, we analyze the role of $\pi_1$, which they treat as a constant. Varying $\pi_1$ is important because it leads to:

- **Better empirical performance.** For our experiments on CIFAR, we provide the hyper-parameters used in Table 7, from which we can see that the optimal is equal to their setting ($\pi_1 = 0.5$) in only 3/14 cases.
- **Interesting theoretical connections to related work.** In Proposition 1, we show that the JS loss has CE and MAE as asymptotes when $\pi_1$ goes to zero and one, respectively. This causes an interesting trade-off between learnability and robustness as discussed in Section 4.3.

Furthermore, we consider the generalization to more than two distributions which have proved helpful while Wei & Liu [19] only study two distributions.

In this work, we use a generalization of the Jensen-Shannon divergence to more than two distributions, which was introduced by Lin [8]. Recently, another generalization of JS was presented by Nielsen [21], where the arithmetic mean is generalized to abstract means. JS is also a special case of a general family of divergences, the f-divergences [20].