# OpenReview forum: "Generalized Jensen-Shannon Divergence Loss for Learning with Noisy Labels"
_NeurIPS.cc/2021/Conference — NeurIPS 2021 Poster_

### Official Review · Reviewer_HdUW · 2021-07-04

**Rating:** 6
**Confidence:** 4

**Summary:**

The paper targets at learning with noisy labels and propose two loss functions: JS and GJS. To justify the loss functions, robustness and consistency properties are discussed.

**Limitations And Societal Impact:**

Yes.

**Main Review:**

**Pros**

-	The theoretical analysis on JS and GJS are clear. The justifications on robustness and consistency make sense.
-	There are ablation studies showing the effect of each component.


**Cons**

*Mythology*

-	The contribution is minor compared with previous works. 1) For the main claims of robustness and underfitting, JS shares the same properties as GCE. 2) The superior of GJS over JS is the consistency property. GJS is equivalent to CE plus a regularization term that encourages low KL between predictions on perturbed inputs. Such a regularization term is widely known in the community: it has long been used in semi-supervised learning, noisy label learning and adversial training. We should be aware that directly introducing such a consistency regularization on any existing methods may improve the performance in learning with noisy labels by a large margin (I have verified this in experiments and the author could conduct experiments if interested). Considering that the consistency regularization is known, the contribution of this paper is minor.
-	The observation on network predictions’ consistency is not novel as claimed in the main contributions. It is straightforward for random noise because the labels are flipped to different classes for similar samples (neighbor samples in some sense) and the input spaces are divided into much more decision regions compared with training on clean. Actually, this has been demonstrated in [Aprit et al. ICML 2017] (please refer to the Critical Sample Ratio which is an estimation of the density of decision boundaries). Naturally, the observation on consistency should be less significant for class-dependent noise and more realistic instance-dependent noise.
-	In this direction, the line of works that study robustness mostly based on the assumption that the label noise is class-conditional (even random as assumed in Theorem 1 in this paper). However, both intuition and rigorous hypothesis testing [Zhang et al. ICLR 2021] imply that real-world noise should be instance dependent.

*Experiments*
-	The empirical results are not significant on more realistic asymmetric noise and real-world noise. This is consistent with the above concerns that the theoretical justification is based on strong assumptions and the consistency observation is most significant for random noise.
-	The author does not compare to methods that propose a full pipeline and claim that the reason is robust loss functions can be considered complementary to existing methods. However, firstly, there are many strong yet simple baselines (sample selection, label correction methods) that are necessary to compare on large-scale experiments on CIFAR-10 and CIFAR-100. Secondly, the paper should demonstrate an improvement by introducing GJS on existing methods since it is claimed as complementary.


[Aprit et al. ICML 2017] A Closer Look at Memorization in Deep Networks.
[Zhang et al. ICLR 2021] Learning with Feature-Dependent Label Noise: A Progressive Approach.


*post rebuttal*

Thanks for the detailed feedback. My concerns are well addressed.

**Time Spent Reviewing:**

4

---

> ### Author Response · Authors · 2021-08-10
> **Thank you for your detailed feedback. [1/2]**
>
> Thank you for your time and effort reviewing our paper. We appreciate the thoughtful feedback. We are glad you found the theoretical analysis of JS and GJS clear, and the justifications for robustness reasonable. We are happy you recognize our ablation studies as a strength of our work.
>
> ***Novelty of the consistency observation***
>
> Thank you for drawing our attention to the work of Aprit et al. (ICML 2017). The Critical Sample Ratio (CSR) in Aprit et al. (ICML 2017) measures the ratio of examples in the dataset having at least one adversarial example in its neighborhood. They observe that this ratio increases for increasing amounts of label noise. That is, training with more label noise makes it easier to find adversarial examples. This is different from our observation in that:
> * Their perturbations (adversarial examples) investigate the behavior of the network locally around the dataset points which are usually off the natural image support, while our perturbations (strong image augmentations) study a more global behavior of the network on naturally-augmented images.
> * They do not observe that the consistency trend is different on clean and noisy labelled examples which is central to our observation.
> * The behavior of CSR is different from what we observe in terms of consistency. CSR increases during training until it plateaus, see Figure 7b) and 8b) in their paper. That is, during training it becomes easier to find adversarial examples. However, what we observe is that the consistency of the network first increases (probably as the network learns from correctly labelled samples), then decreases as the network overfits to noise.
>
>
>
> ***Minor contributions***
>
> **“For the main claims of robustness and underfitting, JS shares the same properties as GCE. The superior of GJS over JS is the consistency property.”**
>
> It is true that our justification for robustness to label noise for JS is similar to GCE. However, this connection is only possible through our non-trivial theoretical contribution in Proposition 1, showing how JS is related to CE and MAE (which GCE also is). The fact that a paper finds a connection in one aspect of two completely different loss functions does not take away the novelty of one of those. In fact, we believe, drawing such connections should be considered a merit and a contribution.
>
> Similarly, we believe that it is unfair to make the superiority of GJS over JS due to the consistency term sound like a trivial observation. Note that such observation is only apparent thanks to our theoretical results in Proposition 2 and a thorough ablation studies. Table 3 that investigates the role of consistency in GJS, would not be possible without our theoretical results in Proposition 2.
>
> **“GJS is equivalent to CE plus a regularization term that encourages low KL between predictions on perturbed inputs.”**
>
> The claim that GJS is equivalent to CE plus a regularization term minimizing KL between predictions on perturbed inputs is incorrect.
> * GJS is not an ad hoc combination of CE and KL: GJS is a single principled loss function supporting more than two distributions, which our non-trivial theoretical contribution in Proposition 2 shows is equivalent to a label-dependent term and a consistency term. GJS, being a principled and well-known divergence, can enable more fruitful future direction than an ad hoc combination of CE and KL. For instance see the concurrent work of Wei & Liu. (ICLR 2021) that analyzes the robustness properties of JS from a different perspective.
> * CE and JS are not equivalent as label-dependent terms: We theoretically show in Proposition 1 that JS is a generalization of CE and MAE and is only equivalent to CE asymptotically as $\pi$ goes to 0. We show both empirically and theoretically that using JS as the loss makes the network robust to label noise. Overall, JS performs much better than CE as can be seen in Table 1 and 2.
> * GJS and KL are not equivalent as consistency terms: GJS provides a principled way of encouraging consistency between more than two distributions, which is non-trivial to do with a KL term. Even if only two distributions are used, JS is symmetric while KL is not. Therefore, KL will treat the two predictions on perturbed inputs differently, while JS would treat them equally. Furthermore JS (GJS) is bounded while KL is unbounded.
>
>
> **”directly introducing such a consistency regularization on any existing methods may improve the performance in learning with noisy labels by a large margin”**
>
> Recognizing that adding consistency regularization *potentially* improves the consistency of other methods would not contradict any of our claims or lessen the significance of our results. On the contrary, we believe we have shown that the consistency of GJS works so well when learning with noisy labels that it opens up interesting opportunities for similar works.
>
> Now, we do an experiment that follows two claims of the review (i) no difference between GJS and CE + KL (ii) no difference between adversarial perturbation (observation in CSR) and natural image augmentation (observation in our paper). Accordingly, we use Virtual Adversarial Training (VAT) which is a well-known consistency regularization technique for semi-supervised learning proposed by Miyato et al. (ICLR 2016). VAT encourages low KL between the predictions on adversarially perturbed images. The full loss is $CE(y,p)+KL(p_{adv},p)$, where $y$ is the one-hot label, $p$ is a prediction on an augmented image and $p_{adv}$ is an adversarially perturbed version of the augmented image. We use exactly the same training setup and hyperparameter search as for all the methods in Table 1, i.e., we first search for the best learning rate and weight decay, then for the best hyper-parameters of VAT ($\epsilon \in [1.0, 2.0, 4.0, 8.0, 16.0]$) for each noise rate. We train the best setting five times and report the mean and standard deviation on the test set. The results are shown in the table below.
>
> Method | &nbsp;&nbsp;&nbsp; Sym 20% &nbsp;&nbsp;&nbsp; | &nbsp;&nbsp;&nbsp; Sym 60% &nbsp;&nbsp;&nbsp;
> :---|:----:|:----:
> CE | 65.74 ± 0.22  | 44.42 ± 0.84
> CE+VAT | 66.06 ± 0.41 | 45.49 ± 0.21
> GJS | **78.05 ± 0.25** | **70.15 ± 0.30**
>
> This experiment shows that adding any type of consistency regularization doesn’t necessarily help much. In this work, we propose a simple yet highly effective method and provide strong evidence through extensive experiments that it works well, not only on synthetic noise, but it also achieves state-of-the-art performance on real-world noise.
>
> Furthermore, we measure the consistency of the network trained with VAT, which is 27.90 ± 0.33. This can be compared with the consistency of 27.88 ± 0.34 and 43.60 ± 0.26 of CE and GJS respectively. Even though VAT explicitly minimizes the KL between an adversarial example and its original sample, the consistency of the method, as measured by our metric, is poor. This questions the relationship between CSR and our consistency measure even more. For more consistency results on other types of synthetic noise, see response to reviewer ogFS.
>
> **“[consistency] regularization term is widely known in the community: it has long been used in semi-supervised learning, noisy label learning and adversial training.”**
>
> We do not claim to have proposed consistency regularization as a new and general method for all deep learning applications. We are specifically interested in learning with noisy labels. To the best of our knowledge, consistency regularization has not been shown to be so effective for this task as we show in our paper. We believe the closest related work in terms of consistency regularization is DivideMix. This elaborate method has several components: 1) It trains two networks at the same time, 2) it tries to separate clean and noisy labelled examples by using Gaussian mixture models based on the cross-entropy loss of the examples, 3) the two networks then swap their partitions of the training set, 4) the partitioned training set is then used in a semi-supervised learning method called MixMatch. This MixMatch approach is itself an elaborate method composed of several steps, one of which is consistency regularization.
>
> Due to all these components, we argue that it is unclear how big of a role consistency regularization has in DivideMix. Our simple yet highly effective method outperforms DivideMix on real-world noise, see Table 2 in the main paper.

---

> > ### Author Response · Authors · 2021-08-10
> > **Thank you for your detailed feedback. [2/2]**
> >
> > ***“Naturally, the observation on consistency should be less significant for class-dependent noise and more realistic instance-dependent noise”***
> >
> > Thanks for drawing our attention to the work of Zhang et al. (ICLR 2021). We perform an experiment similar to Figure 1 in the paper where we analyze the consistency, but this time for asymmetric (10%, 20%, 30%, 40%) and instance-dependent (35% Type I) noise proposed by Zhang et al. (ICLR 2021).
> >
> > For both asymmetric and instance-dependent noise, the consistency of clean examples are mostly increasing or constant during training. However, the consistency of noisy examples increases in the high learning rate scheme (epochs 1-200), followed by a large increase in consistency that is then degrading until epoch 300, where the consistency is increasing again (only exception is for 10% asym).
> >
> > At the end of training, the consistency of clean examples are 8-30% higher than for the noisy labelled examples for asymmetric noise. The corresponding number for the instance-dependent noise is 23%.
> >
> > To summarize, the general trends are similar for all noise types including symmetric and asymmetric class-dependent as well as instance-dependent noises. However, your intuition that the drop in consistency is less on asymmetric and instance-dependent noise is correct. We thank you for this insight. In the revision of the paper, we will do a thorough study of this. This opens up interesting future work to better understand why this happens and how one can tackle it to further improve the performance of GJS.
> >
> > ***"both intuition and rigorous hypothesis testing [Zhang et al. ICLR 2021] imply that real-world noise should be instance dependent."***
> >
> > In this rebuttal, we provide further experiments on instance-dependent noise (Type-I) proposed by Zhang et al. ICLR 2021, see response to reviewer xNPb. Furthermore, we provide results on two more real-world noise datasets below.
> >
> > ***“The empirical results are not significant on more realistic asymmetric noise and real-world noise.”***
> >
> > The claim that our results are not significant on more realistic noise like asymmetric and real-world noise is unfounded. Our empirical results on these noise types are very competitive, as we show below.
> >
> > * CIFAR-10: GJS performs on par with the recent method of Ma et al. (ICML 2020) on 20% asymmetric noise, and outperforms all baselines on 40% noise.
> > * CIFAR-100: GJS outperforms all other baselines on 20% asymmetric, and performs close to the best (64.67% vs 63.7%) on 40% while the third best is far behind (57.8%).
> > * WebVision: In this real-world noise case, our method gives state-of-the-art performance on both the evaluation sets of WebVision and ILSVRC12, outperforming more elaborate methods like DivideMix.
> >
> > To further highlight the effectiveness of our simple method on more realistic noise, we perform experiments on two additional real-world noise datasets that were used in Zhang et al. (ICLR 2021), i.e., ANIMAL-10N and Food-101N and on the synthetic 35% instance-dependent noise (Type I). The results on the real-world noise can be found below. For the synthetic noise, see response to reviewer xNPb. In all cases GJS outperforms the baselines with significant margins.
> >
> > For Food-101N, we follow the same training setup as the recent label correction method called Progressive Label Correction (PLC) by Zhang et al. (ICLR 2021), i.e., we use the same network, augmentation strategy, optimizer, batch size, number of epochs, and learning rate scheduling. We also report the mean and standard deviation on the test set for three runs with different seeds.
> >
> > Similarly, for Animal-10N we use the same training setup(network, optimizer, number of epochs, learning rate scheduling, etc) as PLC, but use cropping instead of random horizontal flipping as augmentation to reduce the risk of both augmentations being equal for GJS. The results are from three runs (same as PLC).
> >
> > The results for the baselines for ANIMAL-10N and Food-101N are from the PLC paper.
> >
> >  Method | &nbsp; ANIMAL-10N &nbsp;
> >  :---|:----:|
> >  CE | 79.4 ± 0.14
> >  SELFIE |  81.8 ± 0.09
> >  PLC | 83.4 ± 0.43
> >  GJS | **84.17 ± 0.07**
> >
> >  Method | &nbsp;&nbsp; Food-101N &nbsp;&nbsp;
> >  :---|:----:|
> >  CE | 81.67
> >  CleanNet |  83.95
> >  PLC | 85.28 ± 0.04
> >  GJS | **86.56 ± 0.13**
> >
> >
> >
> > ***Comparison with full pipeline, sample-selection, and label correction methods***
> >
> > We do compare GJS with two of the best performing full pipeline methods (ELR+ and DivideMix) on WebVision where we obtain one of the best reported results on the benchmark. We will revise the word “complementary” and mention that “robust loss functions can be considered *orthogonal* to full-pipeline methods.
> >
> > Note that we compare with the very recent label correction method PLC on both ANIMAL-10N and Food-101N. Furthermore, we compare with DivideMix, a state-of-the-art sample selection method, on the WebVision dataset.
> >
> > ***References***
> >
> > Ma et al. (ICML 2020): Normalized Loss Functions for Deep Learning with Noisy Labels
> >
> > Miyato et al. (ICLR 2016): Distributional Smoothing with Virtual Adversarial Training
> >
> > Zhang et al. (ICLR 2021): Learning with Feature-Dependent Label Noise: A Progressive Approach.
> >
> > Wei & Liu. (ICLR 2021): When optimizing f-divergence is robust with label noise

---

> > > ### Comment · Reviewer_HdUW · 2021-08-11
> > > **Concernd addressed**
> > >
> > > Thank you for your clarify. My concerns are well addressed.

---

### Official Review · Reviewer_xNPb · 2021-07-16

**Rating:** 6
**Confidence:** 3

**Summary:**

The paper proposed to use JS divergence as a loss function (JS) to address the underfit problem of provably-robust loss functions. Further, the authors generalize the JS loss function (GJS) to include multiple distributions to encourage consistency during learning with noisy labels. They demonstrate that JS interpolates MAS and CE loss, and they proved the noisy robustness property of the JS and GJS. Extensive experiments are provided to show the effectiveness of the proposed loss functions.

**Limitations And Societal Impact:**

Limitations and societal impact have been addressed.

**Main Review:**

The paper is well-written and clearly conveys the authors' thesis.

pros:
1. observing the significant consistency reduction around noisy data samples when the model overfits noise is of theoretical interest and practical importance.
2. the authors proved the noise robustness properties of the proposed novel loss functions, JS and GJS. They further provided a detailed investigation and ablation study to support their theoretical claims and conjectures.
4. they carried out extensive experiments and demonstrated the proposed loss functions could achieve state-of-the-art performance under symmetric and asymmetric label noise.

cons/questions:
1. the consistency regularizer may lead the network to learn a more smooth prediction function. This is similar to the effects of some adversarial defense methods. I wonder if the GJS will lead to better adversarial robustness?
2. the decrease of consistency when overfitting to noise is an important phenomenon. One major hypothesis of this work is that there is a causal relationship between clean validation accuracy and consistency. It would be better to provide consistency plots for JS and GJS in addition to those validation plots.
3. In Figure 5, we see the validation accuracy decrease when M>3 for 60% noise rate case. This observation has not been adequately discussed. Adding more distributions would put more emphasis on consistency. Would it be the case that under heavy noise rate, there are not enough anchors to guide the model and balance the force of encouraging consistency?
4. it will be better to include experiments on instance-based label noise. In that case, the label noise is related to the feature and may introduce "fake consistency". It is of interest to see the performance of GJS under such a setting.

**Time Spent Reviewing:**

5

---

> ### Author Response · Authors · 2021-08-10
> **Thank you for your thorough feedback.**
>
> Thank you for your time and effort reviewing our paper. We appreciate the thoughtful feedback. We are happy you found our paper well-written, the theories and their corresponding claims and conjectures detailed and supported by evidence, the experiments extensive, the method effective and the results state-of-the-art.
>
> ***1. Does GJS improve robustness to adversarial attacks as well?***
>
> Interesting question! As we showed in Proposition 2, GJS can be written as a label-dependent JS term and a GJS consistency term. As you say, the consistency term may lead to learning a smoother prediction function, which could help against adversarial examples.
>
> There is a recent work by Jiang et al. (NeurIPS 2020) showing how pre-training a network with a contrastive loss is improving robustness to adversarial examples. They encourage consistency between the features of perturbations of the same image. However, they consider the perturbations to be both augmentations and adversarially perturbed images. This is related, but since they do contrastive learning, it consists of both consistency and separation.
>
> A very recent and more closely related work (but unpublished) is by Tack et al. (ArXiv 2021), that improves the robustness to adversarial attacks by using a Jensen-Shannon divergence based loss to encourage consistency between two differently augmented and adversarially perturbed versions of the same image. A difference is that they use CE+JS, but we believe GJS should perform as good, if not better.
>
> This is an important and related direction. So, we will include the corresponding discussion above and the related works in the revised version of the paper.
>
> ***2. Reporting consistency in addition to accuracy***
>
> That is a great suggestion. We will provide plots with validation and training accuracy, and consistency for clean and noisy examples for both JS and GJS in the final version of the paper. Since figures are not possible in the rebuttal, in the response to reviewer ogFS who raised a similar point, we provide a table reporting the consistency of CE, GCE and GJS for three different types of synthetic noise on CIFAR-100. The results show a clear correlation between robustness to noise and consistency.
>
> Note that we do not claim a causal relationship between general consistency and validation accuracy, but make a conjecture that consistency regularization might improve robustness to label noise. We found the correlation above between the validation accuracy and the consistency as motivation for looking into consistency regularization for learning with noisy labels.
>
> ***3. Discussion of ablation study of number of distributions (M) in Figure 5***
>
> Our best hypothesis is that our relatively small ResNet34 models used in these experiments are over-regularized when too many distributions are used due to the relatively-increased strength of the consistency term in equation 6.
>
> If we look at the training accuracy of clean examples on 60% noise, we see that M=3 has the highest accuracy, while M>3 and M=2 has lower. This suggests that consistency helps the learning of clean examples (M=3 vs M=2), but it also suggests that more distributions over-regularize and make it harder for the network to get the clean examples correct.
>
> We performed an experiment using a higher capacity network (ResNet26 from Nguyen et al. (ICLR 2020)) and it reduces the difference in terms of validation accuracy on 40% symmetric noise: 76.3, 79.4, 80.52, 79.5, 80.0, 78.9, for M=2,3,4,5,6,7 respectively.
>
> In the ablation study in Figure 5 and in the experiment above, we keep the learning rate, weight decay, $\pi_1$, etc fixed and only increase the number of distributions. Better performance could likely be achieved with a proper hyperparameter search. For example, for Table 1 where we search for $\pi_1$ the optimal value at 60% noise is 0.9 compared to 0.5, which was used here. This would make the JS term become closer to MAE (more robust) and not focus as much on consistency.
>
> We will incorporate this discussion in the revised version.
>
> ***4. Experiments on instance-based label noise***
>
> This is a great suggestion. Our state-of-the-art results on the real-world noise dataset WebVision shows that GJS handles instance-based label noise well. To further show this, we provide results on two other real-world datasets (Food-101N and ANIMAL-10N, see response to reviewer HdUW) and on instance-dependent synthetic noise (Type I) introduced in Zhang et al. (ICLR 2021) in the table below. This is for 35% noise on CIFAR-100. Similar to the results in Table 1, each method searches for the best learning rate and weight decay, then the best method-specific parameters are searched for. We report the mean and standard deviation of five runs for each method on the test set. Similarly to other noise types, GJS significantly outperforms CE and GCE.
>
>  Method   | &nbsp; &nbsp; Instance 35%  &nbsp; &nbsp;
>   :---|:----:|
>   CE | 62.31 ± 0.22
>   GCE |  65.44 ± 0.19
>   GJS | **68.40 ± 0.09**
>
> In the revised version of the paper, we will report results for more types of instance-dependent noise on both CIFAR-10 and CIFAR-100. Thanks for the suggestion!
>
> ***References***
>
> Zhang et al. (ICLR 2021): Learning with Feature-Dependent Label Noise: A Progressive Approach.
>
> Jiang et al. (NeurIPS 2020): Robust Pre-Training by Adversarial Contrastive Learning
>
> Tack et al. (ArXiv 2021): Consistency Regularization for Adversarial Robustness
>
> Nguyen et al. (ICLR 2020): SELF: Learning to Filter Noisy Labels with Self-Ensembling

---

### Official Review · Reviewer_6kDX · 2021-07-16

**Rating:** 7
**Confidence:** 4

**Summary:**

$\textbf{Summary and main contributions:}$

(1) The authors propose to use Jensen-Shannon divergence (JS) to learn with noisy labels, which is shown to interpolate between Cross-Entropy and MAE loss through a hyper-parameter.

(2) Based on the new observation that learning with noisy labels reduces the classifier's prediction consistency for noisy-labeled sets. To overcome this issue, the authors introduce multi-distribution generalization (GJS) which is shown to generalize JS with a consistency regularizer.

(3) Experiment on synthetic noisy labeled datasets and a real-world noisy dataset (WebVision) demonstrate the effectiveness of proposed methods.


**Limitations And Societal Impact:**

Yes, the authors have discussed their limitations in Section 5. I listed constructive suggestions above "Main Review".

**Main Review:**

$\textbf{Originality:}$ the authors introduce two novel robust methods to deal with label noise and alleviate the issue that "a network's prediction consistency got reduced for noisy-labeled data".

$\textbf{Quality:}$ the paper is overall well-written, although there are some additional issues, for example,

$\textbf{Clarity:}$ some notations and technical terms are not well-defined, also some experiment details are missed.

$\textbf{Additional concerns/suggestions:}$

(1) Some technical terms require additional explanations, for example, in Line 16-17, readers may not be familiar with "provably-robust loss functions" and "consistency". The presentation would be much better if the authors explain a bit more about these two terms since these two observations convey some important information about this work.

(2) Line 26, a typo: "labelled" -> "labeled".

(3) In the caption of Figure 1, it would be more accurate if the authors write "... as it overfits to label noise using CE loss". Besides, in Figure 1 (a), is the validation accuracy evaluated w.r.t. a clean or noisy validation set. I assumed that under these four levels of label noise, the learning rate settings are the same. For the plot w.r.t. relative large noise rate such as 60%, will a lower learning rate (after 200 epochs) avoids the corresponding classifier degrades so much? Although I believe that the order of performance will not change, the gap might not be so large?

(4) In Line 43, the notation of simplex $\Delta^{K-1}$ is not defined.

(5) In Line 49, it would be better if the authors firstly introduce a common real-world noise model where $\eta$ is usually instance-dependent, e.g., $\eta(x, y)$ for each instance/feature $x$ and class $y$. Then, state that in this work, the authors are interested in the instance-independent noise model where $\eta(x)=\eta$ and $\eta$ is the same for each class for a theoretical perspective.

(6) The explanation w.r.t. Figure 1 is delayed to Line 63, it might be better if the authors move this in the introduction section?

(7) In Line 77, a formal statement of the properties w.r.t. Jenson Shannon divergence is suggested.

(8) I suggest the authors explain more about the importance of the property "bounded" and "symmetric".

(9) The $m_{>1}$ in Proposition 2 feels like a weird notation since readers may find it hard to get the lingual relationship between $m$ and the weighted $\mathbf{p}^{(j)}$. $\bar{\mathbf{p}}_{>1}$ or another notation might make the presentation better?

(10) In Theorem 1, I feel like the requirement of $\forall x, f$ will somehow make $B_U-B_L$ much larger. Do authors think this Theorem 1 can be stated in this view: with probability at least $1-\delta$, we have $\mathbb{P}(B_L\leq \sum_{i=1}^K \mathcal{L}(e^{(i)}, x, f)\leq B_U)\geq 1-\epsilon.$
Filtering out extreme values in this view, the gap between $B_L$ and $B_U$ appear not so large. I think the proofs are not influenced with probability at least $1-\delta$.

(11) I didn't find any definition of $\mathbf{u}$ in Equ. (6).

(12)  In [1], the authors showed the robustness of the variation-form f-div functions w.r.t. two properly defined distributions (joint distribution and the product of marginal distributions via model prediction and label), where Jensen Shannon is a special case. It would be better if authors could discuss more relationships between these two methods (JS in this work and [1]) in the main paper, or with the experiment comparison.

(13) Experiment details: do authors use the same synthetic noise dataset for all methods? Or generate the noisy labels with a random seed for each training. If it is the latter case, do authors fix the random seed?

(14) In Line 220, the comparison with DivideMix [2] and ELR+ [3] are not reasonable. Although [2] and [3] adopt a stronger network, different architectures may result in different performances. For example, my previous observations show that Pre-Resnet 18 outperforms Resnet 34 in certain noisy label synthetic tasks, while the latter one has a stronger network.

$\textbf{Reference:}$

[1] Wei, Jiaheng, and Yang Liu. "When Optimizing f-Divergence is Robust with Label Noise." In International Conference on Learning Representations. 2020.

[2] Li, Junnan, Richard Socher, and Steven CH Hoi. "DivideMix: Learning with Noisy Labels as Semi-supervised Learning." In International Conference on Learning Representations. 2019.

[3] Liu, Sheng, Jonathan Niles-Weed, Narges Razavian, and Carlos Fernandez-Granda. "Early-Learning Regularization Prevents Memorization of Noisy Labels." Advances in Neural Information Processing Systems 33 (2020).

##############################################################

$\textbf{Post-Rebuttal}$

Thank authors for addressing all my concerns. And after the rebuttal, I decide to raise the rating from 6 to 7.

**Time Spent Reviewing:**

> 12 hours

---

> ### Author Response · Authors · 2021-08-10
> **Thank you for spending so much time on our paper. We appreciate the many great suggestions and thoughful feedback.**
>
> Thank you for your time and effort reviewing our paper. We appreciate the thoughtful feedback. We are happy that you think the paper is well-written and that you find our proposed methods novel and effective. Below we address your questions, suggestions and concerns.
>
> ***Clarifications, layout, typo, and notation.***
>
> We thank you for the good suggestions in (1), (2), (4), (5), (7), (8), (9), and (11). We will address these in the revised version of the paper.
>
> Regarding (8), due to lack of space in the main paper, we moved experiments and discussions of the symmetric and bounded properties to Section B.1 in the appendix. There we gradually construct JS out of KL terms to analyze these two properties.
>
>
> ***(3): Questions regarding the consistency observation in Figure 1***
>
> In Figure 1, we show clean validation accuracy and consistency on the training set. Using a noisy validation set here could lead to inconclusive results if the network overfits to noise.
>
> All noise levels in Figure 1 use the same training setup (e.g., loss, learning rate, learning rate scheduling, weight decay, even weight initialization, etc.). The learning rate is scaled by a factor of 0.1 at 50% and 75% of the total number of epochs (we will clarify this in the figure caption). This is the same learning rate scheduling used for all our results on the CIFAR datasets. Therefore, Figure 1 shows that when the learning rate is reduced, the network is more prone to overfitting. In fact, Tanaka et al. (CVPR 2018) found that a high learning rate makes it harder for the network to overfit to noise.
>
> In the tables below, we compare using a constant learning rate (“Constant”) to when reducing the learning rate (“Reduced”). We observe that using a constant learning rate makes the validation accuracy significantly worse for 0% otherwise similar. Furthermore, the consistency on clean examples is significantly worse for 0-40% and similar for 60%. For noisy labelled examples, a constant learning rate results in significantly higher consistency, which is in line with Tanaka et al. (CVPR 2018).
>
> Overall, the same patterns are observed for constant learning rate: 1) the consistency of noisy examples is much lower than the consistency of clean examples, and 2) the consistency is reduced for higher noise levels.
>
>
> **Validation Accuracy**
>
> Method | 0% | 20% | 40% | 60%
> ---|---|---|---|----
> Constant | 69.3 | 49.1 | 30.0 | 18.1
> Reduced | 78.2 | 50.1 | 32.6 | 14.2
>
> **Consistency Clean**
>
> Method | 0% | 20% | 40% | 60%
> ---|---|---|---|----
> Constant | 69.0 | 61.6 | 54.9 | 47.3
> Reduced | 89.3 | 75.2 | 61.9 | 49.4
>
> **Consistency Noisy**
>
> Method | 0% | 20% | 40% | 60%
> ---|---|---|---|----
> Constant | N/A | 51.9 | 48.6 | 41.9
> Reduced | N/A | 43.7 | 38.7 | 35.1
>
>
> ***(10): Theorem 1***
>
> Investigating if we can formulate the theorems in terms of probabilistic bounds is an interesting idea! Indeed, GJS’s upper bound should be quite unlikely to happen (all augmented samples give rise to uniquely-different but confident predictions). This direction might help to find an even tighter but highly-likely bound. We will look into this during the discussion phase. Thanks!
>
> ***(12): Connection to f-divergences***
>
> Wei & Liu. (ICLR 2021) is a very relevant concurrent work. We have briefly discussed it in the extended related work section in the appendix (Section D). There are strong connections between the Jensen-Shannon divergence in our and their work. The main difference is that we empirically and theoretically analyze the role of $\pi$, which they treat as a constant. Varying $\pi$ is important because it leads to:
> * better empirical performance: For our experiments on CIFAR, we provide the hyper-parameters used in Table 7, from which we can see that the optimal $\pi$ is equal to their setting ($\pi=\frac{1}{2}$) in only 3/14 cases.
> * interesting theoretical connections to related work: In Proposition 1, we show that the JS loss has CE and MAE (Ghosh et al. (AAAI 2017)) as asymptotes when $\pi$ goes to zero and one, respectively. This causes an interesting trade-off between learnability and robustness as discussed in Section 4.3 in the paper.
>
> Furthermore, we consider the generalization to more than two distributions which have proved helpful while Wei & Liu. (ICLR 2021) only study two distributions.
>
> We will add a more thorough discussion of this connection in the revised version of the main paper. Thanks for bringing the importance of it to our attention.
>
> ***(13): Experiment details***
>
> The same synthetic noisy dataset is used for all methods. In fact, we go to great lengths to try to make the comparison with the baselines as thorough and fair as possible. For all results on the CIFAR datasets in the main paper and the appendix, our synthetic noise is stochastically generated using fixed random seeds across methods. We set the seed such that it results in the same train-validation split, noisy labels, order of data from the dataloader, and network weight initialization for all methods. At testing, we train each method with the same five seeds. Therefore, the reported standard deviation in e.g., Table 1, is not only due to a different sampled set of noisy labels, but also due to a different train-validation split and weight initialization.
>
> ***(14): Comparison with DivdeMix and ELR+***
>
> We agree that in general it is not possible to suppose a stronger network performs better when learning with noisy labels, especially since high-capacity models will have more noise-fitting ability. However, in this specific case, we have evidence that DivideMix performs worse with Inception-ResNet-V2 compared to ResNet-50, see Table 2. Also, we have experiments with the same network (ResNet-50), where GJS outperforms DivideMix on both WebVision and ILSVRC12, while being a much simpler method. We will revise the text to clarify this.
>
> ***References***
>
> Tanaka et al. (CVPR 2018): Joint Optimization Framework for Learning with Noisy Labels
>
> Wei & Liu. (ICLR 2021): When optimizing f-divergence is robust with label noise
>
> Ghosh et al. (AAAI 2017): Robust loss functions under label noise for deep neural networks.

---

> ### Author Response · Authors · 2021-09-01
> **Theoretical Improvement to GJS**
>
> Hi reviewer 6kDX,
>
> Thanks again for your time and effort.
>
> We want to let you know that we have improved our theoretical results for GJS’s upper bound of the excess risk. As you mentioned in (10), the bound for GJS becomes large due to considering any $f$. This causes an unlikely upper bound where all M-1 predictions would have to be on different vertices of the probability simplex (as inconsistent as it can be).
>
> With an assumption on the consistency of $f^*$ and $f^*_{\eta}$, we are able to significantly improve the bound. Please see the new comment to reviewer ogFS.

---

### Official Review · Reviewer_ogFS · 2021-07-16

**Rating:** 6
**Confidence:** 3

**Summary:**

This paper considers learning with noisy labels.  The author showed that the
Jensen-Shannon (JS) divergence sits in between the mean absolute error (MSE) and
the cross entropy (CS).  The key proposition is to use the generalized JS (GJS)
as the loss, which presents a trade-off between robustness with respect to the
noisy labels and to avoid under-fitting from which robust loss function (such as
the MSE) suffers.

**Main Review:**

Overall, this contribution has some theoretical analysis in section 2,
as well as experimental results on CIFAR/WebVision showing that the proposed GJS loss
leads to more accurate classification under the setting of noisy labels.
Based on these contributions, I tend to vote for acceptance.

Essentially, based on the experimental results, the most useful tool proposed
by the authors is the GJS (rather than JS) divergence as a robust loss. Based on
proposition 2, the loss has two terms. Minimizing the first term leads to
learnability, as it tries to align the target (noisy) label with the centroid
of the predictions; the second term leads to robustness, as it measures
the deviation of the predictions from their centroid.

The authors provided the recipe to choose the \pi factor,
which is important to trade-off between learnability and robustness and to make the experiments reproducible.

Theoretically, the main argument in section 2.5 is that the proposed GJS loss
is (trivially) bounded on both sides, and any bounded loss satisfies
the property stated in theorem 1: the deviation of the empirical risk
with respect to the label noise can be bounded. Note the GJS loss is
only robust at its extreme point when pi tends to be deterministic,
as "robustness" requires the deviation to be absolute zero.

These theoretical results can be improved by considering the parametric
family of the perturbation A(\eta), and specific label noise, which is not considered in this
paper. Is the GJS loss fitted for any noise labels and noisy perturbations
of the predictions? How are these two difference noise related to affect the learning?
Can we define \epsilon-robustness and bound the robustness
under the GJS loss? In general, the presented theoretical results
lack significance and can be enhanced.

Most of the experimental evaluation is based on the classification accuracy.
There should be some consistency measurements as shown in figure 1
(I trust the GJS should perform well because of its 2nd regularization term).

Minor comments:

L70
"Assuming causality, suggests that" -> please rewrite the sentence

L74: is $\Delta$ introduced before?

L78: "... is another crucial difference" with what?

L102: mention GCE, SCE are shorthand for ...

L107 and Figure 2: {L}_{JS} instead of JS (JS = 0 is $\pi$ goes to extreme)

L223: come from -> comes from

_____
**Post Rebuttal remarks**

The reviewer thanks the authors for the response, and for agreeing on the review and several extensions, and addressing all minor comments. The disagreement on the significance is more on whether this type of contribution on applying existing tools from information theory into current machine learning problems are interesting enough for NeurIPS.  Ideally, the statements can use the structure of the JS divergence, which will improve the significance. For example, section 2.5 only use the fact that the JS divergence is bounded. These aspects can definitely be improved.

Besides the current review, in preparing the next version (whether it is accepted or not), the authors are suggested to include proper reference (I leave the authors to do the literature review) on the following:
- The definition of GJS and other generalizations of JS;
- Existing analysis on the limit case of GJS;
- Existing applications of JS/GJS into machine learning/pattern recognition problems, and as loss functions.

**Time Spent Reviewing:**

3

---

> ### Author Response · Authors · 2021-08-10
> **Thank you for your comments and insightful suggestions!**
>
> Thank you for your time and effort reviewing our paper. We appreciate the thoughtful feedback. We are happy that you acknowledge the contributions of our work and suggest acceptance.
>
> ***Theory***
>
> **Significance of the theory**
>
> While it is trivial to bound GJS for a particular one-hot label, it is non-trivial to find a tight upper bound for the sum of GJS over different classes: $\sum_{i=1}^K GJS(e^{(i)}, p^{(1)}, p^{(2)}, .., p^{(M-1)})$. Furthermore, our theoretical results in Proposition 1 relate JS to CE and MAE and therefore draws interesting connections to related work. Moreover, we derive gradients of the loss with respect to the logits which makes it possible to analytically compare with other robust losses such as Generalized Cross-Entropy (GCE). Finally, we investigate a gradual construction of GJS in terms of KL terms in Section B.1 in the appendix.
>
> **Parametric perturbations**
>
> The reason we have focused on image augmentations is purely empirical, since such natural perturbations are more conducive to better performance in general than simple perturbations such as norm-constrained or Gaussian and thus are more common in ML practice. However, we agree that, when it comes to consistency regularization, the theoretical study of robustness to label noise can be strengthened if we consider the consistency over a parametric family of perturbations. That is indeed an interesting idea for future extension of the work! Thanks!
>
> **$\epsilon$-robustness**
>
> The idea of formulating the robustness theorems using probabilistic but potentially even tighter bounds (similar to PAC analysis) is also a good idea. We will look into this during the discussion phase. Thanks!
>
> **Relation between label noise and noisy perturbation**
>
> This can indeed be another way to look at how input perturbations can improve the robustness to label noise. In fact, recently, Chen et al. (ICLR 2021), have shown that adding artificial label noise can help with robustness to inherent label noise. Therefore, it might be possible to similarly extend the argument for inducing artificial input noise to help the robustness against inherent label noise. The possibility makes this question another interesting future direction. However, we believe our presented analysis is providing a more direct explanation i.e. consistency regularization with a controlled trade-off between robustness and learnability.
>
> We find all these raised points important for follow-up works, therefore, we will discuss them throughout the revised version.
>
> ***Experiments***
>
> **Is the GJS loss fit for different label noise types and perturbations?**
>
> In the submission we have focused both the theory and experiments on class-dependent label noise, both symmetric and asymmetric, and have shown that GJS is fit for both types as it significantly improves the performance. While a theoretical analysis of other types of label noise cannot be done during the rebuttal, we did additional experiments indicating the efficacy of GJS for instance-dependent noise. See response to reviewers xNPb and HdUW.
>
> **Is the GJS loss fit for different perturbations?**
>
> We have empirically tried various image augmentations and GJS seems to benefit from all of them either weak or strong augmentations. See Table 5 in the main paper.
>
> **In addition to classification accuracy, also report consistency**
>
> This is a great suggestion, thanks! Below we provide consistency measures for CE, GCE and GJS for different noise types and rates on CIFAR-100 (“Instance” is a new synthetic noise that is dependent on the input). As you suspected, GJS performs well under this metric. We will provide a full table similar to Table 1 with consistency values in the revised version of the paper.
>
>  Method   | &nbsp; &nbsp; Sym 40%  &nbsp; &nbsp; | &nbsp; &nbsp; Asym 20% &nbsp; &nbsp; | Instance 35%
>   :---|:----:|:---:|:---:|
>   CE | 27.88 ± 0.34 | 36.59 ± 0.22 | 34.49 ± 0.39
>   GCE | 38.06 ± 0.41 | 37.61 ± 0.34 | 36.06 ± 0.26
>   GJS | **43.60 ± 0.26** | **42.03 ± 0.28** | **39.69 ± 0.27**
>
> **Minor Comments**
>
> Thank you for the minor comments. We will address all of these in the revision.
>
> **References**
>
> Chen et al. (ICLR 2021): Noise against noise: stochastic label noise helps combat inherent label noise.

---

> ### Author Response · Authors · 2021-09-01
> **Theoretical Improvements for GJS**
>
> Hi reviewer ogFS,
>
> Thanks again for your review. We hope you will find the time to review our rebuttal as well. As per your original review, we have now further enhanced our theoretical results for GJS’s upper bound of the excess risk as follows. Please note that this new theoretical result can be easily incorporated in the final version.
>
> First let us make a simple but useful assumption: motivated by the consistency observation in Figure 1, we assume the optimal classifier on clean data ($f^*$) is at least as consistent as the optimal classifier on noisy data ($f^*_{\eta}$). Formally, we assume
> $ \mathbb{E}\_{ \mathbf{x} }[\text{GJS}\_{ \boldsymbol{\pi''} }
> (\mathbf{x}, f^*)] \leq \mathbb{E}\_{ \mathbf{x} }[\text{GJS}\_{\boldsymbol{\pi''}}(\mathbf{x}, f^*_{\eta})]$.
>
> Note that $\text{GJS}\_{\boldsymbol{\pi''}}(\mathbf{x}, f)$ measures the "spread" of predictive distributions, hence is lower if the network is more consistent. Given this assumption, next, we obtain a new bound for the risk difference $R^{\eta}\_\mathcal{L}(f^*) - R^\eta_\mathcal{L}(f^*_{\eta})$ for GJS.
>
> $R^{\eta}_\mathcal{L}(f) = \mathbb{E}\_{\mathcal{D}\_{\eta}}[\mathcal{L}(e^{(\tilde{y})}, \mathbf{x}, f)] = \mathbb{E}\_{\mathbf{x},\tilde{y}}[\mathcal{L}(e^{(\tilde{y})}, \mathbf{x}, f)]$.
>
> Using the decomposed form of GJS in Proposition 2, we get
>
> $R^{\eta}_\mathcal{\text{GJS}\_{\boldsymbol{\pi}}}(f) = \mathbb{E}\_{\mathbf{x},\tilde{y}}[\text{JS}\_{\boldsymbol{\pi'}}(e^{(\tilde{y})}, \mathbf{m}_f)+ (1-\pi_1)\text{GJS}\_{\boldsymbol{\pi''}}(\mathbf{x}, f)] =  \mathbb{E}\_{\mathbf{x},\tilde{y}}[\text{JS}\_{\boldsymbol{\pi'}}(e^{(\tilde{y})}, \mathbf{m}_f)] + (1-\pi_1)\mathbb{E}\_{\mathbf{x}}[\text{GJS}\_{\boldsymbol{\pi''}}(\mathbf{x}, f)]$
>
> where $\mathbf{m}\_f = \sum_{m=2}^M \frac{\pi_m}{1-\pi_1}f(A(\mathbf{x}))$.
>
> We can now bound the risk difference.
>
> $\begin{align*} 0 &\leq R^{\eta}\_{\mathcal{\text{GJS}\_{\boldsymbol{\pi}}}}(f^*) - R^{\eta}\_\mathcal{\text{GJS}\_{\boldsymbol{\pi}}}(f^*_{\eta})
> \cr &=  \mathbb{E}\_{\mathbf{x},\tilde{y}}[\text{JS}\_{\boldsymbol{\pi'}}(e^{(\tilde{y})}, \mathbf{m}\_{f^*})] - \mathbb{E}\_{\mathbf{x},\tilde{y}}[\text{JS}\_{\boldsymbol{\pi'}}(e^{(\tilde{y})}, \mathbf{m}\_{f^*_{\eta}})] + (1-\pi_1) \Big(\mathbb{E}\_{\mathbf{x}}[\text{GJS}\_{\boldsymbol{\pi''}}(\mathbf{x}, f^*)]  - \mathbb{E}\_{\mathbf{x}}[\text{GJS}\_{\boldsymbol{\pi''}}(\mathbf{x}, f^*_{\eta})] \Big)
> \cr &\leq \mathbb{E}\_{\mathbf{x},\tilde{y}}[\text{JS}\_{\boldsymbol{\pi'}}(e^{(\tilde{y})}, \mathbf{m}\_{f^*})] - \mathbb{E}\_{\mathbf{x},\tilde{y}}[\text{JS}\_{\boldsymbol{\pi'}}(e^{(\tilde{y})}, \mathbf{m}\_{f^*_\eta})]
> \end{align*}$.
>
> where the lower bound comes from $f_{\eta}^*$ being the global optima of the risk on the noisy dataset and the upper bound comes from the assumption on the consistency of $f^*$ and $f^*_{\eta}$. Hence, we have upper bounded the risk difference for GJS in terms of a risk difference for JS.
>
> Interestingly, this result holds for any type of noise, e.g., symmetric, asymmetric and even instance-dependent. Therefore, we can use the bounds we derived for symmetric and asymmetric noise for JS to bound the risk difference for GJS above. Using Theorem 1 and 2 and Proposition 3 for M=2, we had in the paper
>
> $\begin{align*} 0 \leq R^{\eta}\_\mathcal{\text{JS}}(f^*) - R^{\eta}\_\mathcal{\text{JS}}(f^*_{\eta})
> &= \mathbb{E}\_{\mathbf{x},\tilde{y}}[\text{JS}\_{\boldsymbol{\pi'}}(e^{(\tilde{y})}, f^*(\mathbf{x}))] - \mathbb{E}\_{\mathbf{x},\tilde{y}}[\text{JS}\_{\boldsymbol{\pi'}}( e^{(\tilde{y})},f^*_{\eta}(\mathbf{x})) ]
> \cr &\leq \left \\{  \begin{array}{cl} \eta \Delta B^{\text{JS}} / (K-1) & : \ \text{symmetric noise} \\\ \Delta B^{\text{JS}} \mathbb{E}\_\mathcal{D}[1-{\eta}\_{yy}] & : \ \text{asymmetric noise} \end{array} \right. \end{align*}$
>
> where $\Delta B^{\text{JS}}= B^{\text{JS}}\_U - B^{JS}\_L = \sum_{k=1}^K \big( \text{JS}\_{\boldsymbol{\pi'}}(e^{(k)}, e^{(1)}) -  \text{JS}\_{\boldsymbol{\pi'}}(e^{(k)}, u)\big)$. Since the upper bound above holds for any $f, \mathbf{x}$, we get
>
> $\begin{align*} 0 \leq R^{\eta}\_\mathcal{\text{GJS}\_{\boldsymbol{\pi}}}(f^*) - R^{\eta}\_\mathcal{\text{GJS}\_{\boldsymbol{\pi}}}(f^*_{\eta})  &\leq \mathbb{E}\_{\mathbf{x},\tilde{y}}[\text{JS}\_{\boldsymbol{\pi'}}(e^{(\tilde{y})}, \mathbf{m}\_{f^*})] - \mathbb{E}\_{\mathbf{x},\tilde{y}}[\text{JS}\_{\boldsymbol{\pi'}}(e^{(\tilde{y})}, \mathbf{m}\_{f^*_{\eta}})]
> \cr &\leq  \left \\{ \begin{array}{cl} \eta \Delta B^{\text{JS}}/ (K-1) & : \ \text{symmetric noise}
> \cr \Delta B^{\text{JS}}\mathbb{E}\_\mathcal{D}[1-\eta_{yy}] & : \ \text{asymmetric noise} \end{array} \right. \end{align*}$.
>
> Hence, we have bound the GJS risk difference by a similar bound as the corresponding risk difference bound for JS.
>
> Why is this interesting?
> * With our initial theory in the paper, the upper bound of the risk difference of GJS became worse for increasing M (number of distributions). This was due to the upper bound of GJS in Proposition 3 growing with increasing M. This upper bound was unlikely to happen since all M-1 predictions would have to be on different vertices of the probability simplex. With our new results, the upper bound is independent of M.
>  * GJS now has the same risk difference upper bound as JS. This was not the case before, as GJS had a worse bound.
>  * It gives rise to an observation that over-regularizing the consistency of $f^*_{\eta}$ could increase the risk difference by making $ \mathbb{E}\_{ \mathbf{x} }[\text{GJS}\_{ \boldsymbol{\pi''} }
> (\mathbf{x}, f^*)] > \mathbb{E}\_{ \mathbf{x} }[\text{GJS}\_{\boldsymbol{\pi''}}(\mathbf{x}, f^*_{\eta})]$.
>
> Even though our assumption on consistency is for the global optima $f^*$ and $f^*_{\eta}$ on the full data distribution, we have empirically observed this to be true on the training set for our local optima networks which is reached when trained with GJS for a large number of epochs.
>
> We hope you will find these results interesting.

---

### Author Response · Authors · 2021-08-10
**Comment to the reviewers**

We would like to thank all the reviewers for their constructive and thoughtful feedback. While the average rating is already on the acceptance side, the thoughtful reviews made us hopeful that we can also engage in a fruitful discussion to alleviate the remaining concerns (if any) and thereby increase the average rating even further into the acceptance range.

For that, we have tried to answer all the concerns of the reviewers which come below as separate answers to the reviewers. Therefore, we would be grateful if the reviewers, during the discussion phase, point out any remaining issue that keeps them from increasing their score so that we can try to alleviate those through active discussions. On the other hand,  if the concerns are (mostly) addressed, it will be helpful that this is reflected in your final ratings.

Thanks again for the time and effort you have put in these constructive reviews and looking forward to an active discussion!

---

> ### Author Response · Authors · 2021-08-18
> **Thank you for the quick responses.**
>
> We would like to thank reviewer 6kDX and HdUW for taking the time and effort to read the rebuttal, reflect, and increase the ratings.
>
> Reviewer ogFS and xNPb, we are more than happy to discuss any remaining concerns you might have.
>
> Thanks again.

---

### Decision · Program_Chairs · 2021-09-27

**Decision:**

Accept (Poster)

**Comment:**

The reviewers deemed the paper of interest. I would like to thank the authors for providing focused discussion content for both theoretical and experiments, which contributed to explain further the paper's content and its improvements with respect to its submission history.

In the revised version, the authors should include the generalisation to Theorem 1 proposed at discussion phase to cover asymetric noise and in experiments, find a place to report consistency and instance-based label noise, a short discussion vs Wei + Liu's approach as developed in the discussion (point (12) 6kDX) and complete references to be more extensive. This is extremely important given the context of the paper and the richness of the relevant literature in the past decade or so.

AC.